# CHARTMIMIC: EVALUATING LMM'S CROSS-MODAL REASONING CAPABILITY VIA CHART-TO-CODE GENERATION

**Cheng Yang**[1*]  **Chufan Shi**[1*]  **Yaxin Liu**[1*]  **Bo Shui**[1*]  **Junjie Wang**[1*]
**Mohan Jing**[1]  **Linran Xu**[1]  **Xinyu Zhu**[1]  **Siheng Li**[1]
**Yuxiang Zhang**[1]  **Gongye Liu**[1]  **Xiaomei Nie**[1]  **Deng Cai**[2]  **Yujiu Yang**[1†]
[1]Tsinghua University   [2]Tencent AI Lab
chartmimic@gmail.com

## ABSTRACT

We introduce a new benchmark, ChartMimic, aimed at assessing the visually grounded code generation capabilities of large multimodal models (LMMs). Chart-Mimic utilizes information-intensive visual charts and textual instructions as inputs, requiring LMMs to generate the corresponding code for chart rendering. Chart-Mimic includes $4,800$ human-curated (figure, instruction, code) triplets, which represent the authentic chart use cases found in scientific papers across various domains (e.g., Physics, Computer Science, Economics, etc). These charts span $18$ regular types and $4$ advanced types, diversifying into $201$ subcategories. Furthermore, we propose multi-level evaluation metrics to provide an automatic and thorough assessment of the output code and the rendered charts. Unlike existing code generation benchmarks, ChartMimic places emphasis on evaluating LMMs' capacity to harmonize a blend of cognitive capabilities, encompassing visual understanding, code generation, and cross-modal reasoning. The evaluation of 3 proprietary models and $14$ open-weight models highlights the substantial challenges posed by ChartMimic. Even the advanced GPT-4o, InternVL2-Llama3-76B only achieved an average score across Direct Mimic and Customized Mimic tasks of $82.2$ and $61.6$, respectively, indicating significant room for improvement. We anticipate that ChartMimic will inspire the development of LMMs, advancing the pursuit of artificial general intelligence.

## 1 INTRODUCTION

Code generation (Sun et al., 2024) is a rather demanding task that requires advanced abstract thinking and logical reasoning, reflecting the unique intelligence of human beings. Recently, advances in artificial general intelligence (AGI) have demonstrated the potential of large foundation models (Google, 2023; OpenAI, 2024; Anthropic, 2024; AI@Meta, 2024) to solve the tasks that are once the exclusive domain of human abilities (Achiam et al., 2023; Zhu et al., 2024). However, existing code generation benchmarks (Chen et al., 2021; Austin et al., 2021; Hendrycks et al., 2021; Lai et al., 2023) solely use text as input, while humans receive information from multiple modalities when coding (Liang et al., 2023; Fan et al., 2024). Such real-life scenarios have yet to be fully explored.

Taking a common scene in Fig. 1, researchers often need to write code for data visualization and may already have preferred chart templates at hand. However, they usually lack either the source code or the expertise to reproduce these chart templates. As a result, they turn to large multimodal models (LMMs) as assistants to aid in code generation. In this scenario, coding for scientific charts entails *code generation grounded on visual understanding* (i.e., chart-to-code generation), which necessitates LMMs to integrate a variety of advanced cognitive capabilities, including visual understanding, code generation, and cross-modal reasoning. Therefore, evaluating the performance of LMMs on this real-world task also enables researchers to pinpoint potential areas for improving models' capabilities.

---

[*]Equal Contribution.  Data and code are available at https://github.com/ChartMimic/ChartMimic.
[†]Corresponding author.

Figure 1: The real-world example. LMMs assist scientists and researchers in understanding, interpreting and creating charts during the reading and writing of academic papers. These models serve as assistants that enhance the comprehension and presentation of data in scholarly communications.

To this end, we present **ChartMimic** (Fig. 2), a multimodal code generation benchmark. ChartMimic is characterized by its *(1) information-intensive visual inputs*, *(2) diverse chart types*, and *(3) multi-level evaluation metrics*. Specifically, compared to natural images, scientific charts convey nuanced semantic meanings through intricate visual logic, thereby exhibiting higher information density. Based on this, we define two tasks, Direct Mimic and Customized Mimic (Sec. 2.1), which utilize charts and textual instructions as inputs. These tasks challenge LMMs to generate the corresponding code for a given chart or to incorporate new data specified in the instructions, respectively. Through the collection of academic documents and scientific papers, we identify 22 commonly used chart types and 201 subcategories. Subsequently, we manually annotate a total of $4,800$ (figure, instruction, code) triplets for these types (Sec. 2.2). Furthermore, we establish automatic evaluation metrics from both high-level and low-level perspectives to thoroughly assess the performance of LMMs (Sec. 2.4).

We conduct the examination of 17 LMMs on ChartMimic (Sec. 3.2), including 3 proprietary models and 14 open-weight models across parameter sizes from 2.2B to 76.0B. We observe that while several open-weight models can match the performance of proprietary models such as GPT-4o on public leaderboards (OpenCompass, 2023), a significant performance gap still persists on ChartMimic. Specifically, the best open-weight model, InternVL2-Llama3-76B, lags behind GPT-4o, with an average score gap of $20.6$ on two tasks, indicating substantial room for improvement in the open-source community. Our analysis of prompting methods (Sec. 4.2) reveals that GPT-4o can improve itself through self-reflection, which is a key manifestation of System 2 reasoning (Sloman, 1996; Kumar et al., 2024). This discovery highlights the vital role that the System 2 reasoning plays in LMMs when tackling the complex challenges presented by ChartMimic. Meanwhile, Correlation analysis (Sec. 4.3) demonstrates a high correlation between our automatic metrics and human evaluation, validating the effectiveness of these metrics. Further error analysis (Sec. 4.4) reveals that hallucinations notably hinder the performance of LLMs on ChartMimic, as they lead to the insertion of non-existent text into ground-truth figures and confusion between similar types of charts.

We envision ChartMimic as a comprehensive suite of benchmarks designed to guide researchers in understanding their LMMs' capabilities. By providing a comprehensive evaluation framework, ChartMimic aims to facilitate the growth of foundation models for the community, offering insights into various aspects such as visual understanding, code generation, and cross-modal reasoning.

## 2 THE CHARTMIMIC BENCHMARK

In this section, we first introduce the definition of two tasks involved in ChartMimic (Sec. 2.1), and then delineate the data curation process (Sec. 2.2). Subsequently, we conduct a quantitative analysis to assess the quality and diversity of ChartMimic (Sec. 2.3), establish evaluation metrics (Sec. 2.4), and compare it with existing related benchmarks (Sec. 2.5).

### 2.1 TASK DEFINITION

LMMs' ability to generate chart-rendering code demonstrates their visual understanding, coding, and cross-modal reasoning skills. Specifically, given the chart $X$ and the instruction $I$, the LMM $f$ is expected to output the code $C$ that satisfies the requirements outlined in the instruction:

$$C = f(X, I). \tag{1}$$

As shown in Fig. 2, based on the information provided in the instructions, we propose two tasks:

**Direct Mimic.** The LMMs are tasked to directly generate code that can reproduce the provided chart, thereby assessing their visual comprehension and reasoning capabilities.

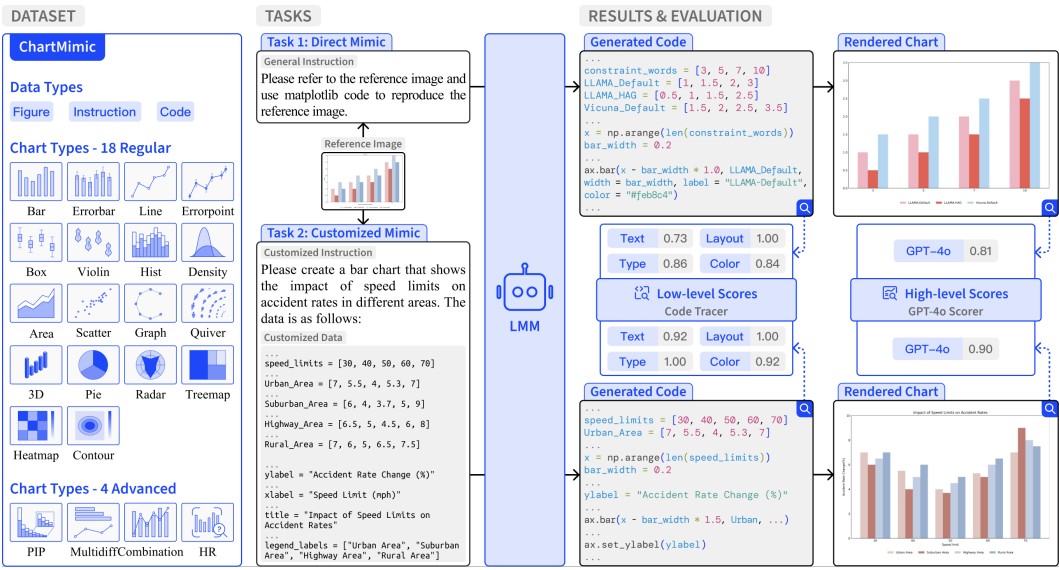

Figure 2: The pipeline of ChartMimic. We provide $4,800$ human-curated (figure, instruction, code) triplets. We use ChartMimic to evaluate LMMs' proficiency in the multimodal chart-to-code generation, resulting in both high-level and low-level evaluation results.

**Customized Mimic.** The LMMs are requested to generate code for a new chart that incorporates customized data provided in the instruction while preserving the original chart's aesthetic and design, assessing their ability to integrate visual and textual information.

After obtaining the code generated by LMMs, we execute it to render the corresponding chart and subsequently compare its similarity with the ground-truth chart. Illustrative examples of the two tasks are shown in Fig. 2. To accommodate the defined tasks above, we propose ChartMimic, a benchmark designed to evaluate the comprehension of charts and their conversion into executable code.

## 2.2 DATA CURATION PROCESS

ChartMimic distinguishes itself through $4$ fundamental considerations: *(1) diversity of chart types, (2) balance of chart complexity, (3) reduction of data leakage, (4) integration of authentic user requirements*. We keep on these four principles to complete the data curation for ChartMimic through a five-step pipeline. Here, we provide an overview here and more details in Appendix B.

**General Filtering.** We scrape figures from source files of publications on arXiv[1] that hold a CC BY 4.0 license with a publication date after February 2024 with PDF format. This yields approximately $174,100$ figures across various domains (e.g., Physics, Computer Science, Economics, etc). We then filter the figures based on the criteria of how designers select inspiring visualization examples, including clarity and visual appeal, color schemes (Bako et al., 2022; Quispel et al., 2018), and uniqueness of the chart within its category, resulting in a refined collection of about $15,800$ figures.

**Diversity and Information Density Filtering.** The filtering process involves two stages. In the first stage, we establish a data pool and categorize chart types. Charts with significant differences in complexity or information density are added to ensure diversity and effective communication (Bako et al., 2022). In the second stage, five experts from various fields independently evaluate the data, creating separate selection pools. We preserve the intersection of their selections and finalize the set through a voting process. This meticulous approach refines our collection to $279$ charts.

**Manual Selection and Filling.** In addition to sourcing from arXiv, we also collect charts from various platforms, including the Matplotlib gallery, Stack Overflow, and plotting-related forums on Twitter and Reddit. These charts are selected for their unique styles not represented in our arXiv curated $279$ charts. To mitigate the risk of data leakage, we rigorously process the data and color styling of these charts, replacing existing color schemes with those unseen in our data pool while maintaining their aesthetic appeal. Consequently, we obtain $600$ prototype charts for ChartMimic.

---

[1] https://arxiv.org/

Table 1: Statistics of ChartMimic. We measure code length in terms of tokens, utilizing the Llama3 tokenizer. In the level count, "A/B/C" denotes the number of "easy/medium/hard" level, respectively.

| Type | Bar | Line | ErrorBar | Heatmap | Box | Scatters | Hist | Radar | 3D | Pie | ErrorPoint | Violin |
|---|---|---|---|---|---|---|---|---|---|---|---|---|
| Count | 320 | 280 | 120 | 120 | 100 | 100 | 80 | 80 | 80 | 80 | 80 | 80 |
| Subcategories | 16 | 8 | 12 | 4 | 6 | 4 | 3 | 6 | 5 | 8 | 5 | 3 |
| Code Length (AVG.) | 689.6 | 794.0 | 681.2 | 685.8 | 689.0 | 655.0 | 529.6 | 779.8 | 655.4 | 418.4 | 624.3 | 975.6 |
| Code Length (STD.) | 237.8 | 244.4 | 144.7 | 258.7 | 228.2 | 253.0 | 147.1 | 144.3 | 241.4 | 99.5 | 197.7 | 252.3 |
| Level Count | 176 / 120 / 24 | 256 / 24 / 0 | 68 / 52 / 0 | 0 / 76 / 44 | 60 / 40 / 0 | 80 / 20 / 0 | 52 / 28 / 0 | 52 / 28 / 0 | 8 / 48 / 24 | 52 / 28 / 0 | 44 / 28 / 8 | 32 / 44 / 4 |

| Type | Area | Contour | Density | Graph | Quiver | Treemap | Combination | HR | Muiltidiff | PIP | Total | |
|---|---|---|---|---|---|---|---|---|---|---|---|---|
| Count | 80 | 80 | 80 | 80 | 80 | 80 | 120 | 100 | 100 | 80 | 2400 | |
| Subcategories | 2 | 3 | 4 | 4 | 4 | 4 | 30 | 25 | 25 | 20 | 201(101 + 100) | |
| Code Length (AVG.) | 774.4 | 489.4 | 540.0 | 564.5 | 893.4 | 342.2 | 697.4 | 718.9 | 798.2 | 1083.9 | 696.0 | |
| Code Length (STD.) | 161.8 | 87.8 | 104.7 | 117.5 | 631.0 | 36.3 | 163.6 | 265.5 | 271.2 | 290.1 | 278.4 | |
| Level Count | 52 / 28 / 0 | 0 / 28 / 52 | 44 / 32 / 4 | 56 / 24 / 0 | 0 / 52 / 28 | 52 / 28 / 0 | 12 / 76 / 32 | 4 / 16 / 80 | 0 / 48 / 52 | 0 / 0 / 80 | 1100 / 868 / 432 | |

**Code and Instruction Writing.** We propose to manually write codes and instructions for ChartMimic based on the collected 600 prototype charts. Initially, annotators are tasked with meticulously reproducing the 600 prototype charts using Python code, resulting in a set of 600 (figure, code, instruction) triplets for the Direct Mimic task. Although other coding languages such as JavaScript and R can be used to create charts, current LMMs perform poorly when doing chart-to-code using these other languages (Sun et al., 2024). Therefore, we focus on Python code generation in the current version of ChartMimic. Subsequently, to simulate the scenario of Customized Mimic, annotators are instructed to modify the chart data in the Direct Mimic task by integrating new data from various domains. They are then required to modify the corresponding code and instructions, leading to the 600 (figure, code, instruction) triplets for the Customized Mimic task. Consequently, we establish the ChartMimic benchmark, comprising 1,200 high-quality seed data.

**Data Augmentation.** Following the development of seed triplets, we initiate a process of manual data augmentation. Annotators are tasked with altering various elements of each seed triplet, including data, color schemes, mark styles, etc., to produce augmented triplets. For each seed triplet, we create 3 additional augmented triplets. This process enhances our dataset, yielding a total of 4,800 triplets that reflect a wide range of realistic and practical chart use cases.

## 2.3 DATA STATISTICS AND DIVISION

The (figure, code, instruction) triplets for both Direct and Customized Mimic tasks share the same figure. Therefore, we detail the data statistics for the 2,400 triplets in the Direct Mimic task. As depicted on the left side of Fig. 2, ChartMimic encompasses a total of 22 categories, with 18 types of regular charts and 4 types of advanced charts. For the 18 regular chart types, we identify 101 subcategories, with the definitions and examples of each subcategory provided in Appendix D. The advanced chart types such as Plot-in-Plot (PIP), Multidiff, and Combination are distinct forms of amalgamating multiple chart sets into a singular chart. Given the variety of their internal combination elements, each of these can be treated as a unique subcategory. Meanwhile, the Hard-to-Recognize (HR) category encapsulates unclassifiable charts, with each chart in the seed data being considered a category unto itself. When we factor in the additional 100 subcategories represented by these advanced chart types, ChartMimic encompasses a total of 201 subcategories. This extensive diversity underscores the comprehensive nature of our benchmark. We employ Llama3 (AI@Meta, 2024) tokenizer to measure the code length. As shown in Tab. 1, ChartMimic has an average code token length of 696.0 with a standard deviation of 278.4. Additionally, we manually categorize charts into three complexity levels: easy (1,100), medium (868), and hard (432). The detailed categorization criteria and assessment methodology are thoroughly documented in Appendix B.

We further divide the 4,800 examples of ChartMimic into two subsets: test and testmini set. The test set comprises 3,600 examples, while the testmini set is composed of 1,200 examples. The testmini set is designed for rapid model development validation. Our partitioning strategy ensures each chart type is proportionally represented, preserving a distribution in the testmini set that closely aligns with the test set. Detailed comparative experimental results, discussed in Appendix C, demonstrate the consistency across two subsets. Unless otherwise stated, we report results on the testmini set.

## 2.4 EVALUATION METRICS

For tasks within ChartMimic, an appropriate evaluation necessitates comparing the visual similarity between the generated and ground-truth figures. To achieve this, we propose multi-level metrics (i.e., high-level and low-level) to assess the similarity at different granularities. Specifically, the high-level metric encompasses the GPT-4o Score, and the low-level metric encompasses Text, Layout, Type and

Table 2: A comparison of our proposed ChartMimic to other benchmarks. "I" and "NL" indicate "Image" and "Natural Language" respectively.

| Benchmarks | Source | # of Chart Types | # of Test Instances | Input Format | Output Format | Evaluation Metric |
|---|---|---|---|---|---|---|
| *Chart Understanding Benchmarks* | | | | | | |
| ChartQA (Masry et al., 2022) | Human Curated | 3 | 10K | I+NL | NL | Accuracy |
| Chart-to-Text (Kantharaj et al., 2022) | Crawl | 6 | 44K | I+NL | NL | Match-based |
| ChartSumm (Rahman et al., 2023) | Human Curated | 3 | 84K | I+NL | NL | Match-based |
| CharArXiv (Wang et al., 2024b) | Human Curated | 18 | 93K | I+NL | NL | GPT-4 Score |
| *Code Generation Benchmarks* | | | | | | |
| HumanEval (Chen et al., 2021) | Human Curated | - | 164 | Code | Code | Pass Rate |
| MBPP (Austin et al., 2021) | Human Curated | - | 500 | NL+Code | Code | Pass Rate |
| MMCode (Li et al., 2024b) | Crawl | - | 263 | I+NL | Code | Pass Rate |
| MatPlotBench (Yang et al., 2024) | Human Curated | 13 | 100 | NL | Code | GPT-4 Score |
| Plot2Code (Wu et al., 2024a) | Crawl | 15 | 132 | I+NL | Code | Multi-Level |
| Design2Code (Si et al., 2024) | Crawl | HTML | 484 | I+NL | Code | Multi-Level |
| ChartMimic (Ours) | Human Curated | 22 | 4,800 | I+NL | Code | Multi-Level |

**Color Score.** We compute the average score between the **high-level** and **low-level scores**, ranging from 0 to 100, as the **overall score**. The illustration of the multi-level metrics is depicted in Fig. 2.

**GPT-4o Score.** Following the successful use of large foundation models for evaluation in both natural language processing (Zheng et al., 2024; Li et al., 2023; Dubois et al., 2024) and computer vision (Zhang et al., 2023; Yang et al., 2024; Wu et al., 2024b), we adopt GPT-4o score as our high-level metric. Specifically, we input both the ground-truth figure and the generated figure into GPT-4o, and instruct it to output a high-level similarity score ranging 0 to 100. Although CLIP Score (Radford et al., 2021) is widely used for assessing image similarity, in our preliminary experiments, it has struggled to distinguish variations in types and other critical elements in charts, resulting in a low correlation with human evaluation results. Therefore, we use only GPT-4o Score as our high-level evaluation metric. The detailed description of GPT-4o Score can be found in Appendix E.

In addition to high-level similarity, evaluating the similarity among low-level elements between generated and ground-truth figures can provide a more fine-grained analysis of LMMs. Therefore, we propose to evaluate four key low-level elements in charts (Savva et al., 2011; Poco & Heer, 2017): text, layout, type and color. Extracting them from figures is a challenging task, as existing extraction models often fall short in terms of accuracy (Meng et al., 2024). Considering that figures are rendered based on the code, we design a code tracer to monitor the execution processes of the ground-truth code and generated code. The code tracer records the text, layout, type and color information. We calculate the F1 score of these elements as their corresponding score. And we average text, layout, type and color scores to get the low-level score. The methodology for obtaining these elements is briefly introduced as follows, with detailed descriptions provided in Appendix E.

**Text Score.** During the code execution process, for the function responsible for adding text elements to the rendered figures, the code tracer monitors it and records each text element parameter.

**Layout Score.** Layout refers to the arrangement of subplots in the figure. At the end of the code execution, the code tracer traverses all subplots in the figure and obtains their layout information.

**Type Score.** For each plot function, which serves the purpose of adding a specific chart type instance to the figure, the code tracer monitors its invocation status and records every invoked plot function.

**Color Score.** For each plot function, it will return the chart type instance at the end of the function invocation. The code tracer accesses the color attributes of these chart type instances.

It is important to note that code execution success rate is a standard metric for code generation tasks (Sun et al., 2024). We have implicitly incorporated this aspect into our high-level and low-level scores. Specifically, if the code fails to execute successfully, both the low-level and high-level scores are assigned a value of 0. Therefore, we do not separately weight it into the overall score.

## 2.5 COMPARISONS WITH EXISTING BENCHMARKS

To further distinguish the difference between ChartMimic and other existing ones, we elaborate the benchmark details in Tab. 2. From the chart understanding perspective, the prior benchmarks (Masry et al., 2022; Kantharaj et al., 2022; Rahman et al., 2023; Wang et al., 2024b) are only focused on questions about the data in the charts without necessitating the advanced cognitive capabilities of LMMs, which include visual understanding, code generation, and cross-modal reasoning. In the code generation aspect, HumanEval (Chen et al., 2021), MBPP (Austin et al., 2021) and MatPlot-Bench (Yang et al., 2024) only consider tasks with text inputs, which may not meet requirements

Table 3: The ChartMimic leaderboard with **Direct Mimic** task. The best scores are in **bold**. We also include the code execution success rate (Exec. Rate) and model size (Params).

| Model | Params | Exec. Rate | Low-Level | | | | | High-Level | Overall |
|---|---|---|---|---|---|---|---|---|---|
| | | | Text | Layout | Type | Color | Avg. | GPT-4o | |
| *Proprietary* | | | | | | | | | |
| GeminiProVision | - | 68.2 | 52.6 | 64.2 | 51.3 | 47.1 | 53.8 | 53.3 | 53.6 |
| Claude-3-opus | - | 83.3 | 66.8 | 83.1 | 49.9 | 42.1 | 60.5 | 60.1 | 60.3 |
| GPT-4o | - | **93.2** | **81.5** | **89.8** | **77.3** | **67.2** | **79.0** | **83.5** | **81.2** |
| *Open-Weight* | | | | | | | | | |
| IDEFICS2-8B | 7.6B | 49.0 | 6.2 | 33.1 | 9.2 | 9.0 | 14.4 | 17.6 | 16.0 |
| DeepSeek-VL-7B | 7.3B | 41.3 | 15.3 | 26.6 | 19.7 | 14.5 | 19.0 | 20.4 | 19.7 |
| LLaVA-Next-Yi-34B | 34.8B | 50.2 | 15.9 | 29.6 | 17.6 | 15.2 | 19.6 | 20.6 | 20.1 |
| LLaVA-Next-Mistral-7B | 7.6B | 59.7 | 14.0 | 31.1 | 19.8 | 17.8 | 20.7 | 21.3 | 21.0 |
| Qwen2-VL-2B | 2.6B | 47.0 | 20.1 | 29.5 | 21.3 | 17.9 | 22.2 | 23.4 | 22.8 |
| Cogvlm2-llama3-chat-19B | 19.2B | 50.5 | 21.3 | 31.8 | 18.4 | 17.0 | 22.1 | 24.5 | 23.3 |
| InternVL2-2B | 2.2B | 52.5 | 23.6 | 35.8 | 16.0 | 15.4 | 22.7 | 24.2 | 23.5 |
| Qwen2-VL-7B | 8.2B | 67.0 | 26.4 | 51.0 | 31.0 | 23.3 | 32.9 | 35.0 | 34.0 |
| InternVL2-4B | 4.2B | 66.2 | 34.7 | 51.7 | 25.2 | 23.6 | 33.8 | 38.4 | 36.1 |
| InternVL2-8B | 8.1B | 61.8 | 31.5 | 51.1 | 28.6 | 26.2 | 34.4 | 38.9 | 36.6 |
| MiniCPM-Llama3-V-2.5 | 8.4B | 80.3 | 30.7 | 49.6 | 38.6 | 27.6 | 36.6 | 42.1 | 39.4 |
| Phi-3-Vision-128K | 4.2B | 66.7 | 37.5 | 49.6 | 37.4 | 29.8 | 38.6 | 41.0 | 39.8 |
| InternVL2-26B | 26.0B | 69.3 | 39.2 | 58.7 | 35.9 | 31.8 | 41.4 | 47.4 | 44.4 |
| InternVL2-Llama3-76B | 76.0B | **83.2** | **54.1** | **74.5** | **49.2** | **41.5** | **54.8** | **62.2** | **58.5** |

in the era of LMMs. Recently, MMCode (Li et al., 2024b) attempted to create a benchmark for multimodal code generation, but the vision inputs for their task are still overly simple and only have a single pass rate evaluation metric. Design2Code (Si et al., 2024) and Plot2Code (Wu et al., 2024a) are the most similar ones to ours. Although they use multi-level evaluation metrics like ours, their test is crawled directly from the internet or existing datasets, which may pose a risk of data leakage. Our ChartMimic benchmark gives complex scientific charts as inputs, demanding capabilities on grounding visual understanding into code generation, and provides multi-level evaluation metrics.

## 3 EXPERIMENT

### 3.1 BASELINE SETUP

We benchmark 17 widely utilized proprietary and open-source models currently available in the field. For proprietary models, we consider 3 representative models: GPT-4o (OpenAI, 2024), Claude-3-opus (Anthropic, 2024) and GeminiProVision (Google, 2023). For the open-weight models, we choose 14 competitive models with total parameter size from 2.2B to 76.0B: InternVL2(2B, 4B, 8B, 26B, 76B) (Chen et al., 2023), Qwen2-VL(2B, 7B) (Wang et al., 2024a), Phi-3-Vision (phi, 2024), DeepSeek-VL-7B (Lu et al., 2024), LLaVA-Next(7B, 34B) (Li et al., 2024a), IDEFICS2-8B (Laurençon et al., 2024), MiniCPM-Llama3-V2.5 (Xu et al., 2024) and Cogvlm2-llama3-chat-19B (Wang et al., 2023a). We start with direct prompting, which provides the reference chart with direct instructions. The specific instructions and model configurations can be found in Appendix F.

### 3.2 MAIN RESULTS

We present the main results of 17 LMMs on ChartMimic. Tab. 3 and Tab. 4 show the results on the Direct Mimic and Customized Mimic task, respectively. The key findings are as follows:

**GPT-4o performs best among proprietary models, while InternVL2-Llama3-76B excels among open-weight models.** In proprietary models, GPT-4o achieves an overall score of 81.2 in Direct Mimic and 83.2 in Customized Mimic. Within open-weight models, InternVL2-Llama3-76B, reaches the overall score of 58.5 in Direct Mimic and 64.7 in Customized Mimic, which have compareble perfromance with proprietary model like Claude-3-opus (60.3 in Direct Mimic and 65.4 in Customized Mimic). Meanwhile, Phi-3-Vision-128K despite its 4.2B parameters, reaches the overall score of 39.8 in Direct Mimic and 42.1 in Customized Mimic, which outperforms LLaVA-Next-Yi-34B (20.1 in Direct Mimic and 35.3 in Customized Mimic). This indicates that even models with fewer parameters can achieve decent performance through a refined training process.

Table 4: The ChartMimic leaderboard with **Customized Mimic** task. The best scores are in **bold**. We also include the code execution success rate (Exec. Rate) and model size (Params).

| Model | Params | Exec. Rate | Low-Level | | | | | High-Level | Overall |
|---|---|---|---|---|---|---|---|---|---|
| | | | Text | Layout | Type | Color | Avg. | GPT-4o | |
| *Proprietary* | | | | | | | | | |
| GeminiProVision | - | 76.2 | 52.2 | 70.9 | 56.0 | 49.4 | 57.1 | 59.6 | 58.4 |
| Claude-3-opus | - | 88.2 | 75.2 | 86.8 | 54.1 | 44.3 | 65.1 | 65.7 | 65.4 |
| GPT-4o | - | **96.5** | **88.5** | **92.9** | **79.2** | **67.6** | **82.1** | **84.3** | **83.2** |
| *Open-Weight* | | | | | | | | | |
| Qwen2-VL-2B | 2.6B | 35.8 | 17.4 | 23.9 | 19.7 | 16.5 | 19.4 | 21.4 | 20.4 |
| Cogvlm2-llama3-chat-19B | 19.2B | 38.7 | 19.0 | 27.9 | 16.5 | 15.7 | 19.8 | 21.6 | 20.7 |
| LLaVA-Next-Mistral-7B | 7.6B | 49.0 | 20.0 | 32.0 | 22.6 | 19.9 | 23.6 | 24.7 | 24.2 |
| IDEFICS2-8B | 7.6B | 49.2 | 21.6 | 32.2 | 18.1 | 12.2 | 21.0 | 27.3 | 24.2 |
| InternVL2-2B | 2.2B | 49.3 | 22.2 | 35.4 | 20.0 | 18.1 | 23.9 | 27.8 | 25.9 |
| LLaVA-Next-Yi-34B | 34.8B | 64.2 | 28.7 | 44.8 | 32.9 | 27.7 | 33.5 | 37.1 | 35.3 |
| DeepSeek-VL-7B | 7.3B | 59.3 | 27.5 | 47.5 | 36.8 | 31.5 | 35.8 | 39.3 | 37.6 |
| Phi-3-Vision-128K | 4.2B | 67.8 | 29.7 | 52.5 | 42.3 | 36.5 | 40.3 | 44.0 | 42.1 |
| InternVL2-4B | 4.2B | 74.0 | 41.3 | 55.6 | 39.6 | 33.1 | 42.4 | 47.8 | 45.1 |
| Qwen2-VL-7B | 8.2B | 73.3 | 41.0 | 56.3 | 43.5 | 34.2 | 43.8 | 47.8 | 45.8 |
| InternVL2-8B | 8.1B | 73.0 | 43.1 | 54.4 | 39.9 | 35.4 | 43.2 | 48.9 | 46.1 |
| MiniCPM-Llama3-V-2.5 | 8.4B | 78.7 | 40.8 | 58.0 | 44.8 | 33.2 | 44.2 | 51.5 | 47.9 |
| InternVL2-26B | 26.0B | 73.7 | 43.9 | 62.3 | 43.5 | 34.3 | 46.0 | 51.1 | 48.6 |
| InternVL2-Llama3-76B | 76.0B | **89.8** | **57.8** | **79.0** | **63.5** | **50.5** | **62.7** | **66.7** | **64.7** |

**A performance gap persists between open-weight LMMs and proprietary ones.** Although several open-weight LMMs can exhibit performance comparable to GPT-4o across various benchmarks (OpenCompass, 2023), even the best-performing InternVL2-Llama3-76B falls short of achieving the performance of GPT-4o both Direct Mimic and Customized Mimic. This apparent performance disparity demonstrates the challenging nature of our ChartMimic benchmark for current open-weight LMMs. Moreover, open-weight LMMs even exhibit notable deficiencies in generating executable code, with a majority of them showing an execution rate below 75%. These findings highlight that there is still a considerable scope for the open-source community to enhance LMMs' capabilities regarding complex visual understanding, code generation and cross-modal reasoning.

**When given additional user-customized data, LMMs exhibit performance improvements.** In the Customized Mimic task, most LMMs get performance improvement compared to the Direct Mimic task, particularly in terms of Text Score. This enhancement can be attributed to the provision of user-customized data, which alleviates the burden on LMMs to recognize textual and data information within the charts. Notably, Cogvlm2-llama3-chat-19B and Qwen2-VL-2B experienced a decrease in their overall score and execution rate. This decline may arise from the reason that user-customized data increases the burden on the models to process different modalities of information. Overall, except for GPT-4o, the performance of the remaining LMMs is still below 65.0 in the Customized Mimic task. This indicates that, in addition to the capability to recognize data in the chart, understanding the layout of the chart and code generation are also key factors for LMMs to better accomplish the task.

## 4 DISCUSSION

### 4.1 DIFFERENT COMPLEXITY LEVELS

We report the performance of the top-6 models across different complexity levels for ChartMimic in Fig. 3. There is a consistent decline in performance across all tasks as the difficulty increases. For example, in Direct Mimic task, the performance of GPT-4o at easy, medium, and hard levels are 86.5, 77.7, and 74.8, respectively. These results demonstrate that ChartMimic are inherently challenging and confirm the efficacy of the established difficulty levels. Additionally, for weak LMMs such as MiniCPM-Llama3-V-2.5, providing customized data at the Easy Level can help it perform tasks better (from 46.8 to 59.7). However, at the Hard Level, even when given customized data, their performance

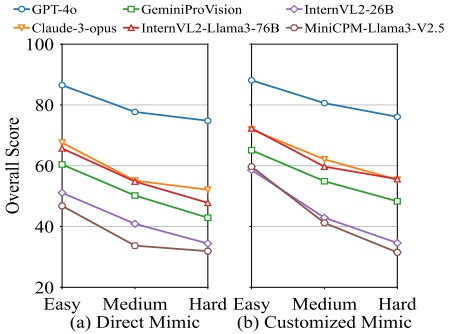

Figure 3: Overall scores of models at different complexity levels.

Table 5: Results for different prompting methods for GPT-4o and different size of InternVL2 LMMs on **Direct Mimic** task. The best scores for each model are in **bold**.

| Model | Params | Method | Exec. Rate | Low-Level | | | | | High-Level | Overall |
|---|---|---|---|---|---|---|---|---|---|---|
| | | | | Text | Layout | Type | Color | Avg. | GPT-4o | |
| GPT-4o | - | Direct | 93.2 | 81.5 | 89.8 | 77.3 | 67.2 | 79.0 | 83.5 | 81.2 |
| | | HintEnhanced | 92.0 | 80.9 | 89.7 | 78.2 | 67.9 | 79.2 | 83.0 | 81.1 |
| | | SelfReflection | 95.2 | 86.6 | 93.7 | 81.1 | 68.2 | 82.4 | 87.1 | **84.8** |
| | | Scaffold | 91.0 | 81.3 | 90.6 | 75.1 | 63.9 | 77.7 | 80.3 | 79.0 |
| InternVL2-Llama3-76B | 76.0B | Direct | 83.2 | 54.1 | 74.5 | 49.2 | 41.5 | 54.8 | 62.2 | 58.5 |
| | | HintEnhanced | 78.0 | 54.3 | 71.7 | 50.2 | 41.6 | 54.5 | 60.3 | 57.4 |
| | | SelfReflection | 82.5 | 57.0 | 76.2 | 56.4 | 45.0 | 58.7 | 63.6 | **61.1** |
| | | Scaffold | 76.7 | 52.1 | 71.4 | 49.1 | 40.5 | 53.3 | 60.1 | 56.7 |
| InternVL2-26B | 26.0B | Direct | 69.3 | 39.2 | 58.7 | 35.9 | 31.8 | 41.4 | 47.4 | **44.4** |
| | | HintEnhanced | 66.8 | 40.4 | 54.8 | 36.6 | 33.0 | 41.2 | 47.0 | 44.1 |
| | | Self | 69.0 | 39.5 | 57.5 | 35.6 | 30.5 | 40.8 | 47.1 | 43.9 |
| | | Scaffold | 64.3 | 36.0 | 53.5 | 35.4 | 30.3 | 38.8 | 44.0 | 41.4 |
| InternVL2-8B | 8.1B | Direct | 61.8 | 31.5 | 51.1 | 28.6 | 26.2 | 34.4 | 38.9 | **36.6** |
| | | HintEnhanced | 58.7 | 30.1 | 38.3 | 30.1 | 27.5 | 31.5 | 35.9 | 33.7 |
| | | SelfReflection | 56.8 | 27.4 | 39.8 | 25.1 | 22.9 | 28.8 | 33.4 | 31.1 |
| | | Scaffold | 61.8 | 22.8 | 43.5 | 24.8 | 20.2 | 27.8 | 32.0 | 29.9 |
| InternVL2-2B | 2.1B | Direct | 52.5 | 23.6 | 35.8 | 16.0 | 15.4 | 22.7 | 24.2 | **23.5** |
| | | HintEnhanced | 41.5 | 17.3 | 19.4 | 14.3 | 12.9 | 16.0 | 18.5 | 17.2 |
| | | SelfReflection | 35.5 | 16.3 | 22.2 | 12.8 | 14.8 | 16.5 | 17.3 | 16.9 |
| | | Scaffold | 38.0 | 6.5 | 17.8 | 3.2 | 2.2 | 7.4 | 15.6 | 11.5 |

do not show improvement (from 31.9 to 31.5). This indicates that for Easy Level, complex reasoning abilities are not required, and providing data is sufficient to complete the task well. However, for Hard Level charts, LMMs need complex code generation capabilities grounded on visual understanding. In this case, even when given data, if the LMMs cannot perform the corresponding code reasoning, they still cannot complete the task. Moreover, when given addtional data, if the data is too long and complex, the LLM may still become confused, leading to a certain decrease in performance.

## 4.2 DIFFERENT PROMPTING METHODS

We further examine the impact of different prompting methods on the performance of ChartMimic benchmark. Specifically, we choose the GPT-4o and InternVL2 series LMMs (2B, 8B, 26B, 76B) to study their performance on the Direct Mimic task. We select three representative prompting methods: HintEnhanced, SelfReflection, and Scaffold Prompting. HintEnhanced uses prompt with chain-of-thought (Wei et al., 2022), explicitly prompting the LMMs to pay attention to important details (e.g., layout, type, text, etc). SelfReflection (Shinn et al., 2024) involves inputting the LMMs' own output and the corresponding rendered chart as additional information, instructing the LMM' to self-reflect their output. Scaffold Prompting (Lei et al., 2024) overlays a dot matrix within the figure as visual information anchors and leverages multi-dimensional coordinates as textual positional references. We detail the experimental setups in Appendix F.

As shown in Tab. 5, SelfReflection enables GPT-4o and InternVL2-Llama3-76B to reflect on and correct their outputs, demonstrating notable improvements over Direct Prompting. This underscore self-reflection as a key manifestation of System 2 reasoning (Sloman, 1996; Kumar et al., 2024). For GPT-4o, SelfReflection enhances Text, Layout, and Type Scores, though Color Score remains unchanged due to persistent challenges in fine color discrimination. However, LMMs with less developed reasoning capabilities (InternVL2-26B, 8B, and 2B) show no improvement or even decline with self-reflection. This indicates that only LMMs with substantial reasoning capabilities can effectively engage in self-reflection and result optimization, underscoring the critical role of System 2 reasoning in addressing ChartMimic's challenges. Regarding HintEnhanced and Scaffold methods, they do not enhance the performance of models and reduce performance to varying degrees. Upon examining the cases, we find that the HintEnhanced method, which involves generating captions first, can introduce hallucinations that lead to errors in the subsequently generated code. As for the Scaffold method, introduction of an additional dot matrix can interfere with existing coordinate axis information in charts with high information density, thereby negatively impacting performance. These negative effects intensify as LMMs' reasoning capabilities decrease, further highlighting the importance of advanced reasoning capabilities in handling complex prompting strategies. We provide case studies of three prompting methods in Appendix F.

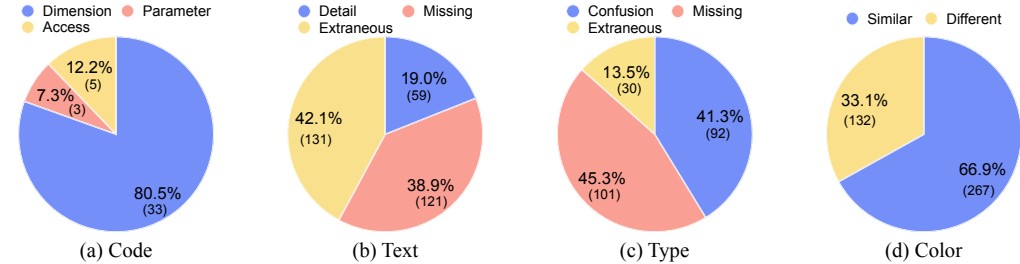

Figure 4: Error analysis of GPT-4o across four error types on the Direct Mimic task. The number in brackets indicates the count of error cases. Error examples can be found in Appendix Appendix I.

### 4.3 CORRELATION WITH HUMAN EVALUATION

To evaluate the reliability of the proposed multi-level metrics, we calculate their correlation with human evaluations. Specifically, we collect $1,200$ charts collected from GPT-4o using four different prompting methods (Sec. 4.2) on Direct Mimic. Each charts is evaluated by three individual evaluators, who assign scores ranging from $0$ to $100$ based on the similarity with the ground-truth chart. Details about the human evaluation process are provided in Ap-

Table 6: Pearson correlation coefficient between multi-level evaluation metric and human evaluation.

| Metric | Coefficient | p-value |
|---|---|---|
| High-Level | 0.7041 | < 0.0001 |
| Low-Level | 0.7681 | < 0.0001 |

pendix G. As shown in Tab. 6, the Pearson correlation coefficient ($r$) indicates that both the high-level ($r = 0.7041$) and low-level metric ($r = 0.7681$) have a high correlation with human judgment, demonstrating the reliability of our multi-level metrics. Moreover, the low-level metric demonstrates a higher correlation with human judgments compared to the high-level metric. This is attributed to its dependence on code execution logic, enabling it to capture detailed elements.

### 4.4 ERROR ANALYSIS

We conduct a detailed error analysis of the state-of-the-art GPT-4o model to elucidate the current limitations of LMMs on ChartMimic, thereby identifying potential areas for further improvement. Our analysis identifies and category four main types of errors, detailed below.

**Code-related Errors.** (1) Dimension: Errors associated with data dimension, e.g., data dimension does not satisfy required conditions for operations; (2) Access: Errors occurring when accessing iterable elements out of bounds or accessing undeclared variables; (3) Parameter: Errors related to passing incorrect parameters when invoking functions. As observed in Fig. 4 (a), the majority of errors stem from Dimension, suggesting analysis and operation on data poses a challenge for GPT-4o.

**Text-related Errors.** (1) Detail: The text in the generated chart is largely consistent with the ground-truth chart text but contains minor discrepancies, such as mixing up "-" and "_"; (2) Missing: The generated chart omits text present in the ground-truth chart; (3) Extraneous: The generated chart contains text not found in the ground-truth chart, e.g., adding text for titles, reflecting the model's own interpretation of the chart; As depicted in Fig. 4 (b), the majority of errors are Missing and Extraneous, which indicates that for GPT-4o, due to intensive information in charts, it is challenging for it to comprehend the whole scope of the charts, even the basic text recognition task.

**Type-related Errors.** (1) Confusion: GPT-4o misinterprets the chart type as one that appears or functions similarly, such as mistaking violin plots for box plots, and so on; (2) Missing: The generated chart omits chart types present in the ground-truth chart; (3) Extraneous: The generated chart includes chart types not present in the ground-truth chart. As demonstrated in Fig. 4 (c), although GPT-4o exhibits remarkable capability in object recognition for natural images, it still struggles with scientific charts, which contain more nuanced semantic meanings through visual logic.

**Color-related Errors.** (1) Similar: The colors are not the same as the ground-truth colors but appear analogous. (2) Different: The colors are entirely dissimilar to the ground-truth colors. As shown in Fig. 4 (d), though GPT-4o can not exactly recognize the accurate colors, it can identify similar ones.

To sum up, GPT-4o still faces challenges in code generation and exhibits notable visual understanding deficiencies on ChartMimic. It has difficulty accurately recognizing visual elements in figures or may hallucinate incorrect elements and struggle with complex data analysis. Combining these insights, there is a need for further improvement in both visual understanding and code generation.

## 5    RELATED WORK

**Large Multimodal Models.**   The proprietary LMMs such as GPT-4o (OpenAI, 2024), Gemini (Google, 2023), and Claude-3 (Anthropic, 2024) have enabled complex multimodal interactions. Similarly, emerging open-weight LMMs such as LLaVA (Xu et al., 2024; Li et al., 2024a), InternVL (Chen et al., 2023), Qwen-VL (Bai et al., 2023), DeepSeek-VL (Lu et al., 2024) have contributed to the community. Despite these advancements, the effective evaluation of LMMs remains a major challenge. However, effectively evaluating LMMs remains a challenge, with open-source models performing well in benchmarks (Liu et al., 2023; Chen et al., 2024; Yue et al., 2023; Lu et al., 2023; Luo et al., 2025) yet falling short in practical applications (Xie et al., 2024; Koh et al., 2024; Si et al., 2024; Ji et al., 2024). This gap emphasizes the need for real-world-based evaluations that reflect authentic use cases (Ma et al., 2024a;b). ChartMimic addresses this by requiring LMMs to translate complex visual information into code, testing their visual and coding capabilities.

**Code Generation.**   Tasks such as HumanEval (Chen et al., 2021), MBPP (Austin et al., 2021), APPS (Hendrycks et al., 2021), and DS-1000 (Lai et al., 2023) are important benchmarks in natural language processing, but the inputs for these tasks are single-modal, consisting only of text, limiting their scope. With the emergence of LMMs, the evaluation of multimodal code generation has become increasingly critical for assessing real-world capabilities (Li et al., 2024b; Si et al., 2024; Wu et al., 2024a; Ge et al., 2025). For example, MMCode (Li et al., 2024b) attempts to address this issue, but their visual inputs may be simplistic and face the problem of a single evaluation method. Design2Code (Si et al., 2024) evaluates LMMs' code generation abilities through HTML web page generation. However, their test data comes from the C4 (Raffel et al., 2020) dataset, which may pose a risk of data leakage. Meanwhile, they just imitate the HTML and do not take the customized instructions; we take this into consideration. Recently, Plot2Code (Wu et al., 2024a) undertakes work similar to ours, aiming to measure models' code generation abilities through chart-to-code generation. Similarly, their approach of directly scraping data from the matplotlib gallery poses a risk of data leakage. Our ChartMimic provides a new set of manually curated $4,800$ data pairs, ensuring chart diversity and offering more fine-grained evaluation methods.

**Chart Understanding.** ChartMimic evaluates the capabilities of LMMs in grounding chart understanding into code generation, bridging visual and programmatic domains. Previous works focus on chart question answering (Masry et al., 2022; Methani et al., 2020; Xu et al., 2023; Wang et al., 2024b; Li et al., 2024c; Zeng et al., 2024) and chart captioning (Rahman et al., 2023; Kantharaj et al., 2022). They assess the LMMs' ability to understand specific data characteristics or summarize key information into text. ChartMimic advances the field by introducing a chart-to-code task, transforming the LMMs' understanding of charts into code, which is neglected before but a realistic scenario for practical, real-world usage. This approach enables a comprehensive evaluation of the LMMs' overall comprehension of charts and their ability to express this understanding in code form. Leveraging the linguistic properties of code, our benchmark introduces fine-grained metrics to assess LMMs' chart understanding capabilities across multiple dimensions, including text, chart type, layout and color.

## 6    CONCLUSIONS

In this study, we develop the ChartMimic benchmark to evaluate LMMs' proficiency capability via chart-to-code generation. ChartMimic focuses on real-world applications for data visualization, aiming to assess LMMs' ability to harmonize a blend of cognitive capabilities, including visual understanding, code generation, and cross-modal reasoning. We propose two distinct levels of evaluation metrics (low and high level) to provide a comprehensive assessment. ChartMimic directly contributes to the understanding of progress towards artificial general intelligence, reflecting the expertise and reasoning abilities expected of skilled adults in various professional fields. Despite its comprehensive nature, ChartMimic, like any benchmark, has limitations. The manual curation process, although thorough, may introduce biases. Additionally, using scientific charts as information-intensive visual inputs to measure LMMs' multimodal code generation capabilities, while effective, still encounters domain-specific challenges. Our evaluation metric, despite considering most elements' similarity, does not uniformly score details of sub-icons, such as markers. We anticipate that ChartMimic will inspire the development of LMMs, advancing the pursuit of artificial general intelligence. Future research could explore various aspects, such as multimodal reasoning prompt strategies, to further reduce the gap between open-weight LMMs and proprietary ones.

## ACKNOWLEDGMENTS

This work was partly supported by the National Key Research and Development Program of China (No. 2024YFB2808903) and the Shenzhen Science and Technology Program JSGG20220831110203007).

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

APPENDIX

## A    AUTHOR CONTRIBUTIONS

Yujiu Yang served as the principal advisor for this project. Cheng Yang and Chufan Shi acted as the student project leads.

**Data Annotation.** Yaxin Liu and Bo Shui co-led the data annotation process (Appendix B). Chufan Shi, Cheng Yang, Mohan Jing and Linran Xu helped to contribute to various aspects of this process. Xiaomei Nie provided expert guidance on data visualization and conducted quality control.

**Code Implementation.** Cheng Yang and Chufan Shi co-implemented the codebase for ChartMimic.

**Experiments.** Cheng Yang and Chufan Shi jointly conducted the evaluation of LMMs on ChartMimic. Yaxin Liu and Bo Shui jointly led the human evaluation.

**Paper Writing.** Cheng Yang and Chufan Shi finished the main manuscript. Junjie Wang and Yujiu Yang provided in-depth reviews and revisions. Yaxin Liu and Bo Shui created all visualizations presented in the paper and contributed to the Data Annotation (Appendix B) and Chart Taxonomy (Appendix D). Deng Cai, Xinyu Zhu, Siheng Li, Yuxiang Zhang and Gongye Liu contributed through proofreading, discussions and feedback.

## B    DATA ANNOTATION

### B.1    DATA ANNOTATION PRINCIPLES

**Diversity of Chart Types.** Data visualization has become an essential tool for conveying information in various fields, and the design practices and requirements for different types of charts vary significantly(Parsons, 2021). Most of previous work (Han et al., 2023; Xu et al., 2023; Masry et al., 2022; Kantharaj et al., 2022; Rahman et al., 2023) focus only on line, bar, pie charts, etc, which are commonly used within the field of computer science. However, with the increasing integration of LMMs into everyday tasks, people from all fields are starting to use generative techniques as daily assistants to enhance their design and creative processes when creating visualizations. In this light, enriching the spectrum of chart diversity is crucial for evaluating LMMs' proficiency in multimodal chart-to-code generation.

**Balance of Charts Complexity.** Charts serve as visual aids for data presentation, enabling users to promptly grasp the underlying patterns and significance within the data. It is essential to adopt the appropriate chart type and complexity level to effectively convey the information (Evergreen, 2019). Previous works have focused mainly on charts with only a single data format and low information density, which are rarely encountered in practical settings such as academic writing. Noting that LMMs like GPT-4o have already demonstrated outstanding data visualization capabilities, our focus is on using charts that are actually employed in practice, such as those from research papers, and on selecting charts with varying levels of complexity when constructing our benchmark.

**Reduction of Data Leakage.** Recognizing that augmenting training data is a primary method for enhancing the performance of LMMs has led researchers to more comprehensively exploit all accessible data during pre-training. However, this approach introduces the possibility of data leakage, especially when pre-training data might already include resources such as the matplotlib gallery[2] or other pre-existing datasets, potentially resulting in inaccurate evaluations. To mitigate this issue, we deliberately avoid the use of code that can be readily found online or code that could be auto-generated by large language models for chart creation in constructing our dataset, thereby reducing the probability of data leakage. To further illustrate the potential data leakage in matplotlib gallery, we select the code for 20 charts from the matplotlib gallery. Specifically, we provide the first half of the code as a prefix for Llama3-8B (AI@Meta, 2024) and let the model complete the remaining code. Then we calculate the edit distance between the generated complete code block and the ground truth. Similarly, we apply the same process to the code in Direct Mimic, calculating the edit distance between the generated results and the ground truth. We find that the edit distance for the code in the matplotlib gallery is 22.1, while the edit distance for Direct Mimic's code is 39.8. This indicates that the code in Direct Mimic have a larger edit distance, which further reduces the risk of data leakage compared to the matplotlib gallery ones.

---

[2]`https://matplotlib.org/`

**Integration of Authentic User Requirements.** Users from diverse domains such as finance, health-care, education, and engineering demonstrate unique needs and preferences for data visualization. These requirements extend beyond mere diversity; they demand the incorporation of charts capable of articulating complex and multi-dimensional data, and conforming to domain-specific aesthetic preferences(Qin et al., 2020; Evergreen, 2019). By aligning our dataset construction process with these real-world demands, we enable a more relevant and precise evaluation of LMMs. Adopting this approach not only reflects actual user patterns but also steers the research community in the iterative improvement of LMMs. This focused development will also lead to an enhanced user experience and increased user satisfaction, as the models become more proficient in meeting the sophisticated and varied needs of users.

## B.2 DATA ANNOTATION PIPELINE

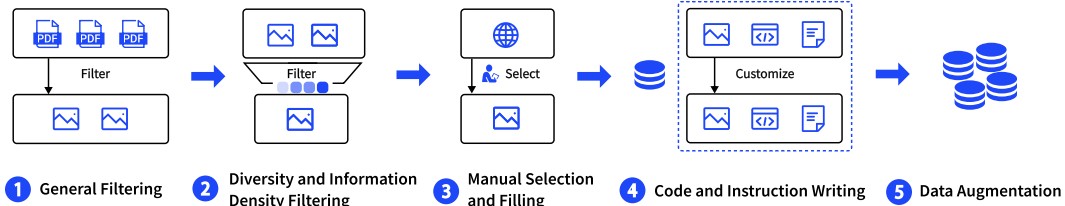

Figure 5: An illustration of the data annotation pipeline, which encompasses five key steps.

We present the detailed description about the five-step data annotation pipeline in this section. Fig. 5 demonstrates an illustration of the data annotation pipeline.

**General Filtering.** To obtain a high-quality dataset aligned with real-world use cases and to avoid data leakage, we initially scrape figures from source files of publications on arXiv[3] that hold a CC BY 4.0 license and have a publication date after February 2024. We then extract figures in PDF format, yielding approximately 174,100 figures across various domains (such as Physics, Mathematics, Computer Science, Quantitative Biology, Quantitative Finance, Statistics, Electrical Engineering and Systems Science, Economics). We filter these figures based on file format and generation method, retaining only Matplotlib-generated PDFs, indicating that these figures can be reproduced using Python. This process results in a refined collection of 15,800 figures.

**Diversity and Information Density Filtering.** This stage involves a two-phase process conducted by five domain experts from information visualization, digital media, industrial design, visual communication design, and computer science. The information visualization expert has three years of research experience in scientific visualization and visual analytics. The digital media specialist focuses on human-computer interaction and multimedia design. The industrial design expert brings perspectives from user experience and product visualization. The visual communication design expert specializes in graphic design principles and information aesthetics. The computer science expert has extensive experience in data visualization programming and scientific computing.

In the first phase, these experts conduct a 7-day manual review of the 15,800 figures, focusing on visual diversity and information communication effectiveness. They reference the Matplotlib gallery, gradually identifying and finalizing chart type while reviewing the figures, and build corresponding type pools. For each new figure, they assess its visual elements—such as layout, axes, line styles, marker styles, and colors—against existing figures in their corresponding type pool. If the figure as long as exhibits a distinctive difference in at least one of these aspects, it is retained; otherwise, it is excluded. This process results in 1,295 figures being selected for the second phase.

In the second phase, these experts independently review the 1,295 figures and further select those figures they deem to exhibit significant distinctions and diversity. Figures selected unanimously by all experts are directly included, while the remaining figures are subjected to a majority voting system requiring at least 3/5 votes for inclusion. This rigorous process, which takes less than 3 days to complete, results in a final set of 279 figures.

---

[3]https://arxiv.org/

**Manual Selection and Filling.** In addition to sourcing from arXiv, we curate chart figures from diverse platforms such as the matplotlib gallery, Stack Overflow, and plotting-related forums on Twitter and Reddit. These charts are deliberately chosen for their distinctive styles, which are not present in our arXiv dataset. Consequently, we obtain 600 prototype charts for ChartMimic. This stage took us less than a week to complete the data selection.

**Code and Instruction Writing.** We propose to manually write codes and instructions for ChartMimic based on the collected 600 prototype charts. To ensure annotation quality, a team of skilled Python users—Python annotators—master's students in computer science with 6+ years of Python and matplotlib experience—reproduce 600 prototype charts using Python 3.9.0 and matplotlib v3.8.4. Since the unannotated data in the figures cannot be fully restored, they can only be approximated when writing the code. This process generates 600 (figure, code, instruction) triplets for the Direct Mimic task and another 600 triplets for the Customized Mimic task by integrating data from various domains into the corresponding code and instructions, comprising 1,200 high-quality seed data.

**Data Augmentation.** After developing the seed triplets, we proceed with a manual data augmentation process. Python Annotators modify various elements of each seed triplet, including data, color schemes, and mark styles, to create augmented triplets. For each seed triplet, we generate three additional augmented triplets. This process enhances our dataset, resulting in a total of $4,800$ triplets. The "Code and Instruction Writing" and "Data Augmentation" stages together take the data annotators approximately 1.5 months to complete.

## B.3 COMPLEXITY LEVELS

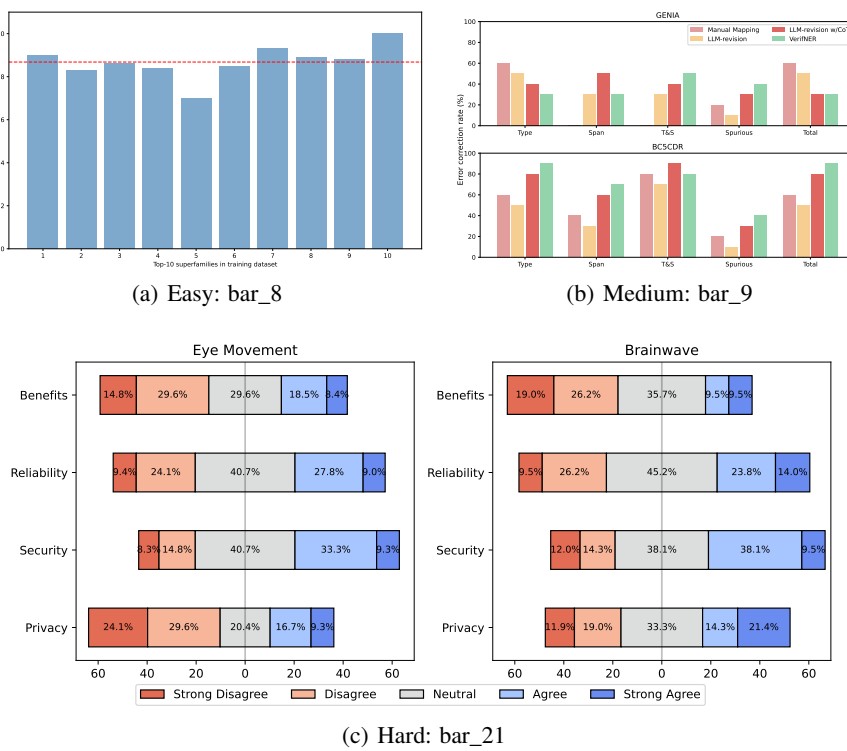

(a) Easy: bar_8     (b) Medium: bar_9

(c) Hard: bar_21

Figure 6: Representative examples of chart complexity in ChartMimic (Easy, Medium, Hard).

Our complexity assessment framework is established upon two fundamental criteria, designed to comprehensively evaluate both the visual and technical aspects of chart generation:

**Visual Elements Complexity** We systematically evaluate the sophistication of each chart through a comprehensive scoring mechanism applied to its constituent visual components. This encompasses the chart typology, data grouping structures, marker configurations, textual elements, chromatic schemes, compositional layouts, and coordinate systems. Each visual component is quantitatively assessed

based on both its frequency and sophistication level (designated as 1-3 points for low/medium/high complexity, respectively). This multi-dimensional scoring approach ensures a thorough evaluation of the visual complexity inherent in each chart.

**Implementation Complexity** We incorporate code complexity as a quantitative metric, measured primarily through code length and structural intricacy. This parameter effectively captures the technical sophistication required for accurate chart reproduction, including the complexity of data preprocessing, visualization logic, and stylistic customizations. The implementation complexity provides insights into the programming challenges associated with each chart type.

Consequently, charts in ChartMimic are systematically categorized into three distinct complexity levels, each representing a specific combination of visual and implementation challenges:

**Easy:** Fundamental chart configurations featuring minimal visual complexity and straightforward implementation requirements (e.g., in Fig. 6(a), monochromatic bar charts with sparse textual elements and simplified data representation).

**Medium:** Charts exhibiting intermediate complexity in visual element composition or implementation requirements (e.g., in Fig. 6(b), dual-subplot bar charts incorporating grouped data structures, diverse chromatic schemes, and moderate textual annotations).

**Hard:** Charts demonstrating sophisticated visual elements or advanced implementation (e.g., in Fig. 6(c), complex dual-subplot bar charts featuring divergent data patterns, extensive color schemes, comprehensive textual annotations, and substantial code complexity).

## B.4 INSTRUCTION EXAMPLES

To illustrate our instruction format and task requirements, we present examples (bar_28 and CB_29) with their corresponding (figure, instruction, code) triplets for both Direct Mimic and Customized Mimic tasks. Fig. 7 and Fig. 8 show the Direct Mimic and Customized Mimic tasks for a bar chart (bar_28), respectively. The instruction provides guidance on creating the visualization. Similarly, Fig. 9 and Fig. 10 demonstrate the tasks for a more complex combination chart (CB_29).

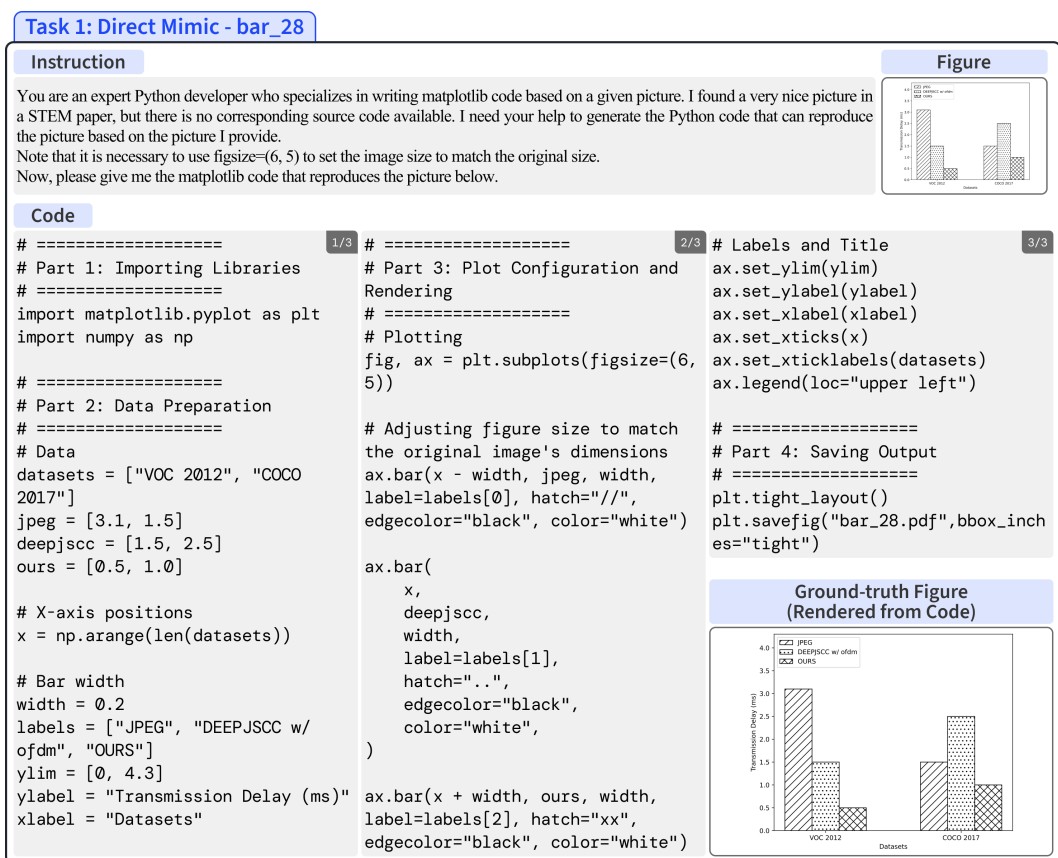

Figure 7: An Example of the Direct Mimic task (bar_28), showing the (figure, instruction, code) triplet. Additionally, we also display the ground truth figure rendered from the code for illustration.

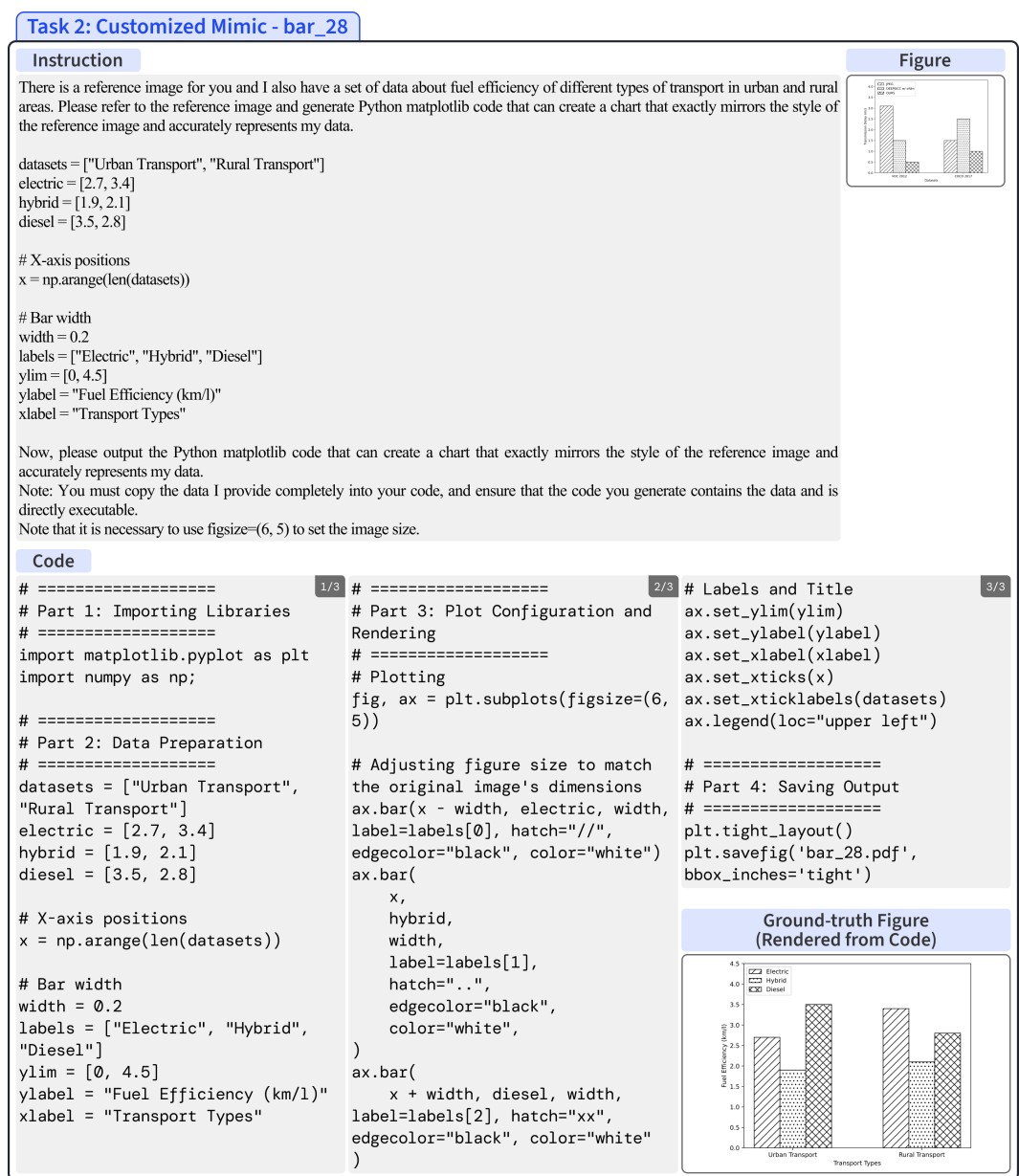

Figure 8: An Example of the Customized Mimic task (bar_28), showing the (figure, instruction, code) triplet. Additionally, we also display the ground truth figure rendered from the code for illustration.

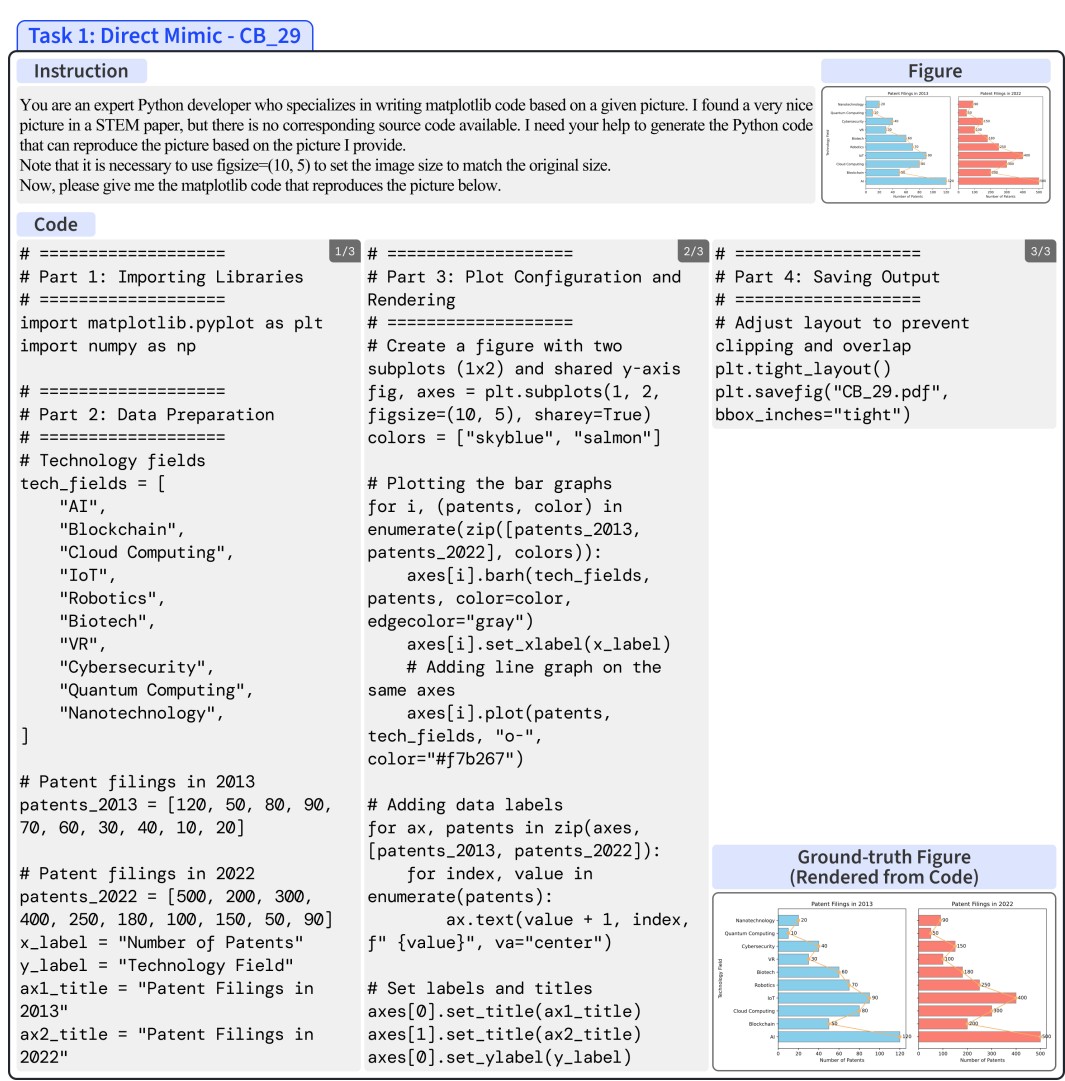

Figure 9: An Example of the Direct Mimic task (CB_29), showing the (figure, instruction, code) triplet. Additionally, we also display the ground truth figure rendered from the code for illustration.

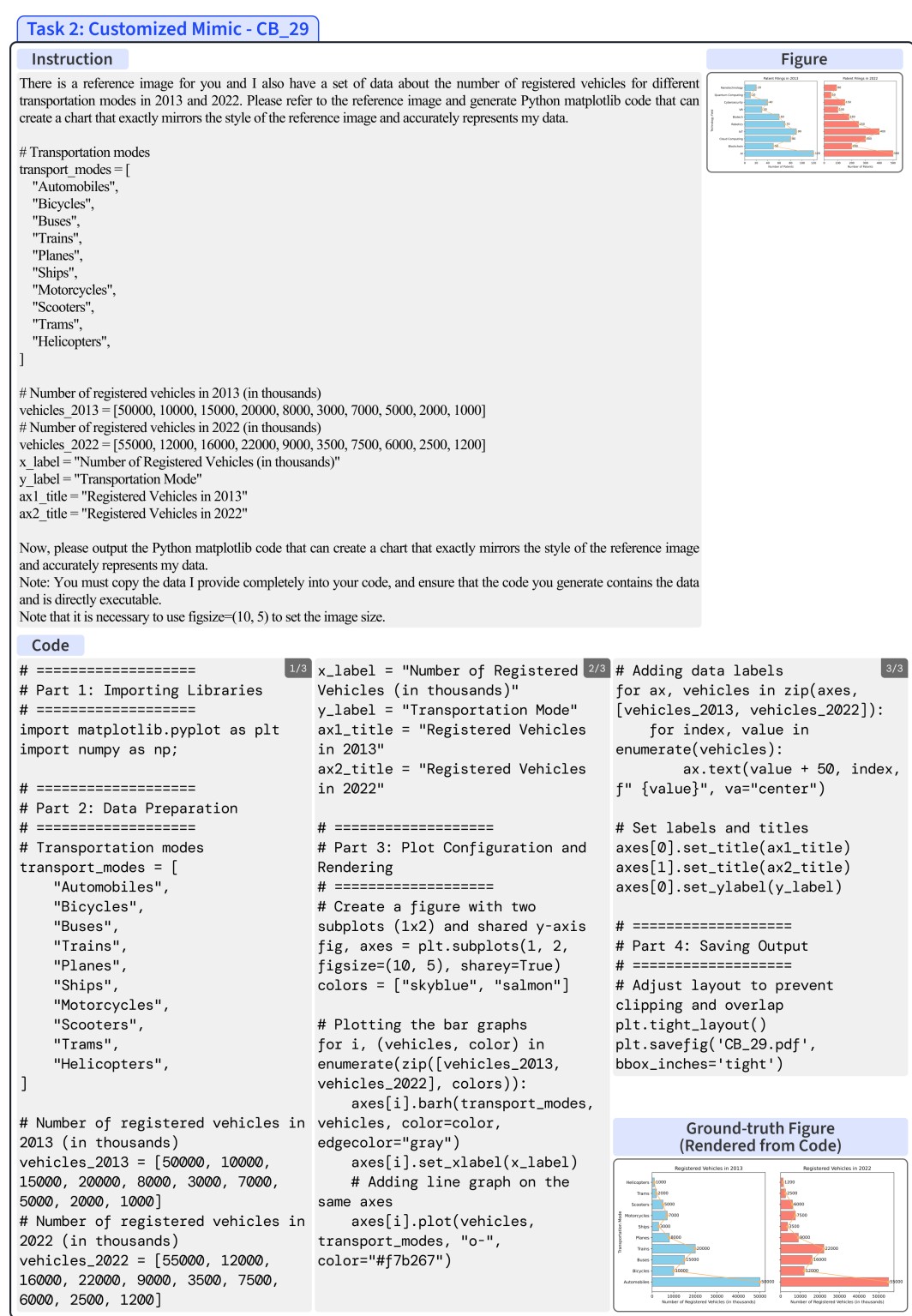

Figure 10: An Example of the Customized Mimic task (CB_29), showing the (figure, instruction, code) triplet. Additionally, we also display the ground truth figure rendered from the code for illustration.

Table 7: The ChartMimic benchmark with **Direct Mimic** task. We report the results of both Testmini and Test set.

| Model | Params | Test set | Exec. Rate | Low-Level | | | | | High-Level | Overall |
|---|---|---|---|---|---|---|---|---|---|---|
| | | | | Text | Layout | Type | Color | Avg. | GPT-4o | |
| *Proprietary* | | | | | | | | | | |
| GPT-4o | - | Testmini | 93.2 | 81.5 | 89.8 | 77.3 | 67.2 | 79.0 | 83.5 | 81.2 |
| | | Test | 93.0 | 83.4 | 90.2 | 76.7 | 66.0 | 79.1 | 84.1 | 81.6 |
| *Open-Weight* | | | | | | | | | | |
| InternVL2-2B | 2.2B | Testmini | 52.5 | 23.6 | 35.8 | 16.0 | 15.4 | 22.7 | 24.2 | 23.5 |
| | | Test | 51.8 | 23.3 | 34.8 | 15.9 | 15.9 | 22.5 | 25.6 | 24.0 |
| InternVL2-8B | 8.1B | Testmini | 61.8 | 31.5 | 51.1 | 28.6 | 26.2 | 34.4 | 38.9 | 36.6 |
| | | Test | 61.6 | 33.4 | 47.4 | 25.7 | 24.2 | 32.7 | 39.9 | 36.3 |
| InternVL2-26B | 26.0B | Testmini | 69.3 | 39.2 | 58.7 | 35.9 | 31.8 | 41.4 | 47.4 | 44.4 |
| | | Test | 69.9 | 41.7 | 58.0 | 35.6 | 31.0 | 41.6 | 48.1 | 44.9 |
| InternVL2-Llama3-76B | 76.0B | Testmini | 83.2 | 54.1 | 74.5 | 49.2 | 41.5 | 54.8 | 62.2 | 58.5 |
| | | Test | 83.3 | 55.6 | 73.5 | 50.4 | 40.8 | 55.1 | 62.7 | 58.9 |

Table 8: The ChartMimic benchmark with **Customized Mimic** task. We report the results of both Testmini and Test set.

| Model | Params | Test set | Exec. Rate | Low-Level | | | | | High-Level | Overall |
|---|---|---|---|---|---|---|---|---|---|---|
| | | | | Text | Layout | Type | Color | Avg. | GPT-4o | |
| *Proprietary* | | | | | | | | | | |
| GPT-4o | - | Testmini | 96.5 | 88.5 | 92.9 | 79.2 | 67.6 | 82.1 | 84.3 | 83.2 |
| | | Test | 96.2 | 87.2 | 91.7 | 80.1 | 66.4 | 81.4 | 84.8 | 83.1 |
| *Open-Weight* | | | | | | | | | | |
| InternVL2-2B | 2.2B | Testmini | 49.3 | 22.2 | 35.4 | 20.0 | 18.1 | 23.9 | 27.8 | 25.9 |
| | | Test | 49.6 | 22.4 | 33.9 | 19.2 | 19.6 | 23.8 | 28.4 | 26.1 |
| InternVL2-8B | 8.1B | Testmini | 73.0 | 43.1 | 54.4 | 39.9 | 35.4 | 43.2 | 48.9 | 46.1 |
| | | Test | 73.5 | 43.7 | 54.1 | 41.1 | 34.1 | 43.3 | 49.8 | 46.5 |
| InternVL2-26B | 26.0B | Testmini | 73.7 | 43.9 | 62.3 | 43.5 | 34.3 | 46.0 | 51.1 | 48.6 |
| | | Test | 74.7 | 44.7 | 66.3 | 46.8 | 35.1 | 48.2 | 50.8 | 49.5 |
| InternVL2-Llama3-76B | 76.0B | Testmini | 89.8 | 57.8 | 79.0 | 63.5 | 50.5 | 62.7 | 66.7 | 64.7 |
| | | Test | 88.1 | 57.9 | 79.6 | 65.6 | 51.7 | 63.7 | 68.2 | 66.0 |

## C    CORRELATION BETWEEN TEST SET AND TESTMINI SET

Tab. 7 and 8 reports the performance of GPT-4o and InternVL2 series LMMs (2B, 8B, 26B, Llama3-76B) on Direct Mimic and Customized Mimic task. The minor differences between scores on the test subset and the testmini subset suggest that testmini effectively mirrors the test subset, serving as a valuable evaluation subset for model development, especially for those who have limited computing resources.

## D    CHART TAXONOMY

This section presents the chart taxonomy in ChartMimic. It encompasses a structure of 22 categories according to chart type characteristics and data composition. The categories comprise of:

- 18 regular types, ordered as follows: Bar, Heatmap, Scatter, Box, Errorbar, Errorpoint, Line, Violin, Radar, Pie, Density, Graph, Quiver, Contour, Histogram, Tree, Area, and 3D charts.

- 4 advanced types: PIP (Plot-in-Plot), Multidiff (Multiple Differences), Combination, and HR (Hard-to-Recognize).

The regular types are further divided into subcategories according to chart feature or data characteristics, whereas each chart of the advanced types represents a unique subcategory. The taxonomy showcases examples to showcase diversity of each category.

**Bar**: Bar chart uses rectangular bars to represent data and can be distinguished by its orientation, horizontal or vertical, with its nuanced data attributes. There are 8 subcategories of the data attributes for each orientation, consisting of 16 subcategories in total, as shown in Figure 11:

1. Base (single positive data set, unordered)

2. Sorted (data in ascending or descending sequence)

3. Grouped (multiple positive data sets, adjacent)

4. Stacked (multiple positive data sets laying atop one another)

5. Normalized (proportioned stacks of positive data summing to one, a Stacked variant)

6. Diverging (multiple stacked data sets expanding from a central axis)

7. With-Negative (data sets including negative values)

8. Reverse (exclusively negative data sets)

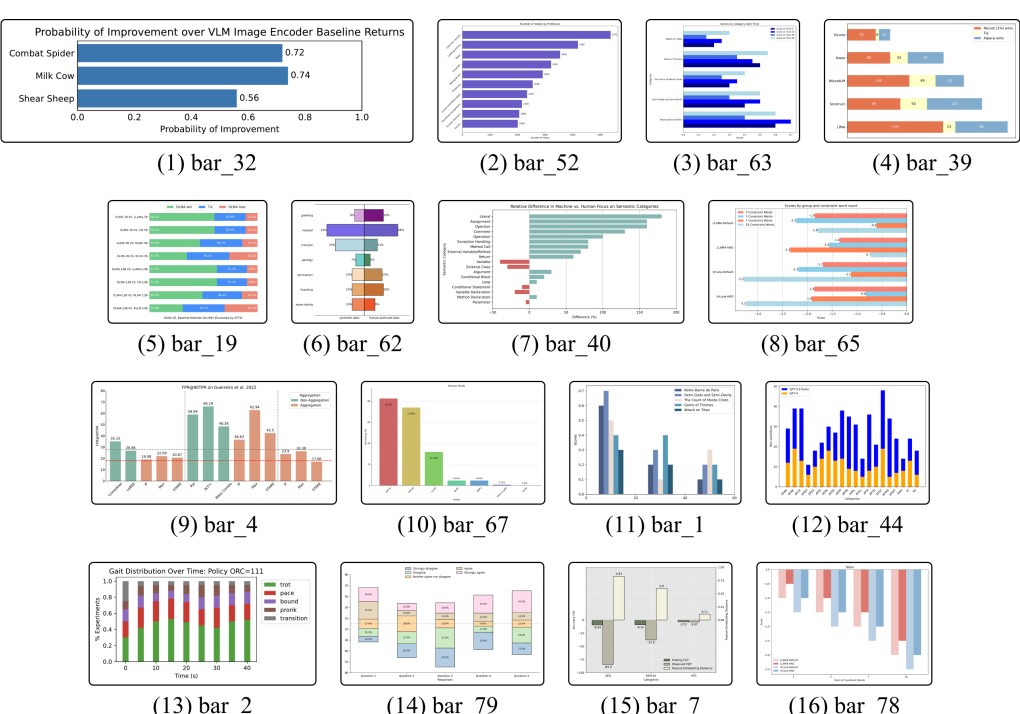

Figure 11: Examples of Bar chart subcategories.

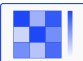 **Heatmap**: Based on the layout and visual representations, Heatmap is categorized into 4 subcategories, each reflecting unique aspects of data presentation, as shown in Figure 12.

1. Base (the general layout of a typical heatmap)
2. Missing-Data (visualization indicating the absence of data points)
3. Triangle-Layout (heatmap configured in a triangular layout)
4. Other-Shaped (heatmap comprising elements in non-rectangular forms, such as circles)

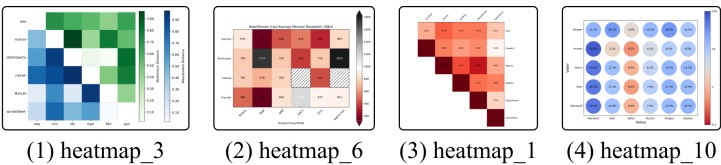

    (1) heatmap_3      (2) heatmap_6    (3) heatmap_1    (4) heatmap_10

Figure 12: Examples of Heatmap subcategories.

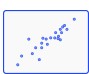 **Scatter**: Scatter plot is classified based on the dot characteristics and data distribution. The taxonomy is divided into 4 subcategories as shown in Figure 13.

1. Base (basic scatter plot, uniform dot size, color may vary)
2. Diff-Shape (different dot shapes)
3. Diff-Size (different dot sizes, such as a bubble chart)
4. Clustered (scatter plot with clear clustering)

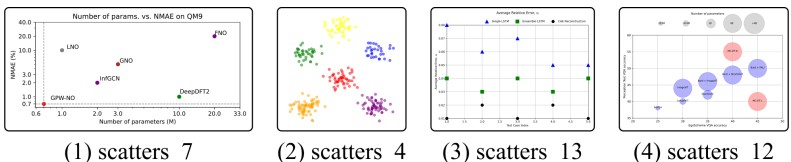

    (1) scatters_7      (2) scatters_4    (3) scatters_13    (4) scatters_12

Figure 13: Examples of Scatter plot subcategories.

**Box**: Box plot is characterized by the orientation of boxes and data characteristics. The orientation can be horizontal or vertical, while data characteristics include the grouping of data and the presence of missing lines. The taxonomy is divided into 3 subcategories for each orientation, consisting of 6 subcategories in total, as shown in Figure 14.

1. Base (single group of data, complete box shape)
2. Grouped (multiple groups of data, complete box shape)
3. Missed-Line (missing parts below the first quartile line and above the third quartile line)

**Error Bar**: An Error bar chart is an enhanced variant of the basic bar chart, augmented with error margins to represent the variability or uncertainty within the data. Unlike Bar chart, Error Bar chart typically does not include the categories "Normalized" and "Sorted" in the dimension of data attributes, as these are less common. Therefore, Error Bar chart is classified into 6 data attributes subcategories for each orientation, resulting in 12 subcategories in total, as shown in Figure 15.

1. Base (single positive data set, unordered)
2. Grouped (multiple positive data sets, adjacent)
3. Stacked (multiple positive data sets laying atop one another)
4. Diverging (multiple stacked data sets expanding from a central axis)
5. With-Negative (data sets including negative values)
6. Reverse (exclusively negative data sets)

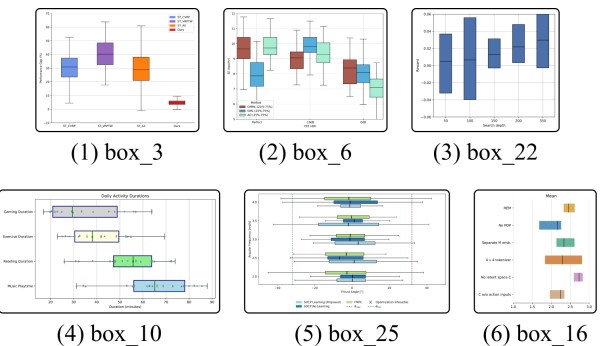

Figure 14: Examples of Box chart subcategories.

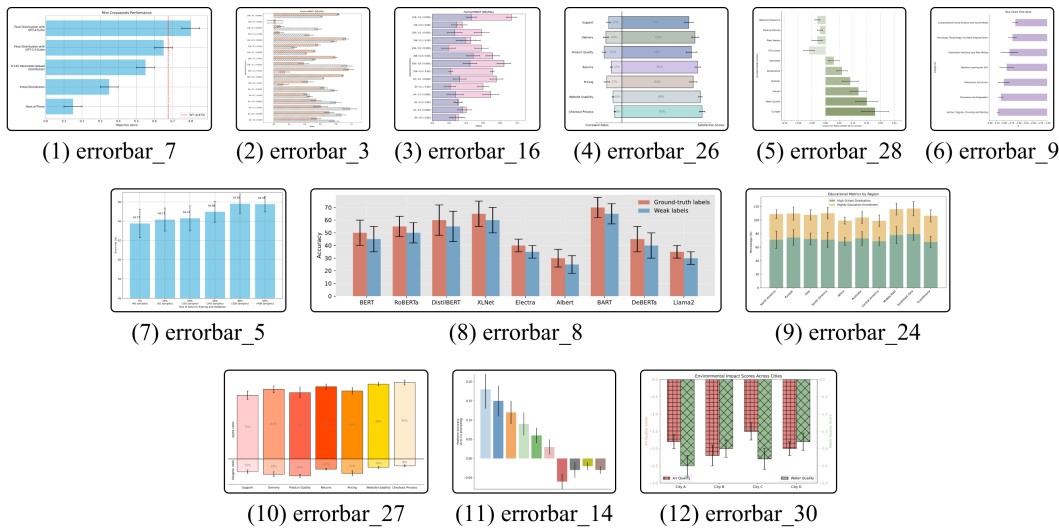

Figure 15: Examples of Error Bar chart subcategories.

**Error Point**: Error Point chart enhances the classic Scatter chart by introducing error bars to each data point, conveying the inherent variability or uncertainty of the data. When classifying Error Point chart, a pivotal consideration lies in the orientation and symmetry of the error bars, leading us to define 5 key characteristics that govern their taxonomy. These characteristics—symmetry and asymmetry in both horizontal and vertical orientations, coupled with a composite category encompassing both directions—culminate into 5 comprehensive subcategories, as illustrated in Figure 16. Here we contend the following delineations:

1. Vertical-Horizontal Orientation: Distinguishing the direction of error bars, which can profoundly affect the interpretation of the data.

2. Symmetry-Asymmetry: Acknowledging whether error bars exhibit a mirrored consistency or an uneven distribution across data points.

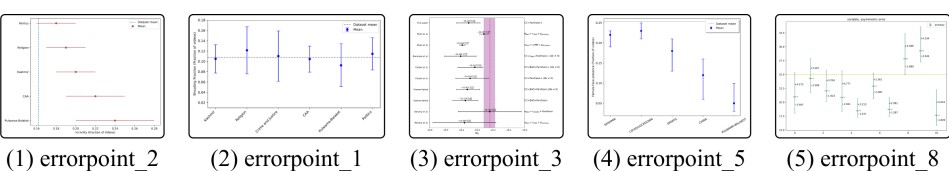

Figure 16: Examples of Error Point chart subcategories.

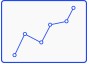 **Line**: Line chart is classified based on three primary attributes: data grouping, error visualization methods, and dot-line characteristics for grouped data, as shown in Figure 17.

1. Data grouping: Single (individual dataset) or Grouped (multiple datasets).

2. Error visualization methods: Base (no error), Striped-Error (striped fill patterns), and Marker-Error (markers above and below data points).

3. Dot-Line Characteristics: For single dataset, No-Marker (data points without markers), Marker (data points with markers). For grouped datasets, Diff-Color (different colors for each group), Diff-Marker (different marker shapes for each group), and Diff-Line (different line styles for each group).

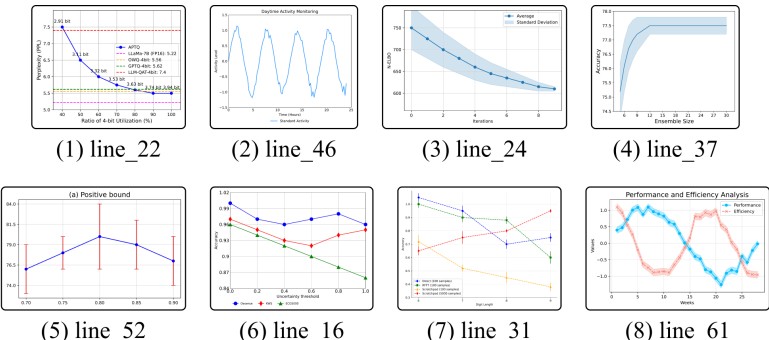

| (1) line_22 | (2) line_46 | (3) line_24 | (4) line_37 |
|---|---|---|---|
| (5) line_52 | (6) line_16 | (7) line_31 | (8) line_61 |

Figure 17: Examples of Line chart subcategories.

**Violin**: Violin chart is a combination of the Box chart and kernel Density chart. It provides a deep insight into the distribution of the data, indicating where individual data points fall within the overall data range. Based on the number of data groups and the shape and distribution of the violin form, the taxonomy is divided into 3 subcategories, as shown in Figure 18.

1. Base (standard shape, single data group)

2. Grouped-Symmetrical (standard shape, multiple data groups)

3. Grouped-Departed (half shapes joined, multiple data groups)

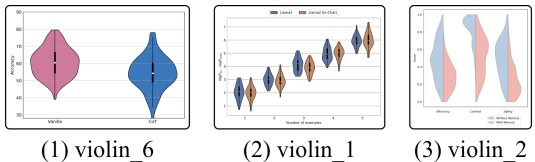

| (1) violin_6 | (2) violin_1 | (3) violin_2 |
|---|---|---|

Figure 18: Examples of Violin chart subcategories.

**Radar**: Radar chart, also known as spider chart or star plot, is a graphical method of displaying multivariate data on a two-dimensional plane. It is particularly useful for showing performance metrics or skill assessments across multiple areas. Radar chart is classified based on three primary attributes: the number of data grouping, area fill, and dot-line characteristics for grouped data, as shown in Figure 19.

1. Data Grouping: Base (a single dataset) or Grouped (multiple datasets).

2. Area Fill: FillArea (areas within the radar chart filled) or NoFillArea (areas without fill to emphasize the outline).

3. Dot-Line Characteristics: For single dataset, NoMarker (dots without markers), Marker (dots with markers). For grouped datasets, Diff-Color (different colors for each group), Diff-Line (different line styles for each group), and Diff-Marker (different marker shapes for each group).

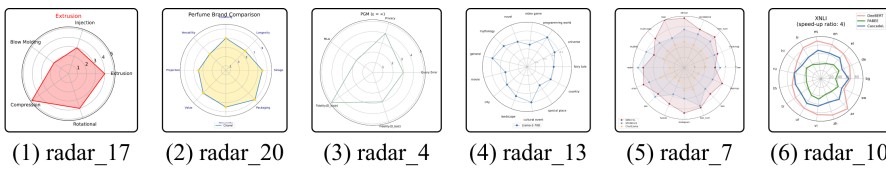

(1) radar_17    (2) radar_20    (3) radar_4    (4) radar_13    (5) radar_7    (6) radar_10

Figure 19: Examples of Radar chart subcategories.

**Pie**: Pie chart is a circular statistical graphic that divides a circle into slices to illustrate numerical proportion, whereas ring charts, also known as donut charts, utilize a hollow circle to serve a similar purpose. The presence of a ring, the number of layers, and the highlighted segment are the primary attributes in the classification of pie and ring charts, as shown in Figure 20.

1. Hollowness: Pie (no ring) or Ring (with a ring).

2. Layering: SingleLayer (single data series) or MultiLayer (multiple series or categories).

3. Highlighting: Base (without highlighted segments) or Explode (with one or more segments emphasized to capture viewer attention).

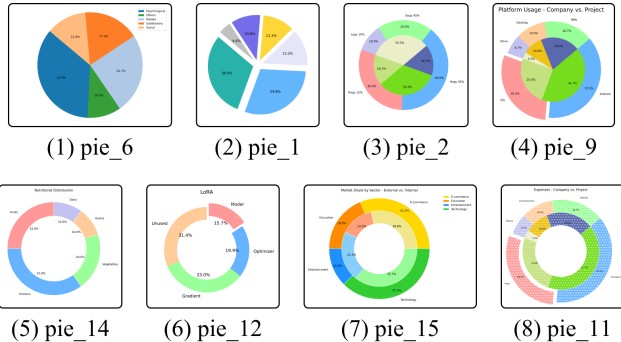

(1) pie_6    (2) pie_1    (3) pie_2    (4) pie_9

(5) pie_14    (6) pie_12    (7) pie_15    (8) pie_11

Figure 20: Examples of Pie and Ring chart subcategories.

**Density**: Density chart conveys the concentration and distribution of data within a space, often used to depict the magnitude or frequency across different areas or intervals. The orientation and data grouping are the primary attributes of the taxonomy of density plots, as shown in Figure 21.

1. Orientation: Vertical (y-axis as density) or Horizontal (x-axis as density).

2. Data Grouping: Base (single dataset) or Grouped (multiple datasets).

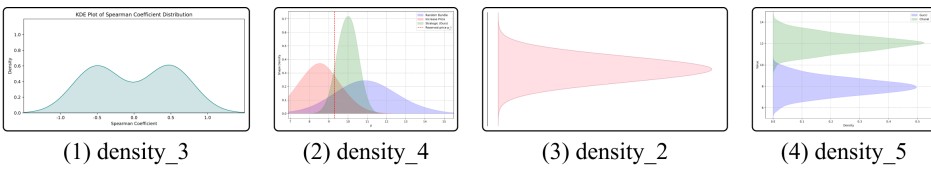

(1) density_3    (2) density_4    (3) density_2    (4) density_5

Figure 21: Examples of Density chart subcategories.

**Graph**: Graph chart commonly represents relationships and interconnected data through nodes (also known as vertices) and edges. It is widely used to depict networks, pathways, and complex inter-dependencies. The taxonomy of graph charts is based on the directionality and weight of the edges, resulting in 4 subcategories, as shown in Figure 22.

1. Directionality: Directed (edges have direction) or Undirected (edges have no direction).

2. Weight: Weighted (edges have weight) or Unweighted (edges have no weight).

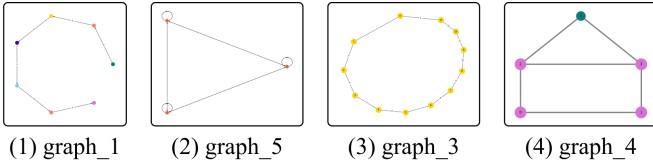

Figure 22: Examples of Graph chart subcategories.

**Quiver**: Quiver chart, also known as vector field plot, is used to display the magnitude and direction of vectors across a two-dimensional plane. This type of chart is particularly useful in physics and engineering to represent velocity fields, gradients, or other vector-based data spatially. The taxonomy of quiver charts is based on the vector quantity and data grouping, resulting in 4 subcategories, as shown in Figure 23.

1. Vector Quantity: Simple (limited vectors) or Field (vector field).
2. Data Groups: Single (single group) or Grouped (multiple groups).

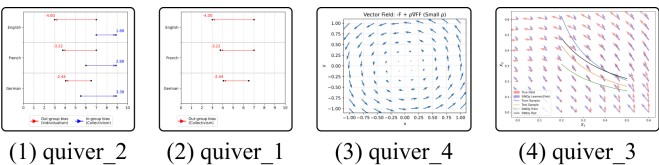

Figure 23: Examples of Quiver chart subcategories.

**Contour**: Contour chart, also called contour map or isoline graph, is used to represent three-dimensional data on a two-dimensional plane by plotting contour lines that connect points of equal value. This method is especially useful in fields like meteorology and geography, where it visually communicates variations in terrain elevation or changes in meteorological elements like temperature and pressure. The taxonomy of Contour chart is based on the representation of the contour lines, resulting in 3 subcategories, as shown in Figure 24.

1. Line (line representation)
2. Fill-Area (color-filled representation)
3. Combination (both line and color-filled)

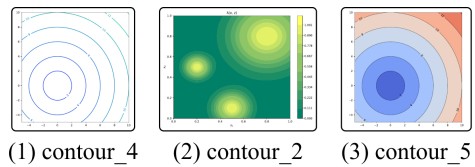

Figure 24: Examples of Contour chart subcategories.

**Histogram**: Histogram chart, often referred as hist chart, is a representation of data distribution where the data is grouped into ranges or "bins" and illustrated as bars to show the frequency of data points within each bin. It is particularly useful for identifying patterns or anomalies in the data, such as skewness, peaks, or gaps in the distribution. The taxonomy of histogram charts is based on the data grouping and positioning, resulting in 3 subcategories, as shown in Figure 25.

1. Base (single dataset)
2. Overlaid (overlapping multiple datasets)
3. Stacked (stacked multiple datasets)

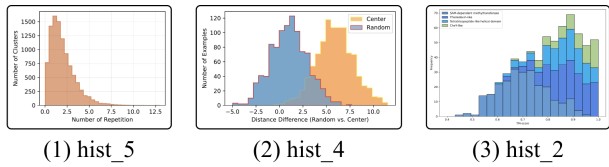

|              |              |              |
| :----------: | :----------: | :----------: |
| (1) hist_5   | (2) hist_4   | (3) hist_2   |

Figure 25: Examples of Histogram chart subcategories.

**Treemap**: Treemap displays hierarchical data through nested rectangles, where each branch of the tree is represented by a rectangle that contains smaller rectangles corresponding to sub-branches. This method allows for efficient use of space, enabling the viewer to quickly compare sizes and proportions within the hierarchy, and is especially useful for analyzing large datasets to reveal relationships and patterns. The taxonomy of treemap charts is based on the compactness and edge presence, resulting in 4 subcategories, as shown in Figure 26.

1. Tight-Edge (compact with border)
2. Tight-NoEdge (compact without border)
3. Loose-Edge (loose with border)
4. Loose-NoEdge (loose without border)

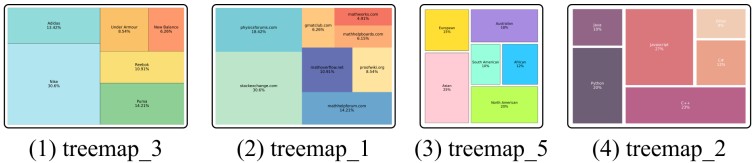

|                |                |                |                |
| :------------: | :------------: | :------------: | :------------: |
| (1) treemap_3  | (2) treemap_1  | (3) treemap_5  | (4) treemap_2  |

Figure 26: Examples of Treemap chart subcategories.

**Area**: Area chart is a graphical representation where data points are connected by line segments and the area between the line and the axis is filled with color or patterns, providing a sense of volume. It is particularly useful for visualizing the cumulative magnitude of values over time, allowing for a clear perception of trends and changes in the data series. The taxonomy of area charts is based on the presence of markers, resulting in 2 subcategories as shown in Figure 27.

1. Base (without markers)
2. Marker (with markers)

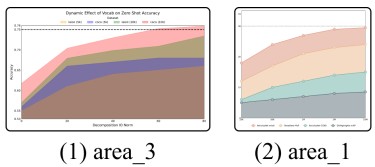

|              |              |
| :----------: | :----------: |
| (1) area_3   | (2) area_1   |

Figure 27: Examples of Area chart subcategories.

**3D charts**: 3D chart extends two-dimensional charting into three dimensions with spatial representations, offering an added layer of depth to represent additional data variables or to enhance visual appeal. Based on the above chart types and the additional surface representation in 3D space, we classify 3D chart into 5 subcategories, as shown in Figure 28.

1. Scatter (3D scatter chart)
2. Surface (3D surface chart)
3. Line (3D line chart)

4. Bar (3D bar chart)

5. Density (3D density plot)

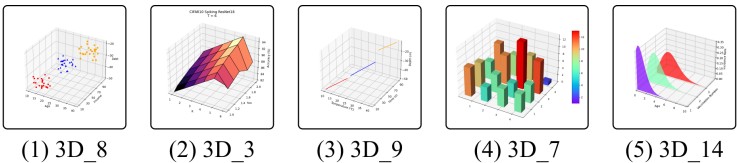

(1) 3D_8     (2) 3D_3     (3) 3D_9     (4) 3D_7     (5) 3D_14

Figure 28: Examples of 3D chart subcategories.

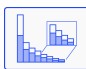 **PIP**: PIP chart insets a magnified or reduced portion of the main plot to highlight key data features, as in Figs. 29 (1) and (2). They enhance readability for complex datasets and facilitate comparative analyses, as demonstrated in Figs. 29 (3) and (4).

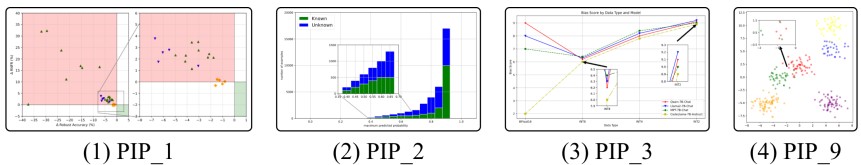

(1) PIP_1     (2) PIP_2     (3) PIP_3     (4) PIP_9

Figure 29: Examples of PIP charts.

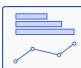 **Multidiff**: A Multidiff chart combines at least two different chart types across multiple subplots, with each subplot presenting one type. Derived from the categories above and using diverse layouts, Multidiff charts offer numerous configurations, as shown in Fig. 30.

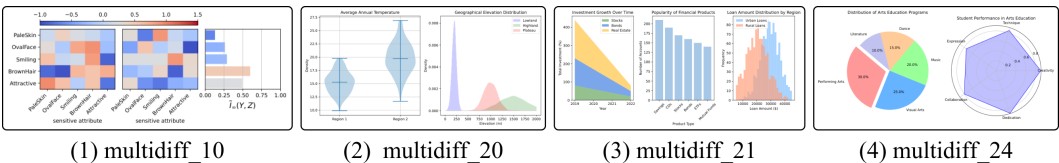

(1) multidiff_10     (2) multidiff_20     (3) multidiff_21     (4) multidiff_24

Figure 30: Examples of Multidiff charts.

**Combination**: Combination chart merges features from different chart types into one plot, offering multilayered presentation. Unlike Multidiff chart with multiple subplots for different categories, Combination chart displays multiple categories in a single plot. As in Fig. 31 (2), a scatter plot illustrates the data distribution while adjacent density plots detail the axis-specific spread. Additional examples are shown in Fig. 31.

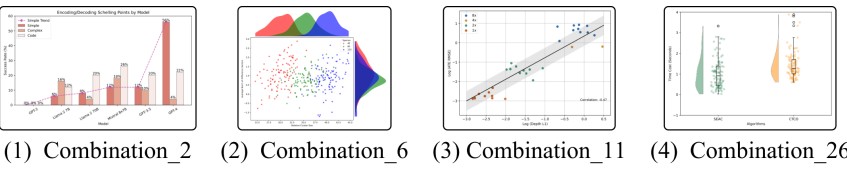

(1) Combination_2     (2) Combination_6     (3) Combination_11     (4) Combination_26

Figure 31: Examples of Combination charts.

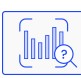 **HR**: An HR chart is one that defies the above 21 categories or is challenging to identify. HR chart is typically modified from common charts with distinctive features like custom visual arrangement or atypical markers, as shown in Fig.32.

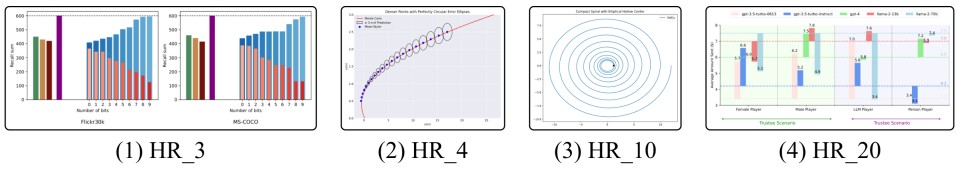

(1) HR_3  (2) HR_4  (3) HR_10  (4) HR_20

Figure 32: Examples of HR charts.

# E DETAILS OF EVALUATION METRICS

In this section, we present the details of evaluation metrics, including GPT-4o score for high-level metrics and Text Score, Layout Score, Type Score and Color Score for low-level metrics. For more information, please refer to our source code.

## E.1 GPT-4O SCORE

We employ GPT-4o (OpenAI, 2024) to assess the extend to which the generated figure corresponds to the ground-truth figure. The specific content of the prompt is presented in Fig. 33. Specifically, we input both the generated and the ground-truth figures into the GPT-4o simultaneously. Then, GPT-4o is instructed to evaluate the similarity between the two figures, taking into account six dimensions: text, layout, type, data, style, and clarity. Subsequently, GPT-4o outputs a score ranging from 0 to 100 to represent the degree of similarity between the figures.

---

**Prompt for GPT-4o Score**

You are an excellent judge at evaluating visualization chart plots. The first image (reference image) is created using ground truth matplotlib code, and the second image (AI-generated image) is created using matplotlib code generated by an AI assistant. Your task is to score how well the AI-generated plot matches the ground truth plot.

### Scoring Methodology:
The AI-generated image's score is based on the following criteria, totaling a score out of 100 points:

1. Chart Types (20 points): Does the AI-generated image include all chart types present in the reference image (e.g., line charts, bar charts, etc.)?
2. Layout (10 points): Does the arrangement of subplots in the AI-generated image match the reference image (e.g., number of rows and columns)?
3. Text Content (20 points): Does the AI-generated image include all text from the reference image (e.g., titles, annotations, axis labels), excluding axis tick labels?
4. Data (20 points): How accurately do the data trends in the AI-generated image resemble those in the original image and is the number of data groups the same as in the reference image?
5. Style (20 points): Does the AI-generated image match the original in terms of colors (line colors, fill colors, etc.), marker types (point shapes, line styles, etc.), legends, grids, and other stylistic details?
6. Clarity (10 points): Is the AI-generated image clear and free of overlapping elements?

### Evaluation:
Compare the two images head to head and provide a detailed assessment. Use the following format for your response:

—
Comments:
- Chart Types: ${your comment and subscore}
- Layout: ${your comment and subscore}
- Text Content: ${your comment and subscore}
- Data: ${your comment and subscore}
- Style: ${your comment and subscore}
- Clarity: ${your comment and subscore}
Score: ${your final score out of 100}
—
Please use the above format to ensure the evaluation is clear and comprehensive.

---

Figure 33: Prompt for GPT-4o Score.

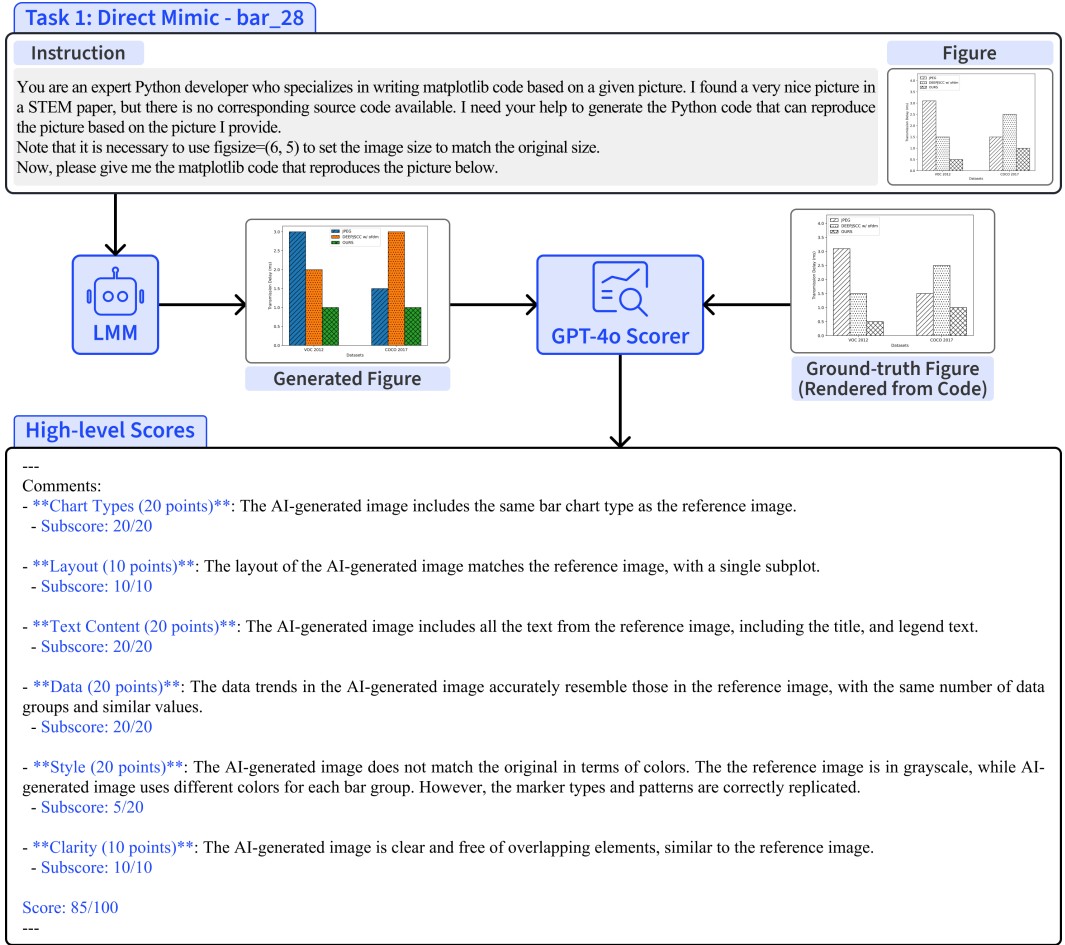

Figure 34: Example of GPT-4o's scoring results for a Direct Mimic example (bar_28).

**Stability of Evaluation with GPT-4o Score.** We conduct the high-level evaluation for GPT-4o on the Direct Mimic task for 5 times to assess the stability of GPT-4o Score. The result indicate a mean GPT-4o Score of $83.4$ with a standard deviation of 0.08, demonstrating the stability of GPT-4o Score.

**Cost of Evaluation with GPT-4o Score.** A single-round evaluation with GPT-4o Score on the Direct Mimic task approximately costs $5.25. Utilizing OpenAI's batch services further reduces this cost to $2.63 per round.

**Longevity of Evaluation with GPT-4o Score.** The GPT-4o Score metric is designed with long-term viability in mind. Our approach provides a meta-evaluation framework that can adapt to evolving language models. While currently leveraging GPT-4o, the method is model-agnostic and can be implemented with other advanced LLMs such as Claude or Gemini. This adaptability ensures that as more capable models emerge, the evaluation process can seamlessly transition to utilize these superior systems. Importantly, this approach has demonstrated increasing correlation with human assessments as model capabilities improve, as evidenced by studies like AlpacaEval (Li et al., 2023) and MT-Bench (Zheng et al., 2023).

**GPT-4o Score Examples.** To demonstrate our evaluation framework, we present GPT-4o's scoring examples for both Direct Mimic and Customized Mimic tasks. Figs. 34 to 37 show the evaluation results for two representative charts of different complexity levels (bar_28 and CB_29). These examples demonstrate how GPT-4o systematically evaluates various aspects of chart reproduction across different chart complexities and task types.

**Correlation Coefficient Comparison with CLIP Score.** CLIP Score (Radford et al., 2021) is widely used for assessing figure similarity. However, our preliminary experiments indicate that it struggles

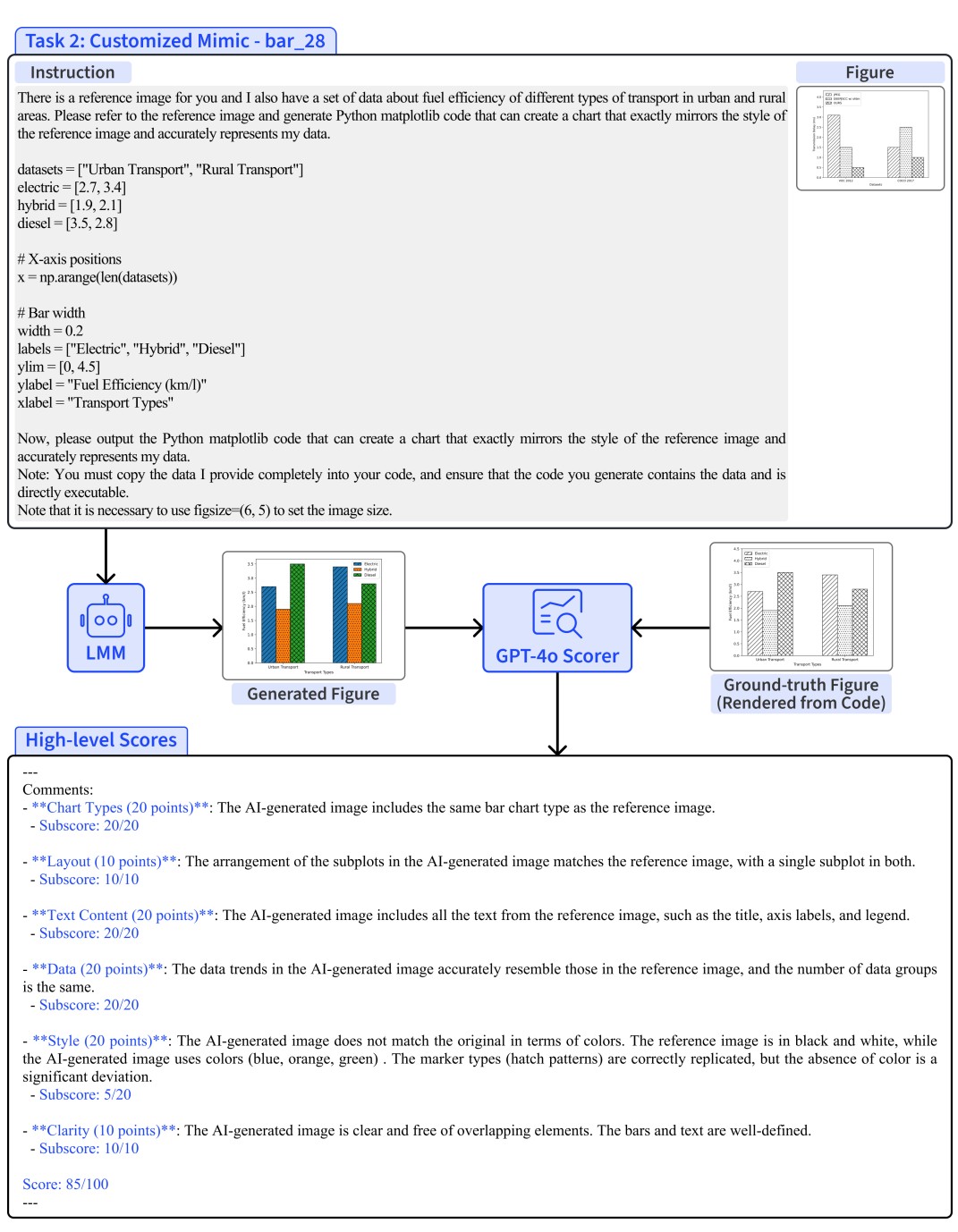

Figure 35: Example of GPT-4o's scoring results for a Customized Mimic example (bar_28).

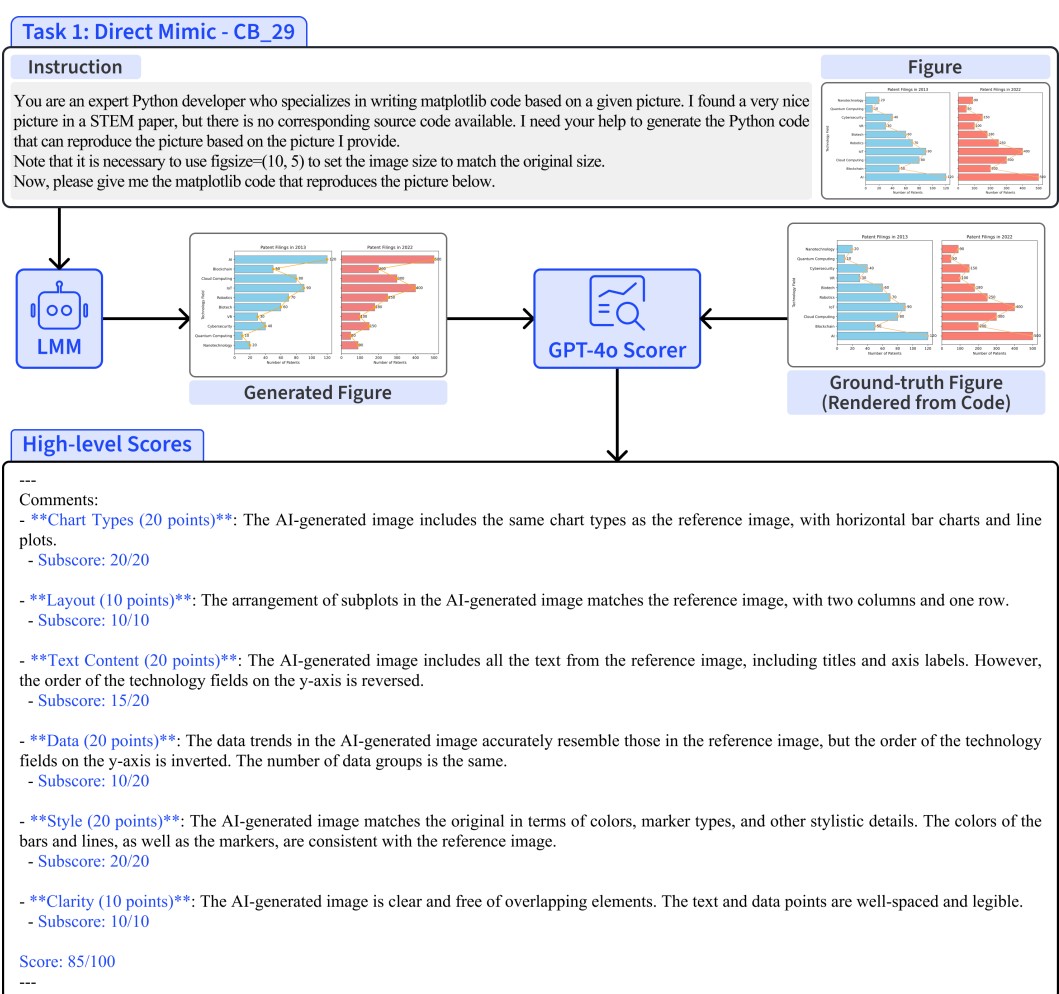

Figure 36: Example of GPT-4o's scoring results for a Direct Mimic example (CB_29).

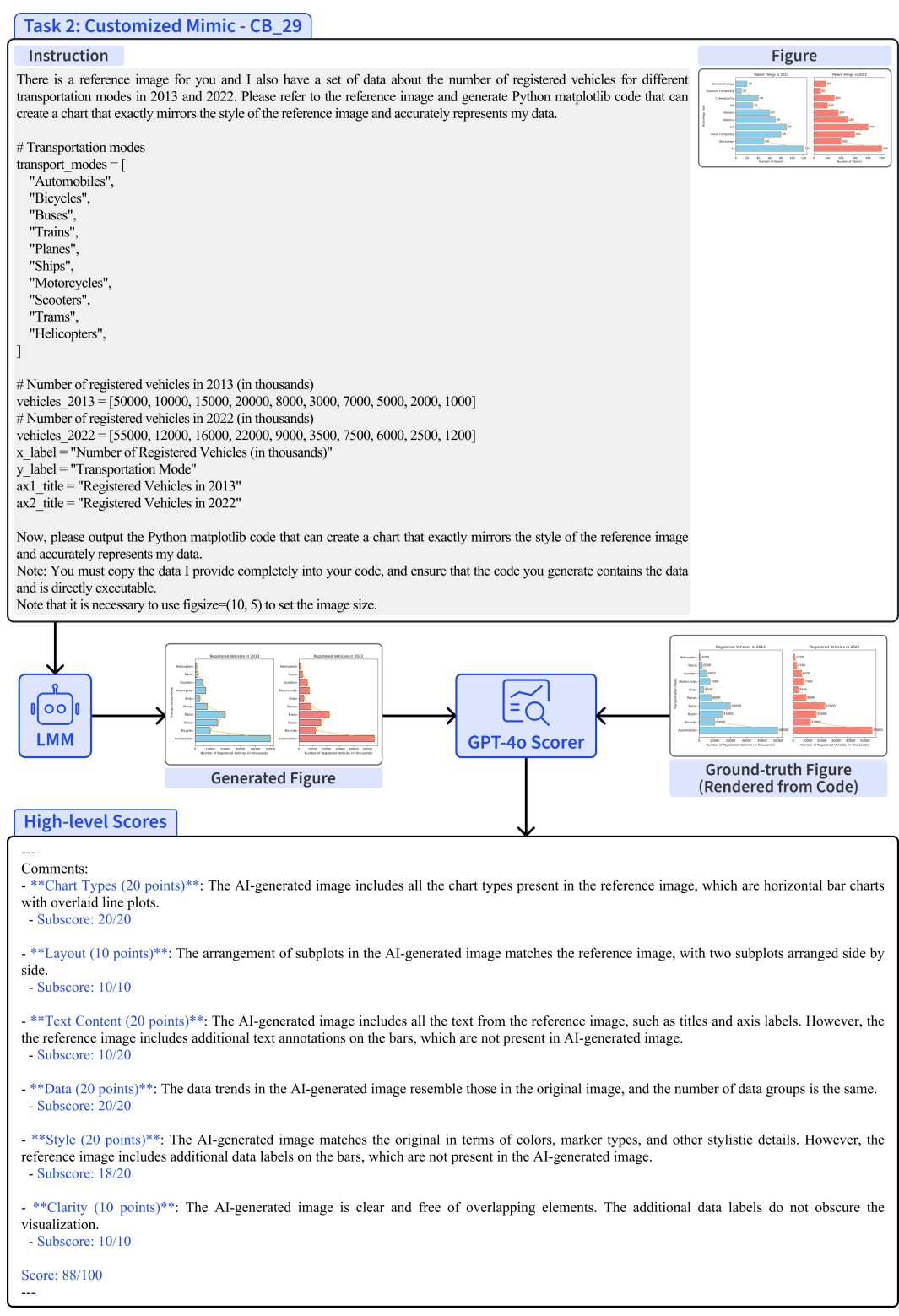

Figure 37: Example of GPT-4o's scoring results for a Customized Mimic example (CB_29).

to differentiate variations in types and other critical elements in charts, resulting in a low correlation coefficient of 0.53 with human evaluation. In contrast, as stated in Sec. 4.3, GPT-4o Score achieves a correlation coefficient of 0.70 with human evaluation. Therefore, we adopt GPT-4o Score as the high-level evaluation metric.

## E.2 TEXT SCORE

Listing 1: An exemplary Python code for logging text information.

```python
from matplotlib.backends.backend_pdf import RendererPdf

drawed_texts = []

def log_function(func):
    def wrapper(*args, **kwargs):
        global drawed_texts

        text_string = args[4]
        drawed_texts.append( text_string )

        return func(*args, **kwargs)

    return wrapper

RendererPdf.draw_text = log_function(RendererPdf.draw_text)
```

In order to accurately capture the textual content presented in the rendered figures, our code tracer monitors the function used to add text to the rendered PDF, logging each textual element. An exemplary Python code is provided in Listing 1, where we record the text elements by adding a log wrapper to the "draw_text()" function in the matplotlib package. Subsequently, we employ this approach to extract the text information from both the generated code and the ground-truth code.

Based on these two groups of texts, precision is defined as the ratio of the number of correctly captured ground-truth text to the total number of text in the generated figure. Recall is defined as the ratio of the number of correctly captured ground-truth text to the total number of text in the ground-truth figure. The F1-score, calculated using precision and recall, serves as Text Score.

## E.3 LAYOUT SCORE

Listing 2: An exemplary Python code for logging layout information.

```python
def get_gridspec_layout_info(fig):

    """
    Get the layout information of a given figure.
    Args:
        fig (matplotlib.figure.Figure): The figure to extract layout
            information from.
    Returns:
        layout_info (list): A list of dictionaries, each containing layout
            information for each subplot in the figure.
    """

    layout_info = {}
    for ax in fig.axes:
        spec = ax.get_subplotspec()
        if spec is None:
            continue
        gs = spec.get_gridspec()
        nrows, ncols = gs.get_geometry()
        row_start, row_end = spec.rowspan.start, spec.rowspan.stop - 1
        col_start, col_end = spec.colspan.start, spec.colspan.stop - 1
```

```
        layout_info[ax] = dict(nrows=nrows, ncols=ncols,
            row_start=row_start, row_end=row_end, col_start=col_start,
            col_end=col_end)
    layout_info = list(layout_info.values())
    return layout_info

layout_info = get_gridspec_layout_info(fig=plt.gcf())
```

The layout refers to the organization of subplots within a chart figure. In each figure implemented using matplotlib, multiple axes objects are present, each containing layout information that specifies its position within the figure. To analyze the layout, we iterate through each axis object in the figure and obtain their respective layout information. An exemplary Python code to accomplish this process is provided in Listing 2, where the position information of each axis is recorded. Subsequently, we gather the layout information from both the ground-truth code and the generated code.

Based on these two groups of layout information, precision is defined as the ratio of the number of correctly identified ground-truth layouts to the total number of layouts in the generated figure. Recall is defined as the ratio of the number of correctly identified ground-truth layouts to the total number of layouts in the ground-truth figure. The F1-score, calculated using precision and recall, serves as Layout Score.

### E.4    TYPE SCORE

Listing 3: An exemplary Python code for logging type information.

```
from matplotlib.axes import Axes
import inspect

called_functions = {}

def log_function(func):
    def wrapper(*args, **kwargs):

        file_name = inspect.getfile(func)
        name = file_name + "/" + func.__name__
        called_functions[name] = called_functions.get(name, 0) + 1
        result = func(*args, **kwargs)
        return func(*args, **kwargs)

    return wrapper

Axes.bar = log_function(Axes.bar)
```

The matplotlib package provides a variety of functions for easily generating diverse chart types, such as using "axes.bar()" to create bar charts. By monitoring the invocations of these functions, we can identify the types of charts being utilized. A successful invocation of a plot functions indicates the incorporation of a specific chart type in the final visualization. Listing 3 presents an exemplary Python code that demonstrates this approach, where we introduce a logger to the "bar()" function. Subsequently, we gather the chart types from both the generated and ground-truth code.

Based on these two groups of chart types, precision is defined as the ratio of the number of correctly identified ground-truth chart types to the total number of chart types in the generated figure. Recall is defined as the ratio of the number of correctly identified ground-truth chart types to the total number of chart types in the ground-truth figure. The F1-score, calculated using precision and recall, serves as Type Score.

### E.5    COLOR SCORE

Listing 4: An exemplary Python code for logging color information.

```
from matplotlib.axes import Axes
```

```python
import inspect

drawed_colors = []

def log_function(func):
    def wrapper(*args, **kwargs):

        func_name = inspect.getfile(func) + "/" + func.__name__
        result = func(*args, **kwargs)

        for item in result:
            color = item.get_facecolor()
            drawed_colors.append( func_name + "--" + color )

Axes.bar = log_function(Axes.bar)
```

In the matplotlib package, each plot function returns a chart type instance at the end of the function invocation. These instances contain various attributes, including those related to color properties, such as facecolor, edgecolor and colormap. To assess the color attributes, we employ the code tracer that captures the color information of each chart type instance. An example Python code demonstrating this approach is provided in Listing 4. Subsequently, we gather the color information from both the ground-truth code and the generated code and calculate the similarity between them.

It is noteworthy that through preliminary experiments, we find that using exact color matching resulted in very low color similarity, as even slight variations would result in a similarity score of zero. To address this issue, we employ the CIEDE2000 color difference formula (Luo et al., 2001), which converts the matching value between two colors from a discrete $[0, 1]$ scale to a continuous range between $0$ and $1$. Finally, we calculate the maximum color similarity between the sets of colors in the ground-truth code and the generated code. Precision is defined as the ratio of maximum color similarity to the total number of color in the generated figure. Recall is defined as the ratio of maximum color similarity to the total number of color in the ground-truth figure. The F1-score, calculated using precision and recall, serves as Color Score.

## F    MODEL CONFIGURATIONS AND PROMPTING METHODS

### F.1    GENERATION CONFIGURATIONS

Following previous setups (Wang et al., 2023b; Shi et al., 2024), for open-weight models, we set the temperature $\tau = 0.1$ to achieve optimal results, while for proprietary models, we set the temperature $\tau = 0$ for greedy decoding. For all models, we set the maximum generation length to 4096. Additionally, we use BF16 for model inference for open-weight models. All models are inferred on A100 80G GPU.

### F.2    PROMPTS

We provide prompts for Direct, HintEnhanced, SelfReflection and Scaffold Prompting in Figs. 38 to 42. for Direct Prompting, we meticulously design a separate prompt for open-source models to achieve optimal results, as shown in Fig. 38. while the prompt for proprietary Models is shown in Fig. 39.

---

**Prompt for Direct Prompting (Open-Weight Models)**

You are an expert Python developer who specializes in writing matplotlib code based on a given picture. I found a very nice picture in a STEM paper, but there is no corresponding source code available. I need your help to generate the Python code that can reproduce the picture based on the picture I provide.

Please note that it is necessary to use figsize=({width}, {height}) to set the image size to match the original size. Additionally, I will not provide you with the actual data in the image, so you have to extract the actual data by yourself and based on the extracted data to reproduce the image. Ensure that the code you provide can be executed directly without requiring me to add additional variables.
Now, please give me the matplotlib code that reproduces the picture below.

---

Figure 38: Prompt for Direct Prompting (Open-Weight Models). {text} in blue font represents placeholders, which varies according to different test examples.

---

**Prompt for Direct Prompting (Proprietary Models)**

You are an expert Python developer who specializes in writing matplotlib code based on a given picture. I found a very nice picture in a STEM paper, but there is no corresponding source code available. I need your help to generate the Python code that can reproduce the picture based on the picture I provide.

Note that it is necessary to use figsize=({width}, {height}) to set the image size to match the original size.
Now, please give me the matplotlib code that reproduces the picture below.

---

Figure 39: Prompt for Direct Prompting (Proprietary Models). {text} in blue font represents placeholders, which varies according to different test examples.

---

**Prompt for HintEnhanced Prompting**

You are an expert Python developer who specializes in writing matplotlib code based on a given picture. I found a very nice picture in a STEM paper, but there is no corresponding source code available. I need your help to generate the Python code that can reproduce the picture based on the picture I provide.

To ensure accuracy and detail in your recreation, begin with a comprehensive analysis of the figure to develop an elaborate caption. This caption should cover, but not be limited to, the following aspects:
1. Layout Analysis: e.g., identify the picture's composition, noting the presence and arrangement of any subplots.
2. Chart Type Identification: e.g., determine how many charts within a subplot. Are they independent, or do they share a common axis?
3. Data Analysis: e.g., summarize the data trend or pattern.
4. Additional Features: e.g.,identify any supplementary elements such as legends, colormaps, tick labels, or text annotations that contribute to the figure's clarity or aesthetic appeal.

Now, given the picture below, please first output your comprehensive caption and then use the caption to assist yourself to generate matplotlib code that reproduces the picture. Note that it is necessary to use figsize=({width}, {height}) to set the image size to match the original size.

Figure 40: Prompt for HintEnhanced Prompting. {text} in blue font represents placeholders, which varies according to different test examples.

---

**Prompt for Scaffold Prompting**

You are an expert Python developer who specializes in writing matplotlib code based on a given picture. I found a very nice picture in a STEM paper, but there is no corresponding source code available. I need your help to generate the Python code that can reproduce the picture based on the picture I provide.

I will provide you with two images. The first image is the original picture. The second image is the picture overlaid with a dot matrix of a shape of {dot_matrix_height} * {dot_matrix_width} to help you with your task, and each dot is labeled with two-dimensional coordinates (x,y). Within each column, the x-coordinate increases from top to bottom, and within each row, the y-coordinate increases from left to right.

Please first use this dot matrix as reference anchors to generate the description of the picture (e.g., between dot A and dot B is something) and generate matplotlib code that reproduces the picture.

Note that it is necessary to use figsize=({width}, {height}) to set the image size to match the original size.

Figure 41: Prompt for Scaffold Prompting. {text} in blue font represents placeholders, which varies according to different test examples.

---

**Prompt for SelfReflection Prompting**

You are an expert Python developer who specializes in writing matplotlib code based on a given picture. I have a code for implementing the reference picture as follows:

Now, I have the Python matplotlib code for implementing the reference picture as follows:
```python
{python_code}
```

The rendered picture of the code is:

Now, please compare whether the renderer picture is the same as the reference picture. The difference may cover, but not be limited to, the following aspects:
1. Chart Types: Does the AI-generated image include all chart types present in the reference image (e.g., line charts, bar charts, etc.)?
2. Layout: Does the arrangement of subplots in the AI-generated image match the reference image (e.g., number of rows and columns)?
3. Text Content: Does the AI-generated image include all text from the reference image (e.g., titles, annotations, axis labels), excluding axis tick labels?
4. Data: How accurately do the data trends in the AI-generated image resemble those in the original image and is the number of data groups the same as in the reference image?
5. Style: Does the AI-generated image match the original in terms of colors (line colors, fill colors, etc.), marker types (point shapes, line styles, etc.), legends, grids, and other stylistic details?
6. Clarity: Is the AI-generated image clear and free of overlapping elements?

- If the generated picture matches the reference, please output the original implementation code.
- If there are discrepancies, first list the specific differences between the two pictures. Then, modify the existing code to address these differences, ensuring the revised code is capable of reproducing the reference picture. Finally, output the revised code.

Note that it is necessary to use figsize=({width}, {height}) to set the image size to match the original size.

---

Figure 42: Prompt for SelfReflection Prompting. {text} in blue font represents placeholders, which varies according to different test examples.

F.3 CASES OF DIFFERENT PROMPTING METHODS

We provide cases of HintEnhanced, SelfReflection and Scaffold Prompting in Figs. 43 to 46. For an analysis of error cases related to Direct Prompting, please refer to Appendix I.

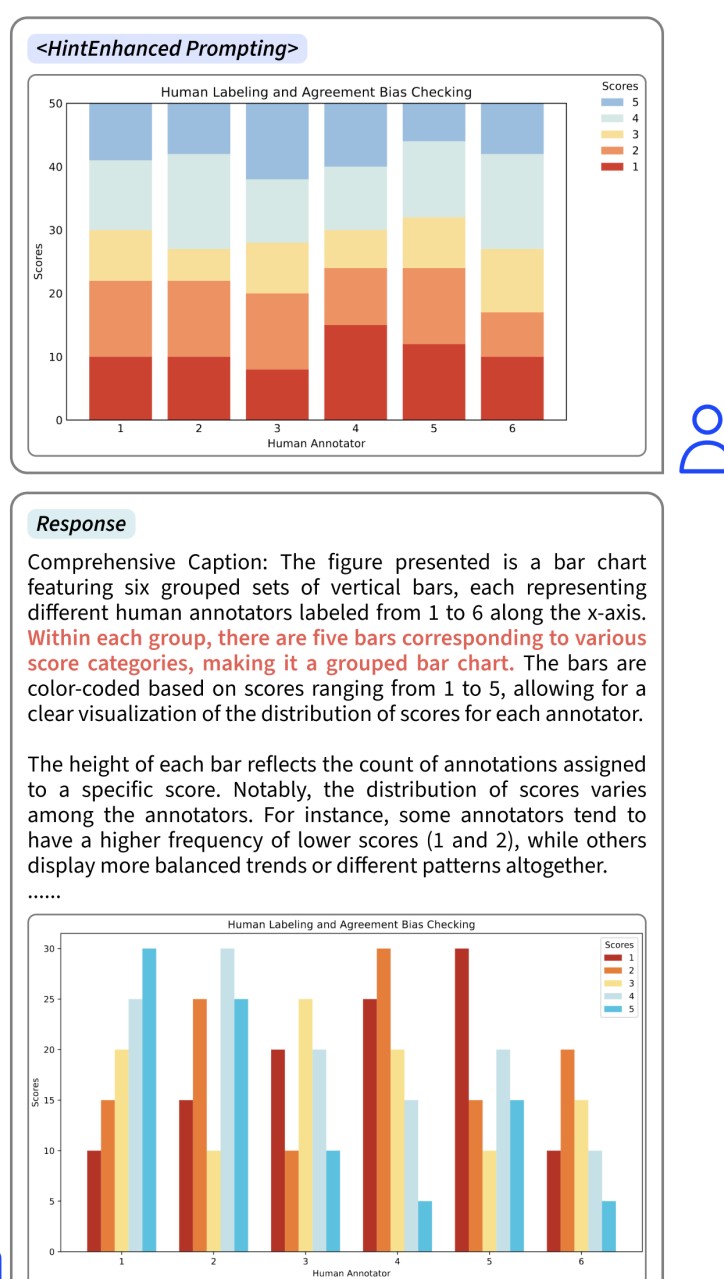

Figure 43: An error case of HintEnhanced Prompting. The **text** highlighted in red is LMM's incorrect understanding regarding provided figures. The remaining part of response is omitted with "......".

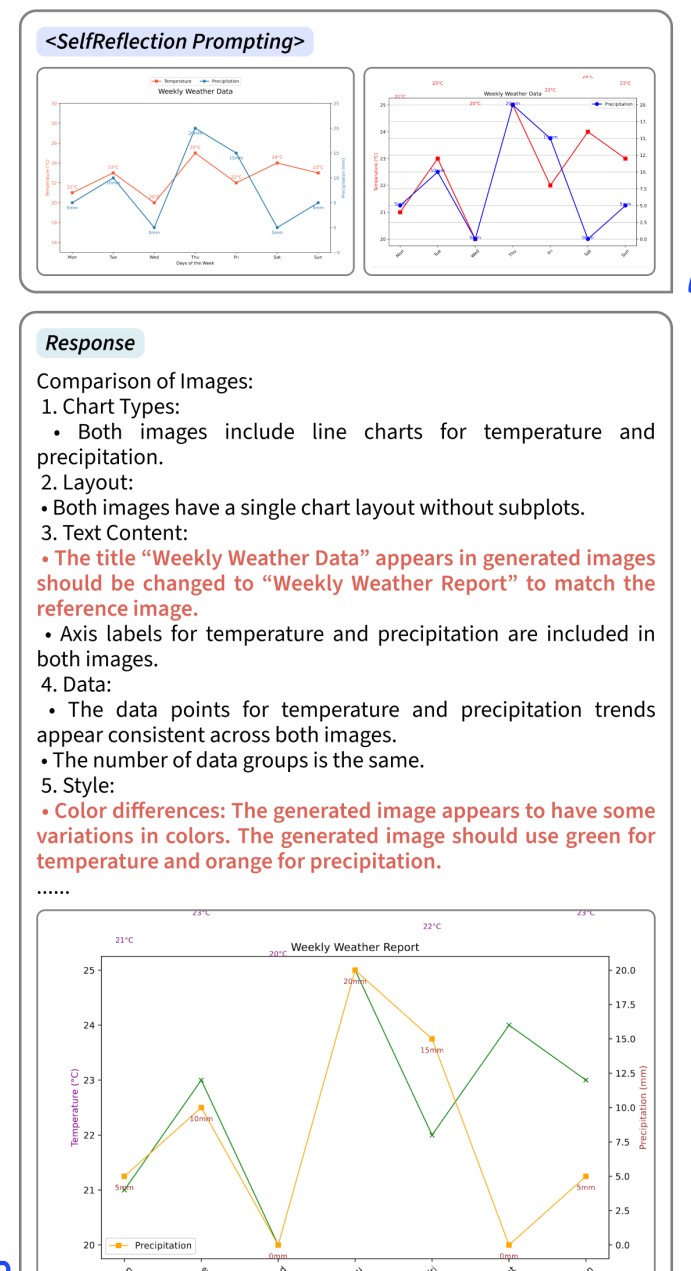

Figure 44: An error case of SelfReflection Prompting. The **text** highlighted in red is LMM's incorrect understanding regarding provided figures. The remaining part of response is omitted with "......".

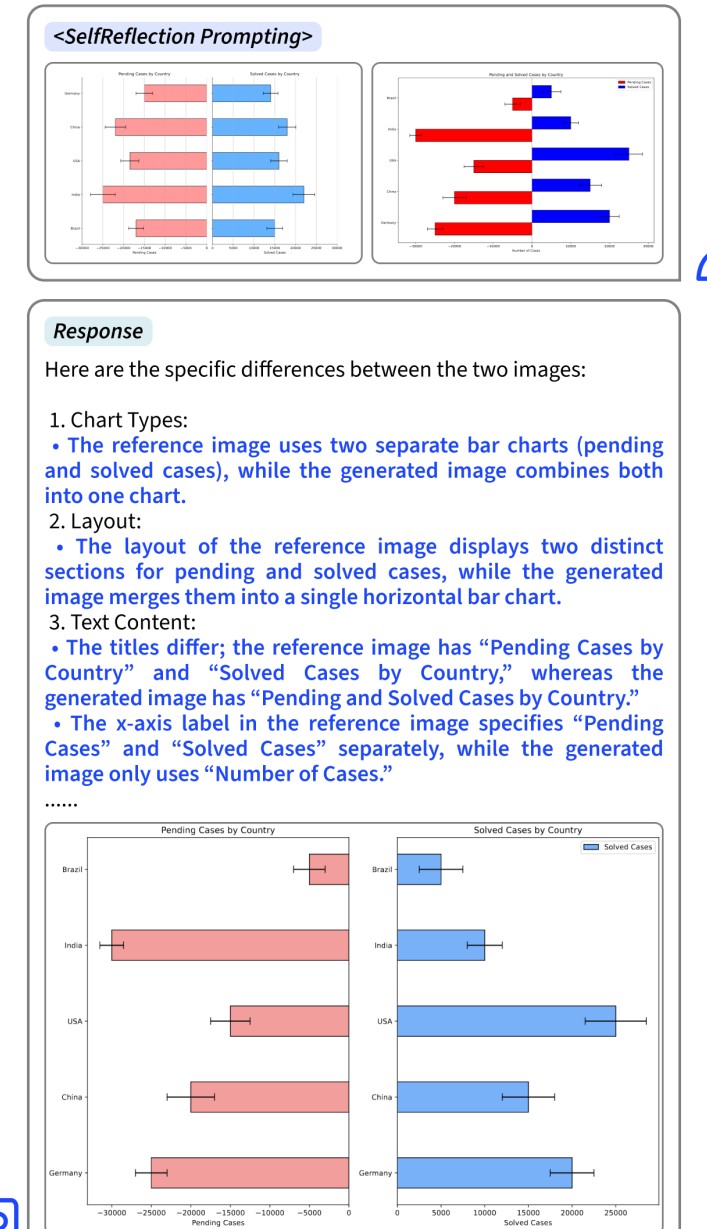

Figure 45: An good case of SelfReflection Prompting. The **text** highlighted in blue is LMM's correct reflection regarding provided figures. The remaining part of response is omitted with "......".

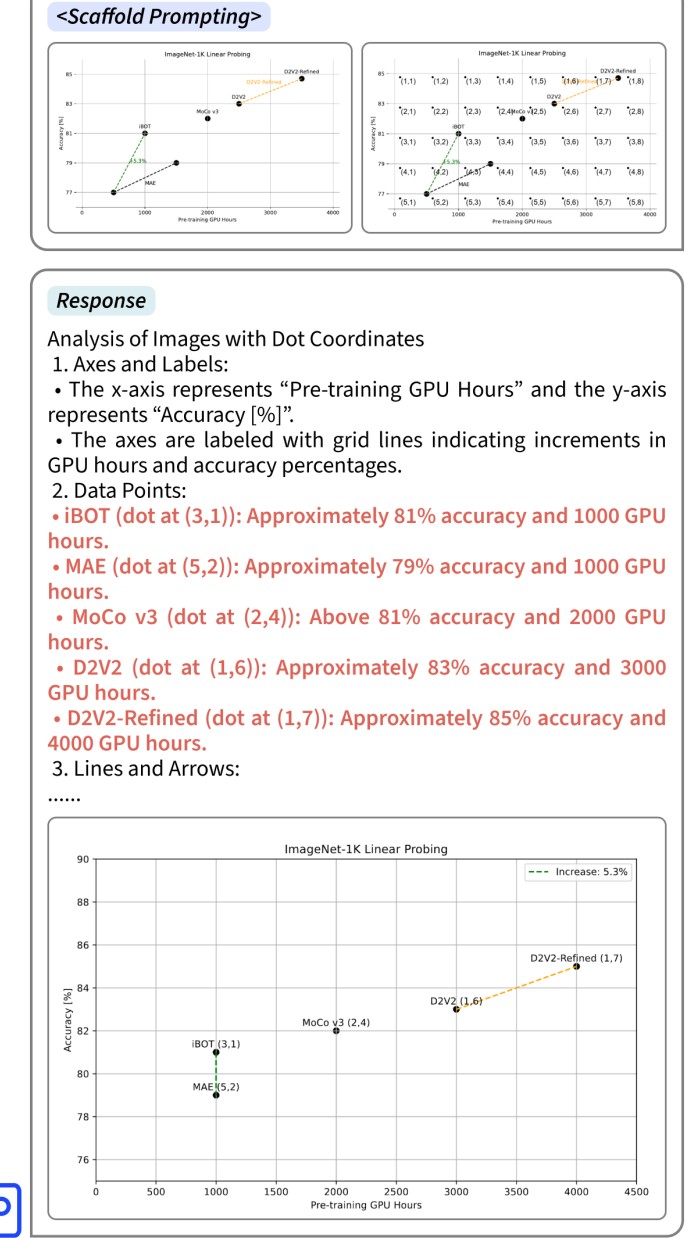

Figure 46: An error case of Scaffold Prompting. The **text** highlighted in red is LMM's incorrect understanding regarding provided figures. The remaining part of response is omitted with "......".

## F.4 DETAILS OF MODELS

We list the evaluated models in Tab. 9.

| Model | Params | Language Model | Vision Model | Model Code/API |
|---|---|---|---|---|
| GPT-4o | - | - | - | `gpt-4o-2024-05-13` |
| Claude-3-opus | - | - | - | `claude-3-opus-20240229` |
| GeminiProVision | - | - | - | `gemini-pro-vision` |
| InternVL2-Llama3-76B | 76.0B | Llama-3-70B-Instruct | InternViT-6B | `OpenGVLab/InternVL2-Llama3-76B` |
| LLaVA-Next-Yi-34B | 34.8B | Yi-34B | CLIP ViT-L/14 | `llava-hf/llava-v1.6-34b-hf` |
| InternVL2-26B | 26.0B | InternLM2-20B | InternViT-6B | `OpenGVLab/InternVL2-26B` |
| Cogvlm2-llama3-chat-19B | 19.2B | Llama-3-8B-Instruct | EVA2-CLIP-E | `THUDM/cogvlm2-llama3-chat-19B` |
| Phi-3-Vision-128K-Instruct | 4.2B | Phi-3 | CLIP ViT-L/14 | `microsoft/Phi-3-vision-128k-instruct` |
| IDEFICS2-8B | 7.6B | Mistral-7B | SigLip-400M | `HuggingFaceM4/idefics2-8b` |
| LLaVA-Next-Mistral-7B | 7.6B | Mistral-7B | CLIP ViT-L/14 | `llava-hf/llava-v1.6-mistral-7b-hf` |
| DeepSeek-VL-7B | 7.3B | DeekSeek-7B | SAM-B & SigLIP-L | `deepseek-ai/deepseek-vl-7b-chat` |
| MiniCPM-Llama3-V2.5 | 8.4B | Llama3-8B-Instruct | SigLip-400M | `openbmb/MiniCPM-Llama3-V-2_5` |
| Qwen2-VL-7B | 8.2B | Qwen2-7B | ViT-600M | `Qwen/Qwen2-VL-7B-Instruct` |
| InternVL2-8B | 8.1B | InternLM2.5-7B | InternViT-300M | `OpenGVLab/InternVL2-8B` |
| InternVL2-4B | 4.2B | Phi-3 | InternViT-300M | `OpenGVLab/InternVL2-4B` |
| Qwen2-VL-2B | 2.6B | Qwen2-1.5B | ViT-600M | `Qwen/Qwen2-VL-2B-Instruct` |
| InternVL2-2B | 2.2B | InternLM2-1.8B | InternViT-300M | `OpenGVLab/InternVL2-2B` |

Table 9: Model code/API of the evaluated models.

## G CORRELATION WITH HUMAN EVALUATION

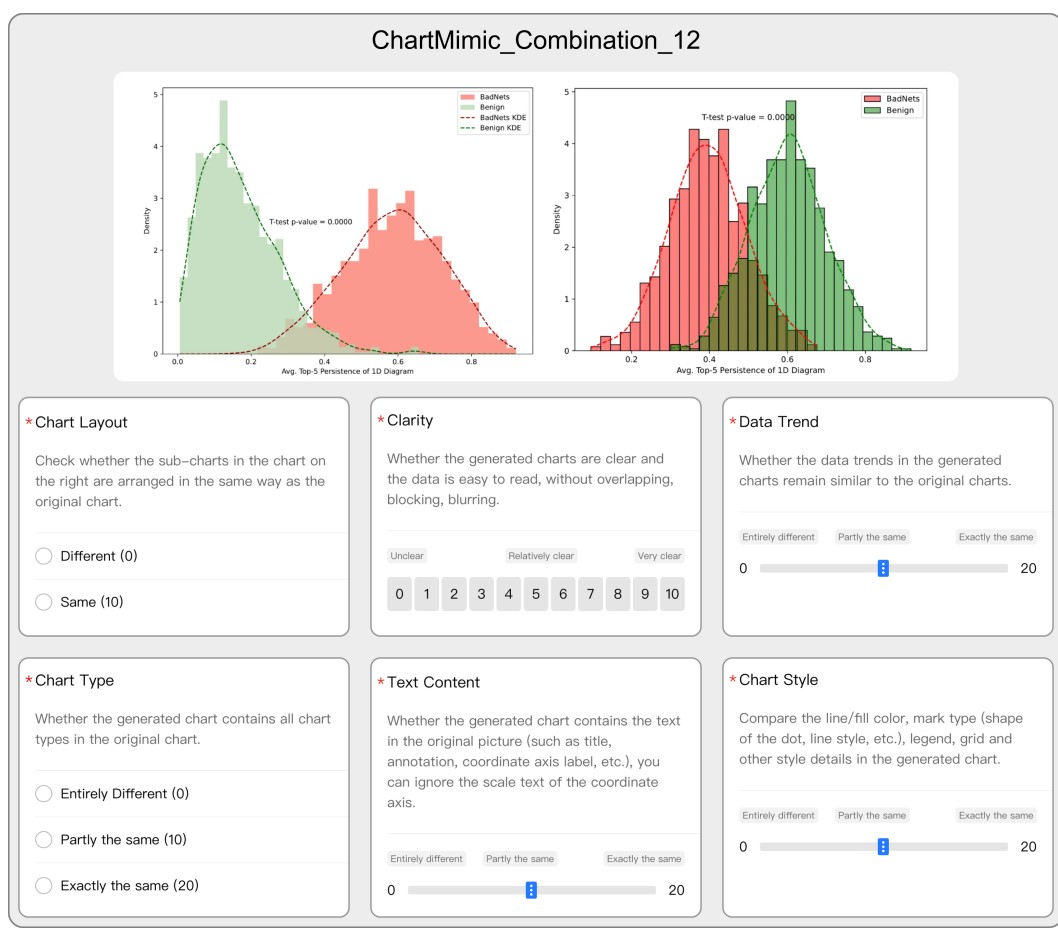

Figure 47: A screenshot of the human evaluation questionnaire.

In order to evaluate the reliability of the proposed automatic metrics, we calculate their correlation with human evaluations. Specifically, a selection of $300$ test examples from the Direct Mimic task is utilized. Subsequently, we gather the results generated by GPT-4o using four different prompting methods (Sec. 4.2) on these test examples. This process yields $1,200$ figures to be assessed.

Each figure is independently assessed by three evaluators through a questionnaire, who assign scores on a scale of $0$ to $100$ based on the similarity between the generated figures and the ground-truth figures. The final score for each figure is calculated as the mean of scores given by the three evaluators. Our evaluators comprise volunteer graduate students holding bachelor's degrees in computer science. To facilitate their assessments, we provide a scoring rubric in the questionnaire, which closely aligns with the low-level and high-level metric evaluations. The scoring criteria encompass the following dimensions: chart type, layout, textual content, data, style, and clarity. A screenshot of the final questionnaire is available in Fig. 47.

Upon collecting the human evaluation results, we calculate the Pearson correlation coefficient to ascertain the correlation between our automatic metrics and human evaluation.

# H   CORRELATION WITH CHART UNDERSTANDING AND CODE GENERATION

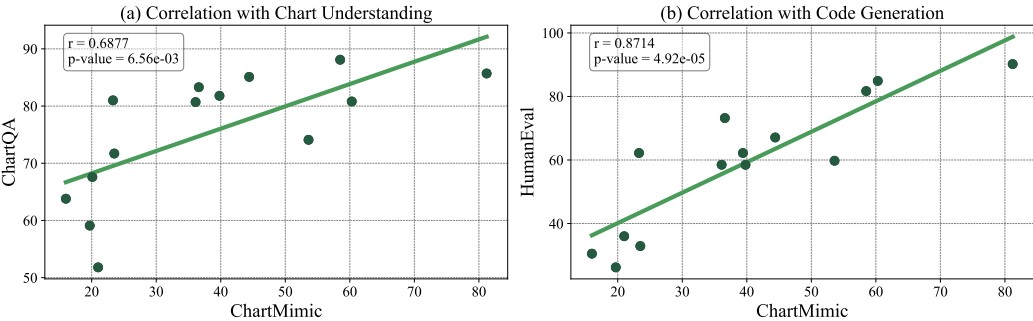

Figure 48: Performance correlation of ChartMimic with benchmarks assessing chart understanding and code generation capabilities.

In an effort to shed light on the factors enhancing performance on ChartMimic, we explore the performance correlation between ChartMimic and existing benchmarks that evaluate chart understanding and code generation capabilities. Specifically, we select ChartQA (Masry et al., 2022) as the benchmark for chart understanding and HumanEval (Chen et al., 2021) as the benchmark for code generation. We then calculate two Pearson correlation coefficients to quantify these relationships: The correlation coefficient between the performance of LMMs in the Direct Mimic task and their performance in the ChartQA task, denoted as $r_{chart}$, and the correlation coefficient between the performance of LMMs in the Direct Mimic task and the performance of their corresponding LLMs in the HumanEval task, denoted as $r_{code}$ .

As depicted in Fig. 48, the calculated values are $r_{chart} = 0.6877$ and $r_{code} = 0.8714$. These results indicate that both chart understanding and code generation abilities influence the performance in the Direct Mimic task, with code generation having a more significant impact. This implies that for future model development aimed at enhancing multimodal code generation capabilities, it is crucial to focus on the foundational code generation abilities of LMMs.

# I CASES OF ERROR ANALYSIS

We provide cases of text-related, type-related and color-related errors in Figs. 49 to 54. These cases encompass various error types mentioned in Sec. 4.4.

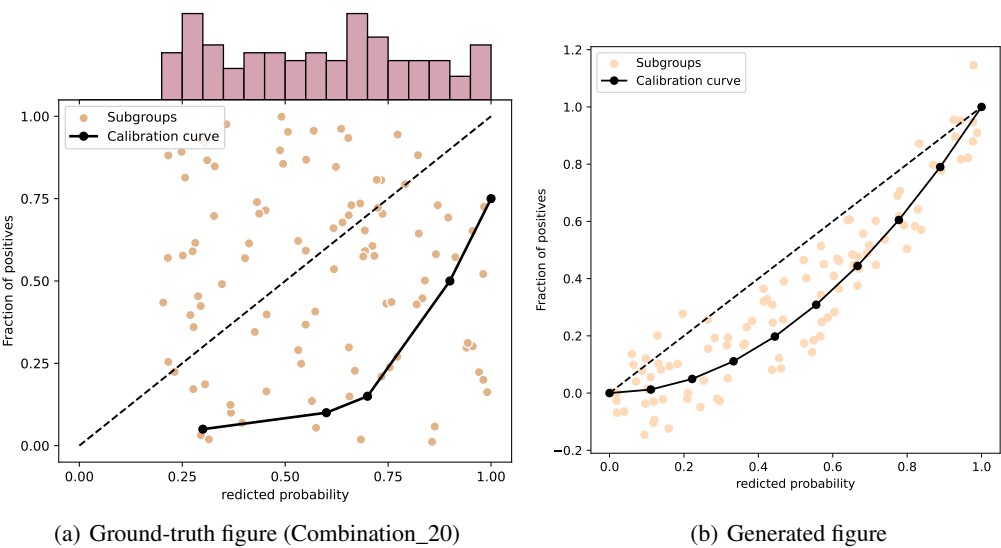

(a) Ground-truth figure (Combination_20)    (b) Generated figure

Figure 49: Error Case 1. In this case, the errors include text-related errors of the *Missing* type and type-related errors of the *Missing* type.

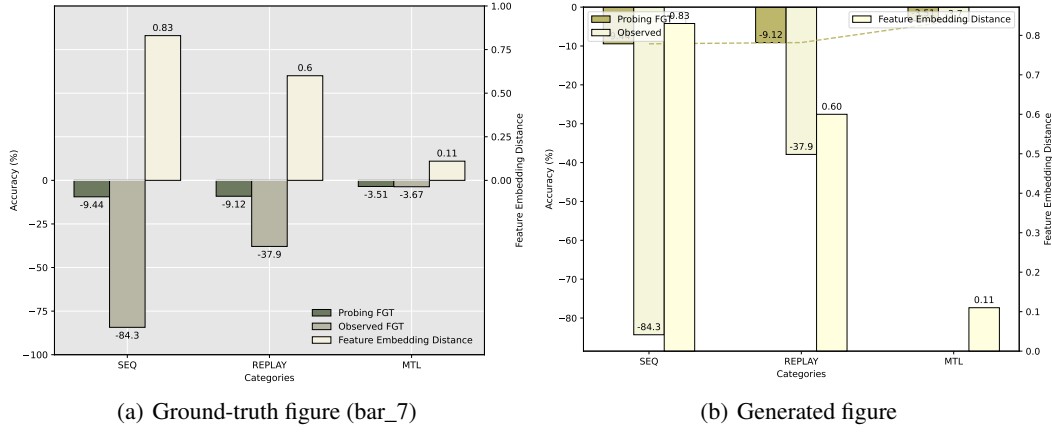

(a) Ground-truth figure (bar_7)    (b) Generated figure

Figure 50: Error Case 2. In this case, the errors include type-related errors of the *Extraneous* type, and color-related errors of the *Different* type.

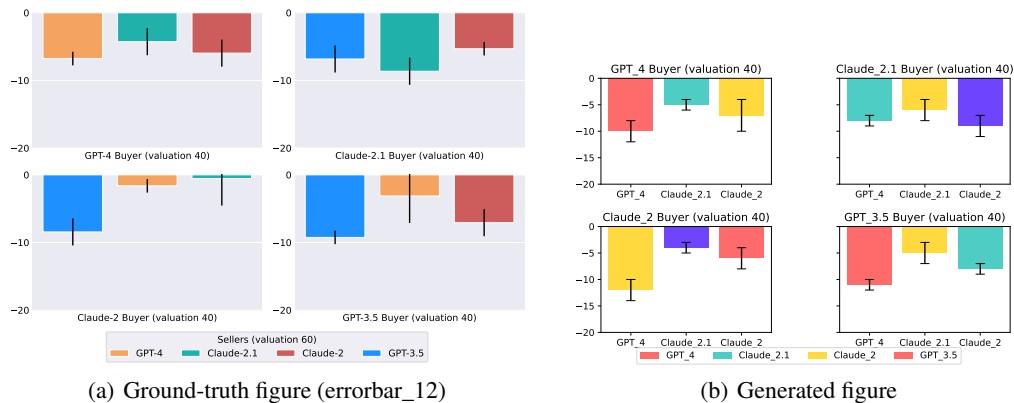

Figure 51: Error Case 3. In this case, the errors include text-related errors of the *Detail* and *Missing* type, and color-related errors of the *Different* type.

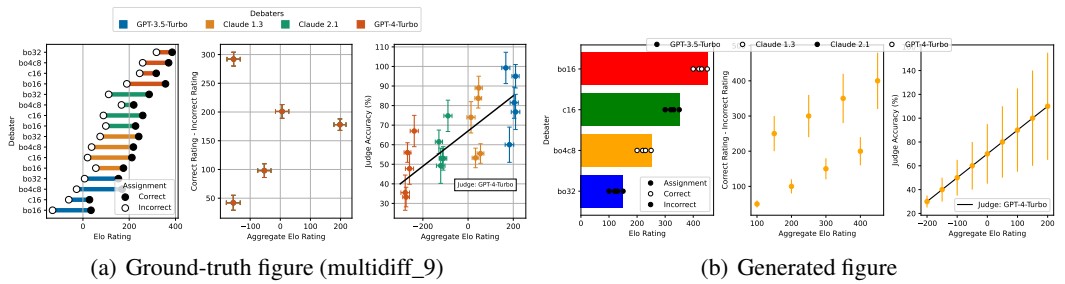

Figure 52: Error Case 4. In this case, the errors include text-related errors of the *Missing* type, type-related errors of the *Missing* type, and color-related errors of the *Different* type.

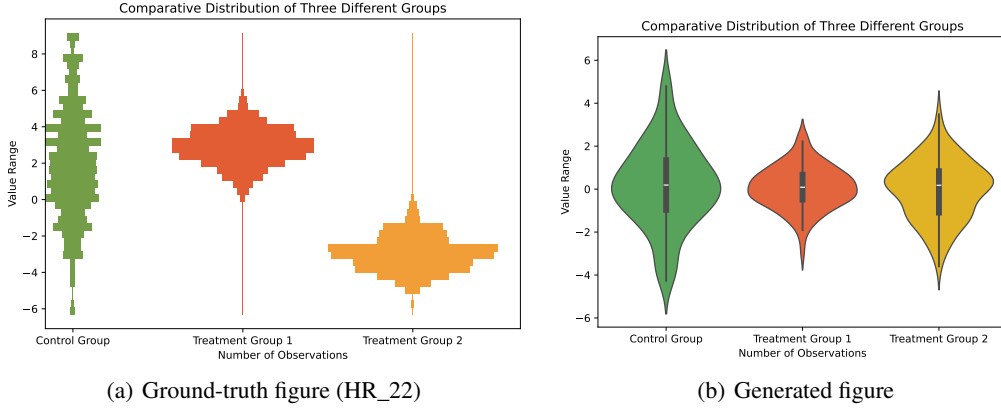

Figure 53: Error Case 5. In this case, the errors include type-related errors of the *Confusion* type, and color-related errors of the *Similar* type.

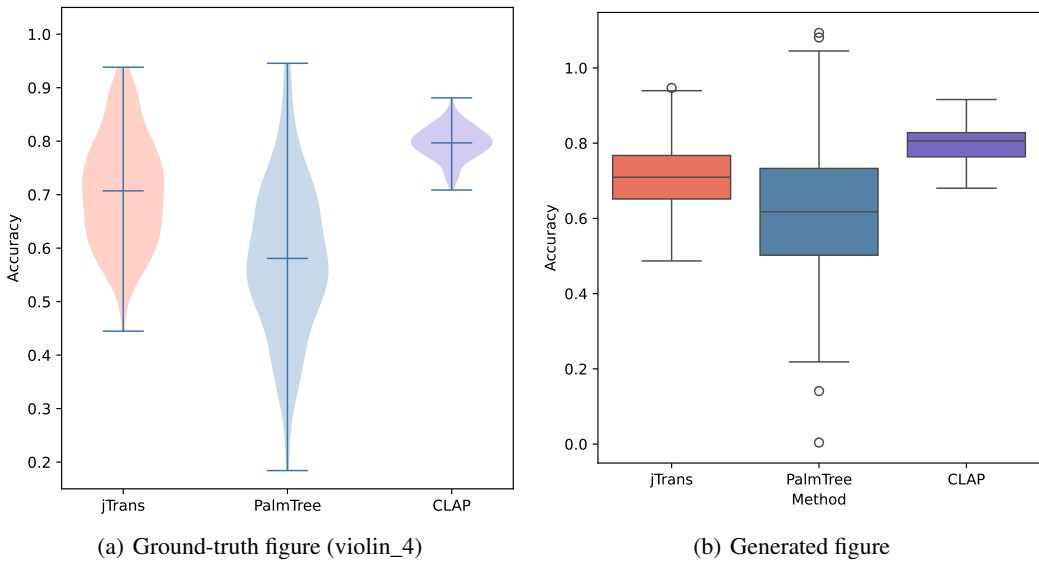

(a) Ground-truth figure (violin_4)

(b) Generated figure

Figure 54: Error Case 6. In this case, the errors include text-related errors of the *Extraneous* type, type-related errors of the *Confusion* type and color-related errors of the *Similar* type.

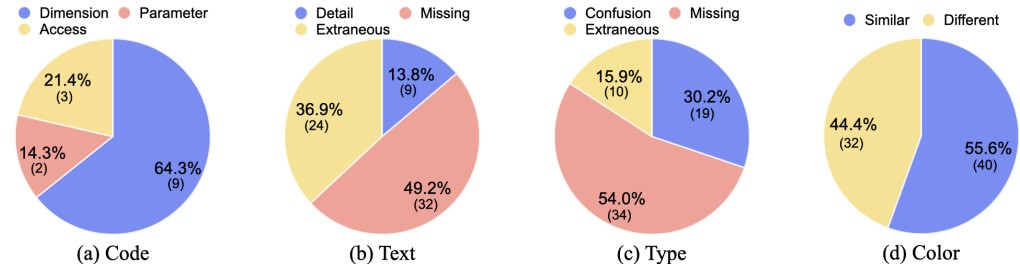

Figure 55: Error analysis of InternVL2-Llama3-76B across four error types on the Direct Mimic task. The number in brackets indicates the count of error case.

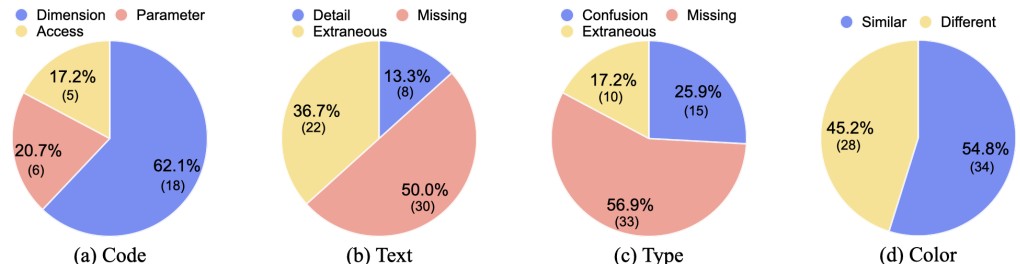

Figure 56: Error analysis of InternVL2-26B across four error types on the Direct Mimic task. The number in brackets indicates the count of error case.

## J   ERROR ANALYSIS OF OPEN-WEIGHT MODELS

We conduct detailed error analysis on representative open-weight models of different scales, sampling 100 cases for each model. For large-scale models such as InternVL2-Llama3-76B and InternVL2-26B (Fig. 55, Fig. 56), the error patterns align closely with those observed in GPT-4o. Specifically, Code-related Errors predominantly arise from dimensional inconsistencies, while Text-related Errors primarily manifest as missing elements. Type-related Errors mainly comprise missing components cases, and Color recognition demonstrates the capability to identify similar, if not identical, chromatic properties. In contrast, smaller-scale models like DeepSeek-VL-7B (Fig. 57) exhibit markedly different error distributions. Code-related issues show a significant increase in Access and Parameter Errors (approximately 53.7%), while Text-related errors demonstrate a higher prevalence of missing elements (approximately 66.7%). Type-related cases reveal a substantial increase in missing errors (approximately 78.4%). Moreover, the degradation in color recognition is more severe, with the model failing to identify even similar colors. These findings indicate that while larger models encounter challenges similar to GPT-4o, smaller models exhibit more fundamental difficulties in both code generation and visual comprehension.

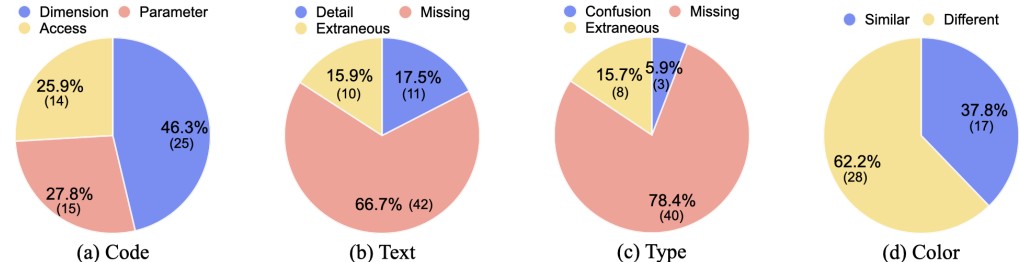

Figure 57: Error analysis of DeepSeek-VL-7B across four error types on the Direct Mimic task. The number in brackets indicates the count of error case.

## K    ETHICS, SOCIETAL IMPACT AND SCALABILITY OF CHARTMIMIC

ChartMimic is designed as a comprehensive evaluation framework for advancing LMMs, with a strong emphasis on ethical considerations, societal impact and scalability. Its primary objective is to provide researchers with a rigorous, ethically sound tool for assessing LMMs' capabilities across critical domains, including visual understanding, code generation and cross-modal reasoning.

**Ethical Considerations.**    The ethical integrity of ChartMimic is foundational to its design and implementation. Our dataset is derived from scientific domains, specifically using figures from arXiv papers distributed under the CC BY 4.0 license. This focus on scientific content significantly mitigates potential ethical concerns related to bias, representation, or sensitive personal information that often arise in datasets derived from social media or general web content. The scientific nature of our data ensures: Minimal risk of perpetuating societal biases or stereotypes; Absence of personally identifiable information; Content that is generally neutral and objective, focusing on factual scientific representations. Our manual annotation process for code generation incorporates stringent ethical controls. Annotators are trained to ensure that the information conveyed in the code does not contain any biased content and strictly adheres to the factual representation of the scientific charts. This process further reinforces the ethical robustness of our dataset.

**Societal Impact.**    By providing a benchmark rooted in scientific data, ChartMimic contributes to the development of LMMs that can better understand and interact with complex, data-driven visualizations. This capability has far-reaching positive implications for scientific communication, data analysis, and knowledge dissemination across various fields.

**Scalability.**    The architecture of ChartMimic is inherently extensible, featuring a modular code-base that facilitates the seamless integration of additional chart types and evaluation metrics. Our data collection methodology leverages the continuous update cycle of arXiv, enabling sustainable expansion of the dataset while maintaining its focus on ethically sound, scientific content. The evaluation framework also demonstrates notable scalability: The concept of low-level metric can be readily adapted to other programming environments such as JavaScript and R. The high-level metric incorporates a meta-evaluation approach using LMMs, which allows for sustainable alignment with human preferences as stronger models emerge.

