# OpenReview forum: "ChartMimic: Evaluating LMM's Cross-Modal Reasoning Capability via Chart-to-Code Generation"
_ICLR.cc/2025/Conference — ICLR 2025 Poster_

### Official Review · Reviewer_gbor · 2024-11-01

**Soundness:** 3
**Presentation:** 3
**Contribution:** 4
**Rating:** 8
**Confidence:** 5

**Summary:**

This paper presents a dataset named ChartMimic for evaluating LMM's ability to generate data visualization code based on (1) an image of a reference chart and (2) user instruction (either a reproduce-type of instruction or a custom instruction asking for creating a chart similar to the reference chart but for a new dataset). This dataset is curated from arxiv paper plots and online galleries following design guidelines to ensure both diversity of charts and high-quality ground-truth references (charts created by human).

Based on this dataset, the paper presents an evaluation pipeline, leveraging GPT-4o's visual reasoning ability to compare chart generated by model and human. Besides a high-level similarity score, the paper introduces low-level metrics considering values logged from the visualization code (texts in the plot, layout, chart type func names, color attributes) to accompany evaluation.

This paper reports evaluation results of 3 closed-sourced models and 14 open source models with varying sizes. Results show the considerable gap between most powerful closed sourced model and open source model. The reports on how prompting strategy influence also point out interesting benefits of self-reflection. It's also interesting to learn that smaller model best benefits from direct prompting.

**Strengths:**

This paper targets the practical problem of how chart authors reference existing visualizations to create their own visualizations, and the dataset/pipeline is fantastic for evaluating LMMs' ability jointly leverage visual and codegen abilities to solve coding task. In general this paper makes the following contributions:

1. Chart reusing is a practical task studied in Visualization and HCI communities, this dataset bridges the gap of LMM development and practical chart authoring tasks.
2. The dataset is sufficiently different from existing chart understanding (QA/summary etc) and codegen (HumanEval etc) that tests both visual reasoning and codegen together. It's a good addition for LMM evaluation tasks.
3. The dataset coverages a diverse set of visualizations with quite convincing human curation process.
4. The evaluate pipeline and results provide baseline for future LMM development.

In general, I find this paper interesting and would make good contribution to the field, pending responses to weakness and questions.

**Weaknesses:**

Here are some weakness of the paper.

1. The comparison with related work is a little hand-wavy. Please (1) elaborate the comparison of chart type coverage, complexity and evaluation metric difference of this paper with close related work like Plot2Code and Design2Code, (2) add a few sentences to discuss the related work with computation notebook related work (e.g., DS1000 and following workstream) since many computation notebook completion tasks involve similar libraries and plotting. If possible, also add a few sentences on whether existing results on models' data science codegen ability aligns with the evaluation results found in this paper.

2. The evaluation metric is not very convincing, especially high-level evaluation score, since existing work (e.g., Meng 2024, VisEval) has shown LMM's don't have impeccable visual reasoning abilities. I think the authors should elaborate why the metric here is convincing in the following ways:
(1) does the high-level metric provide consistent results, with different runs, and with perturbation to images (e.g., manually change a small part of the chart to investigate if the metric is still reasonably conherent)
(2) does the metric performs consistently across different types of charts? E.g., while the model may make precise assessment on popular chart types of bar chart, is the accessment consistent in less popular ones like contour chart?
(3) how does low-level metric relate to high-level metric?
(4) how does current evaluation approach compare to other LMM based evaluation approach?
(5) ideally, show how the evaluation metric is coherent with human evaluation.
Ideally, the authors should perform human evaluation to measure how good this current evaluation metric is. But if that is not the case, the authors should provide more rationales to convince the readers.

3. As the chart mimic task involves both chart understanding and code generation tasks (as comopared in reproduction task versus custom data task), this paper missed an opportunity to identify how does models' chart understanding and code generation abilities individually contribute to the overall task. Some ideas of comparison: cross reference these models' performance on codegen task and chart reading task, to see how the model's scores or rankings in individual tasks relate to their overall performance in chart mimic. (there is no need to run additional experiment, but please include a discussion section about this topic.)

**Questions:**

Please checkout the previous section for questions. Please elaborate (1) comparison to related work (2) justification for the eval metric (3) discuss how individual ability relates to overall task performance.

---

> ### Author Response · Authors · 2024-11-21
> **Response to Reviewer gbor (Part 1/4)**
>
> Thank you for your thorough and constructive feedback. We appreciate your detailed suggestions for strengthening various aspects of our work. We address each of your concerns in detail below.
>
> > **W1: The comparison with related work is a little hand-wavy. Please (1) elaborate the comparison of chart type coverage, complexity and evaluation metric difference of this paper with close related work like Plot2Code and Design2Code, (2) add a few sentences to discuss the related work with computation notebook related work (e.g., DS1000 and following workstream) since many computation notebook completion tasks involve similar libraries and plotting. If possible, also add a few sentences on whether existing results on models' data science codegen ability aligns with the evaluation results found in this paper.**
>
> R1:
>
> **1. Comparison with Closely Related Work:**
>
> Thank you for your valuable feedback. In our original manuscript, we have discussed comparisons with existing benchmarks in Sec 2.5 and Sec 5. Below, we delineate the distinctions between our work, ChartMimic, and closely related works such as Plot2Code and Design2Code, focusing on chart type coverage, complexity, and evaluation metric differences.
>
> Regarding chart type coverage and complexity, ChartMimic stands out by covering a comprehensive range of 22 chart types, each with difficulty levels as easy, medium, and hard. This results in a dataset of 4,800 human-curated figures that exhibit a high degree of diversity and complexity, outstripping the scope of Plot2Code[1].
>
> In terms of evaluation metrics, ChartMimic is distinguished by its adoption of a multi-level metric framework, which includes both high-level and low-level metrics.  Notably, our low-level metrics meticulously assess four critical elements: text, layout, type, and colour, all facilitated by a code tracer. This approach ensures a comprehensive and precise assessment compared to related works. For instance, Plot2Code[1] primarily focuses on text, while Design2Code[2] relies on less precise text detection modules, which can compromise the evaluation's accuracy.
>
> [1] Plot2Code: A Comprehensive Benchmark for Evaluating Multi-modal Large Language Models in Code Generation from Scientific Plots
>
> [2] Design2Code: How Far Are We From Automating Front-End Engineering?
>
> **2. Comparison with Computation Notebook Tasks:**
>
> We appreciate the suggestion to draw parallels with computation notebook-related tasks.
>
> Computation notebook-related benchmarks [3,4,5] concentrate on textual inputs and short code snippets outputs primarily for rendering general and simple charts. Their evaluations typically only focus on certain aspects, such as functional correctness or textual-level execution output match, without comprehensively taking into account the low-level elements and high-level visual perception of the generated charts.
>
> In contrast, ChartMimic's inputs consist of information-intensive and diverse visual charts with textual instructions, requiring LMMs to generate corresponding long code for chart rendering (as illustrated in Table 1 and Table 2). Additionally, ChartMimic offers a comprehensive evaluation through multi-level metrics, blending high-level visual similarity scores from GPT-4o with low-level detailed assessments of four critical chart elements—text, layout, type, and colour. This approach delivers a thorough and precise evaluation of LMM performance in the complex task of chart-to-code generation.
>
> [3] Ds-1000: A natural and reliable benchmark for data science code generation
>
> [4] DA-Code: Agent Data Science Code Generation Benchmark for Large Language Models
>
> [5] Execution-based Evaluation for Data Science Code Generation Models
>
>
> **3. Alignment with Existing Data Science Code Generation Studies**
>
> Thank you for your insightful suggestion to explore the alignment between our evaluation results and existing models' data science code generation capabilities. To address this, we select CIBench[6] as a benchmark for data science code generation. CIBench is a latest real-world data science code generation evaluation benchmark, which covers a broad spectrum of data science concepts by incorporating commonly used Python modules.
>
> We calculate the Pearson correlation coefficient between the performance of LMMs in the Direct Mimic task and the performance of their corresponding LLMs in CIBench, denoted as $r_{ds}$. The result is $r_{ds}=0.7916$, indicating a strong correlation between data science code generation and the ChartMimic task. Furthermore, in terms of error types, we observe that Code-related Errors, as discussed in Sec 4.4, are also significant in CIBench.
>
> [6] CIBench: Evaluating Your LLMs with a Code Interpreter Plugin.

---

> ### Author Response · Authors · 2024-11-21
> **Response to Reviewer gbor (Part 2/4)**
>
> > **W2: The evaluation metric is not very convincing, especially high-level evaluation score, since existing work (e.g., Meng 2024, VisEval) has shown LMM's don't have impeccable visual reasoning abilities. I think the authors should elaborate why the metric here is convincing in the following ways:
> (1) does the high-level metric provide consistent results, with different runs, and with perturbation to images (e.g., manually change a small part of the chart to investigate if the metric is still reasonably conherent)
> (2) does the metric performs consistently across different types of charts? E.g., while the model may make precise assessment on popular chart types of bar chart, is the accessment consistent in less popular ones like contour chart?
> (3) how does low-level metric relate to high-level metric?
> (4) how does current evaluation approach compare to other LMM based evaluation approach?
> (5) ideally, show how the evaluation metric is coherent with human evaluation. Ideally, the authors should perform human evaluation to measure how good this current evaluation metric is. But if that is not the case, the authors should provide more rationales to convince the readers.**
>
> **W2.1 Does the high-level metric provide consistent results, with different runs, and with perturbation to images (e.g., manually change a small part of the chart to investigate if the metric is still reasonably conherent)**
>
> Thank you for your insightful comment regarding the consistency and robustness of our high-level metric, the GPT-4o Score.
>
> - Consistency of the GPT-4o Score: We conduct an analysis to ensure the stability of the GPT-4o Score, as detailed in Appendix D.1 of our original manuscript, "Stability of Evaluation with GPT-4o Score". Specifically, to assess the stability, we run the GPT-4o evaluation on the Direct Mimic task five times, yielding an average score of 83.4 with a standard deviation of merely 0.08. This low deviation indicates a high level of consistency in the GPT-4o Score across different runs.
>
> - Robustness to Perturbations: Regarding the robustness to perturbations, we refer to the study in CharXiv [1], where perturbations are introduced into charts to test the robustness of various models on Chart Understanding tasks. The results demonstrate that GPT-4o is robust against such perturbations. By extension, since the GPT-4o Score primarily relies on the model's understanding capabilities of charts and its ability to compare understood information between two figures, we infer that the robustness observed in CharXiv likely applies to the GPT-4o Score scenario as well.
>
> [1] CharXiv: Charting Gaps in Realistic Chart Understanding in Multimodal LLMs
>
> **W2.2 Does the metric performs consistently across different types of charts? E.g., while the model may make precise assessment on popular chart types of bar chart, is the accessment consistent in less popular ones like contour chart?)**
>
> In Section 4.3 and Appendix F of our original manuscript, we have calculated the correlation between the high-level metric and human evaluation, as well as the low-level metric and human evaluation. To address your concern about consistency across chart types, we further report the correlation coefficients separately for Line, Combination, and HR categories, which represent the largest, medium, and smallest differences in overall correlation coefficient, respectively. The results are presented in the table below:
>
> |             | Low-level | High-level |
> |-------------|-----------|------------|
> | Overall     | 0.7681    | 0.7041     |
> | Line        | 0.7745    | 0.7294     |
> | Combination | 0.7643    | 0.7091     |
> | HR          | 0.7574    | 0.6783     |
>
> From the experimental results, it is evident that chart type has a minimal impact on the correlation between the low-level metric and human evaluation. Regarding the high-level metric, the differences are also within an acceptable range (ranging from a maximum of 0.7294 to a minimum of 0.6783). In fact, we find that in the high-level metric, GPT-4o primarily focuses on comparing the understood information between two figures, which may have little to do with the popularity of the charts.

---

> ### Author Response · Authors · 2024-11-21
> **Response to Reviewer gbor (Part 3/4)**
>
> **W2.3 How does low-level metric relate to high-level metric?**
>
> In Section 4.3 and Appendix F of our original manuscript, we have calculated the correlation between the high-Level metric and human evaluation, as well as the low-level metric and human evaluation. To address your concern, we have also computed the correlation coefficient between the high-Level and low-Level metrics. Our findings reveal a correlation coefficient of 0.5658, indicating a moderate relationship between the two metrics. This value indicates that while there is a relationship between the two metrics, it is not extremely strong, suggesting that each metric captures different aspects of the LMM's performance.
>
> As we mentioned in Sec 2.4, the high-level metric captures the overall similarity between the generated and ground-truth figures, whereas the low-level metric provides a detailed analysis of specific elements such as text, layout, type, and colour. Together, they offer a comprehensive evaluation of the model's capabilities.
>
> **W2.4 How does current evaluation approach compare to other LMM based evaluation approach?**
>
> The increasing adoption of "LMM as a judge" is indeed a significant trend in research [1,2]. In the field of data visualization, the evaluation can be broadly categorized into reference-free [3,4] and reference-based methods [5], depending on the availability of an objectively correct ground-truth figure.
>
> For reference-free evaluations, there is no objectively correct ground-truth figure to reference. Existing work[3,4] in Data Visualization has demonstrated the efficacy of LMM as a judge under this circumstance. On the other hand, reference-based evaluations primarily assess the similarity between the generated figure and the ground-truth figure. Given the presence of an objective ground-truth figure as a constraint, reference-based methods are inherently more stable and precise compared to reference-free methods [2].
> Our evaluation in ChartMimic falls under the reference-based category, relying on GPT-4o's understanding capabilities of charts and its ability to compare understood information between two figures. Additionally, as per the latest literature [6], GPT-4o's descriptive performance in charts, which includes information extraction and enumeration, is considered effective.
>
> In summary, the use of GPT-4o to compute high-level metrics is feasible. Furthermore, in Sec 4.3, our human evaluation demonstrates that it has a high correlation with human judgment..
>
> [1] MLLM-as-a-Judge: Assessing Multimodal LLM-as-a-Judge with Vision-Language Benchmark
>
> [2] A Comprehensive Study of Multimodal Large Language Models for Image Quality Assessment
>
> [3] MatPlotAgent: Method and Evaluation for LLM-Based Agentic Scientific Data Visualization
>
> [4] VisEval: A Benchmark for Data Visualization in the Era of Large Language Models
>
> [5] Plot2Code: A Comprehensive Benchmark for Evaluating Multi-modal Large Language Models in Code Generation from Scientific Plots
>
> [6] CharXiv: Charting Gaps in Realistic Chart Understanding in Multimodal LLMs
>
> **W2.5 Ideally, show how the evaluation metric is coherent with human evaluation. Ideally, the authors should perform human evaluation to measure how good this current evaluation metric is. But if that is not the case, the authors should provide more rationales to convince the readers.**
>
> We appreciate your suggestion for human evaluation studies. In our original manuscript, we have indeed addressed this aspect by calculating the correlation between high-level/low-level metrics and human evaluations in Section 4.3 and Appendix F.
>
> We are open to further discussions and please let us know if you have further questions.

---

> ### Author Response · Authors · 2024-11-21
> **Response to Reviewer gbor (Part 4/4)**
>
> > **W3: As the chart mimic task involves both chart understanding and code generation tasks (as comopared in reproduction task versus custom data task), this paper missed an opportunity to identify how does models' chart understanding and code generation abilities individually contribute to the overall task. Some ideas of comparison: cross reference these models' performance on codegen task and chart reading task, to see how the model's scores or rankings in individual tasks relate to their overall performance in chart mimic. (there is no need to run additional experiment, but please include a discussion section about this topic.)**
>
>
> Thank you for your insightful suggestion. We select ChartQA[1] as the benchmark for chart understanding and HumanEval[2] as the benchmark for code generation.
>
> We then calculate two correlation coefficients:
>
> 1.  The Pearson correlation coefficient between the performance of LMMs in the Direct Mimic task and their performance in the ChartQA task, denoted as $r_{chart}$;
>
> 2. The Pearson correlation coefficient between the performance of LMMs in the Direct Mimic task and the performance of their corresponding LLMs in the HumanEval task, denoted as $r_{code}$.
>
> The calculated values are $r_{chart}= 0.6877$  and $r_{code} = 0.8714$. These results indicate that both chart understanding and code generation abilities influence the performance in the Direct Mimic task, with code generation having a more significant impact. This implies that for future model development aimed at enhancing multimodal code generation capabilities, it is crucial to focus on the foundational code generation abilities of LMMs.
>
> In our revised manuscript, we include a discussion section (Appendix G.) that delves into these findings, highlighting the importance of both chart understanding and code generation in the context of ChartMimic.
>
>
> Thank you again for your thorough and constructive feedback. We appreciate your detailed suggestions for strengthening various aspects of our work
>
>
> [1] Chartqa: A benchmark for question answering about charts with visual and logical reasoning.
>
> [2] Evaluating large language models trained on code

---

> > ### Comment · Reviewer_gbor · 2024-12-02
> > **Thanks for clarification**
> >
> > The authors' responses include details to answer my questions. I especially appreciate the efforts in analyzing stability of LMM as a judge evaluation and code-gen ability relation to chart generation.
> > Thus I am happy to champion this paper.

---

### Official Review · Reviewer_XRf9 · 2024-11-02

**Soundness:** 3
**Presentation:** 3
**Contribution:** 3
**Rating:** 6
**Confidence:** 3

**Summary:**

This paper introduces ChartMimic, a new benchmark aimed at assessing the visually grounded code generation capabilities of large multimodal models (LMMs). ChartMimic uses visual charts and textual instructions as inputs, requiring LMMs to generate the corresponding code for chart rendering. The benchmark includes 4,800 human-curated (figure, instruction, code) triplets across various domains like Physics, Computer Science, and Economics. Additionally, the paper proposes multi-level evaluation metrics for an automatic and comprehensive assessment of the output code and the rendered charts. The evaluation of three proprietary models and 14 open-weight models shows the substantial challenges posed by ChartMimic, with advanced models like GPT-4o and InternVL2-Llama3-76B achieving average scores of 82.2 and 61.6 respectively, indicating significant room for improvement. This benchmark is expected to inspire further development of LMMs and advance the pursuit of artificial general intelligence.

The real-world example in Figure 1 effectively illustrates the benchmark's motivation, addressing the common challenge of replicating well-designed figures computationally. This enriches the research significance by exploring new avenues in multimodal input.

The paper is overall well-crafted; however, it does necessitate further clarifications in several sections. I would like to highlight specific areas where additional details from the authors would be beneficial:
Firstly, the manual annotation process described in section 2.2 ‘data curation process’ requires more comprehensive details. Given the significant involvement of human annotations, the quality of these annotations directly impacts the benchmark's effectiveness. It would be helpful if the authors could provide further information on the backgrounds of the annotators, the diversity within the team, the mechanisms for resolving any conflicts that arise during annotation, the duration of the annotation process, the versions of Python used, and approaches to managing incomplete figures. These details are crucial for assessing potential biases within the process.  The authors could include this information in an appendix or supplementary material if space is limited in the main paper.


Secondly, the manual complexity assessment of charts mentioned in section 2.3 raises questions about the classification criteria.
‘
Additionally, we conduct a manual complexity assessment for each chart, classifying them into 3 levels: easy, medium, and hard.
’
How are charts categorized into easy, medium, and hard levels? What criteria are used, and can the authors provide examples of each level to enhance the transparency of this process? If the classification relies on the subjective judgment of the annotators, how is bias controlled? If objective indicators such as the number of elements is used, more detailed information would be valuable. I suggest authors include a table or figure showing examples of charts at each complexity level, along with the specific criteria used for classification.


Additionally, the use of GPT-4O scores in the evaluation metrics introduces concerns about the inherent instability of LLMs. It would be pertinent for the authors to explain how they address these variabilities to ensure reliable evaluation outcomes. The authors could consider conducting a stability analysis of this metric by executing the evaluation repeatedly and documenting the variability in the resultant scores.

Do the authors plan to make all the data in the benchmark publicly available after publication? I recommend that during the review stage, the authors could use anonymous GitHub repositories or similar platforms to offer more detailed information about the benchmark, including data and usage tutorials. This would significantly enhance the thoroughness of the benchmark evaluation.

Lastly, I am curious if the authors have considered expanding this benchmark into a practical tool for figure-to-code tasks, possibly incorporating methodologies like Retrieval-Augmented Generation, to further its applicability and utility.

**Strengths:**

Innovative benchmark;
the paper is good-writting;
Good motivation.

**Weaknesses:**

Insufficient Detail on Annotation Process
Lack of Transparency in Complexity Assessment

**Questions:**

Can you provide more details about the manual annotation process, including annotator backgrounds, diversity, conflict resolution, and handling of incomplete figures?
What specific criteria are used to classify charts into easy, medium, and hard categories? Are these criteria subjective, or are they based on objective indicators?
How do you address the instability of LLM outputs in your evaluation metrics to ensure consistent and reliable results?
Have you considered developing this benchmark into a tool for practical figure-to-code applications, possibly integrating methodologies like RAG?

---

> ### Author Response · Authors · 2024-11-21
> **Response to Reviewer XRf9 (Part 1/2)**
>
> Thank you for your thorough and constructive feedback. We appreciate your recognition of our benchmark's motivation and its potential impact on advancing multimodal capabilities. We will provide detailed responses to your questions below.
>
> > **Q1: Can you provide more details about the manual annotation process, including annotator backgrounds, diversity, conflict resolution, and handling of incomplete figures?**
>
> R1:
> Thank you for requesting these important details about our data curation process. Below we elaborate on several key aspects of our annotation pipeline, which are also comprehensively documented in this revision (Appendix A.2).
>
> **Annotator Backgrounds and Diversity:** Our annotation process involves two distinct teams. The first team comprises 5 domain experts with diverse backgrounds: an information visualization expert with 3 years in scientific visualization, a digital media specialist focusing on HCI and multimedia design, an industrial design expert in user experience, a visual communication design expert specializing in graphic principles, and a computer science expert in data visualization programming. The second team consists of 6 Python annotators, all master's students in computer science with 6+ years of Python/matplotlib experience.
>
> **Conflict Resolution:** To address conflict resolution during the second phase in the "Diversity and Information Density Filtering" stage , we resolve disagreements through a voting mechanism, requiring at least 3 out of 5 votes for a figure’s inclusion when unanimous agreement is not reached.
>
> **Handling of Incomplete Figures:** In the "General Filtering" stage, we select Matplotlib-generated PDFs to ensure the figures can be reproduced using Python.  For figures where original data cannot be fully restored, our Python annotators approximate the data while maintaining visual consistency. They use Python 3.9.0 and matplotlib v3.8.4 to reproduce the figures.
>
> **Process Timeline and Output:** Our process begins with the "General Filtering" stage (1 week), reducing 174,100 figures to 15,800. This is followed by the "Diversity and Information Density Filtering" stage, carried out over two phases: the first phase (1 week) narrows the selection to 1,295 figures, and the second phase (3 days) further refines it to 279 figures. The "Manual Selection and Filling" stage (less than 1 week) adds 600 prototype figures from non-arXiv sources. Finally, the "Code and Instruction Writing" and "Data Augmentation" stages (1.5 months) culminate in the production of 4,800 (figure, code, instruction) triplets.
>
> > **Q2: What specific criteria are used to classify charts into easy, medium, and hard categories? Are these criteria subjective, or are they based on objective indicators?**
>
> R2:
> Thank you for this insightful feedback regarding complexity levels. To elucidate, our complexity assessment is based on two key factors:
>
> 1. **Visual Elements Complexity:** We meticulously evaluate the intricacy of each chart by assigning a comprehensive score to its visual elements. This includes chart types, data groups, markers, text elements, colour,  layout components and axes, among others. Each visual element is scored based on both its quantity and intricacy (1-3 points for low/medium/high, respectively).
>
> 2. **Implementation Complexity:** We consider the code length required for each chart as an additional complexity indicator. This factor captures the coding challenges involved in reproducing the chart figure.
>
> Thereby, charts in ChartMimic are then classified as:
>
> 1. **Easy:** Basic chart types with minimal visual elements and straightforward implementation (e.g., bar chart with a single colour, minimal text, and sparse data).
>
> 2. **Medium:** Charts with moderate visual element combinations or implementation complexity (e.g., bar chart with two subplots, grouped data,  multiple colours, moderate text annotations).
>
> 3. **Hard:** Charts demonstrating sophisticated visual elements or advanced implementation (e.g., bar chart with two subplots, diverging data, multiple colours, extensive text annotations and long code length).
>
> We have added detailed examples and scoring criteria in Appendix A.3 to enhance transparency of this classification system.
>
> > **Q3: How do you address the instability of LLM outputs in your evaluation metrics to ensure consistent and reliable results?**
>
> R3:
> GPT-4o Score is actually a stable evaluation metric. Like we detailed in Appendix D.1 "Stability of Evaluation with GPT-4o Score.", Lines 1815-1818. To check the stability of GPT-4o, we take GPT-4o evaluation for GPT-4o on Direct Mimic task 5 times, resulting in an average score of  83.4 with a standard deviation of 0.08.

---

> ### Author Response · Authors · 2024-11-21
> **Response to Reviewer XRf9 (Part 2/2)**
>
> > **Q4: Do the authors plan to make all the data in the benchmark publicly available after publication? I recommend that during the review stage, the authors could use anonymous GitHub repositories or similar platforms to offer more detailed information about the benchmark, including data and usage tutorials. This would significantly enhance the thoroughness of the benchmark evaluation.**
>
> R4:
> Thank you for your kind suggestion. Indeed, we have already provided our code and dataset in the Supplementary Material during the submission. We are fully committed to making all benchmark data publicly available after publication to contribute to the open-source community.
>
>
> > **Q5: Have you considered developing this benchmark into a tool for practical figure-to-code applications, possibly integrating methodologies like RAG?**
>
> R5:
> Your suggestion is very insightful. As we explained in our work, ChartMimic is not only aimed at assessing the visually grounded code generation capabilities of LMMs, but also holds significant practical value in assisting scientists and researchers in understanding, interpreting, and creating charts during the reading and writing of academic papers. We believe that developing this benchmark into a tool with methodologies like RAG represents a very promising direction, whether as a demo track work or as a Google Chrome plugin, which would better help people enhance the comprehension and presentation of data in scholarly communications.
>
> Thank you again for your detailed and thoughtful review. We appreciate your insights in helping us strengthen our work.

---

> > ### Comment · Reviewer_XRf9 · 2024-11-27
> >
> > Thank you to the author for their reply, which helped address some of my concerns. I will maintain my positive rating for this paper.

---

### Official Review · Reviewer_XbLj · 2024-11-03

**Soundness:** 3
**Presentation:** 3
**Contribution:** 4
**Rating:** 8
**Confidence:** 3

**Summary:**

This paper introduces a new dataset ChartMimic, which tests LMM's ability to replicate a variety of different plots based on a set of carefully constructed set of examples and metrics. This paper evaluates multiple LMMs on ChartMimic including both proprietary and open models, finding that most models with the exception of GPT-4o perform quite poorly on the tasks in ChatMimic but also benefit from additional natural language information for customization tasks. In terms of prompting, the authors find that self-reflection is effective.

**Strengths:**

- Overall the writing is easy to follow and the paper is well-organized.
- The work provides a clear dataset contribution that is not present in prior work as they overview in Table 2.
- The evaluation is generally well done, covering many different models, prompting strategies, incorporating a comparison to human judgments, and an error analysis of the best-performing model.

**Weaknesses:**

While already did a good job with the presentation of dataset construction and evaluation, here are a few more suggestions that could further improve clarity:
- Provide more examples of both direct and customized mimic tasks. In particular, I am curious how well-defined are instructions for customized mimic tasks are.
- Provide clearer definitions of easy, medium, and hard tasks. This kind of naming obfuscates the actual nature of the tasks themselves.
- Provide examples of GPT-4o scoring on different plots for sanity checks

Another suggestion is to conduct the same error analysis on open models, are there similar trends?

**Questions:**

Please address the specific questions raised in the weaknesses section.

---

> ### Author Response · Authors · 2024-11-21
> **Response to Reviewer XbLj**
>
> Thank you for your positive feedback! We greatly appreciate your recognition of our clear writing, dataset contribution, and comprehensive evaluation. We address your suggestions for improving clarity and providing additional examples below.
>
> > **W1: Provide more examples of both direct and customized mimic tasks. In particular, I am curious how well-defined are instructions for customized mimic tasks are.**
>
> R1:
> Thank you for these valuable suggestions for improving clarity. We have added more comprehensive examples of both direct and customized mimic tasks in revision (Appendix A.4) for better clarity.
>
> > **W2: Provide clearer definitions of easy, medium, and hard tasks. This kind of naming obfuscates the actual nature of the tasks themselves.**
>
> R2:
> Thank you for this insightful feedback regarding complexity levels. To elucidate, our complexity assessment is based on two key factors:
>
> 1. **Visual Elements Complexity:** We meticulously evaluate the intricacy of each chart by assigning a comprehensive score to its visual elements. This includes chart types, data groups, markers, text elements, colour,  layout components and axes, among others. Each visual element is scored based on both its quantity and intricacy (1-3 points for low/medium/high, respectively).
>
> 2. **Implementation Complexity:** We consider the code length required for each chart as an additional complexity indicator. This factor captures the coding challenges involved in reproducing the chart figure.
>
> Thereby, charts in ChartMimic are then classified as:
>
> 1. **Easy:** Basic chart types with minimal visual elements and straightforward implementation (e.g., bar chart with a single colour, minimal text, and sparse data).
>
> 2. **Medium:** Charts with moderate visual element combinations or implementation complexity (e.g., bar chart with two subplots, grouped data,  multiple colours, moderate text annotations).
>
> 3. **Hard:** Charts demonstrating sophisticated visual elements or advanced implementation (e.g., bar chart with two subplots, diverging data, multiple colours, extensive text annotations and long code length).
>
> We have added detailed examples and scoring criteria in Appendix A.3 to enhance transparency of this classification system.
>
>
> > **W3: Provide examples of GPT-4o scoring on different plots for sanity checks**
>
> R3:
> Thank you for these valuable suggestions. We have included detailed examples of GPT-4o's scoring examples in Appendix D.1 in the revision.
>
> > **W4: Another suggestion is to conduct the same error analysis on open models, are there similar trends?**
>
> R4:
> Thank you for raising this important question about error patterns in open models. We conduct detailed error analysis on representative open-weight models of different scales, sampling 100 cases for each model. For large-scale models such as InternVL2-Llama3-76B and InternVL2-26B, the error patterns align closely with those observed in GPT-4o. Specifically, Code-related Errors predominantly arise from dimensional inconsistencies, while Text-related Errors primarily manifest as missing elements. Type-related Errors mainly comprise missing components cases, and Color recognition demonstrates the capability to identify similar, if not identical, chromatic properties. In contrast, smaller-scale models like DeepSeek-VL-7B exhibit markedly different error distributions. Code-related issues show a significant increase in Access and Parameter Errors (approximately 53.7%), while Text-related errors demonstrate a higher prevalence of missing elements (approximately 66.7%). Type-related cases reveal a substantial increase in missing errors (approximately 78.4%). Moreover, the degradation in colour recognition is more severe, with the model failing to identify even similar colours. These findings indicate that while larger models encounter challenges similar to GPT-4o, smaller models exhibit more fundamental difficulties in both code generation and visual comprehension. We have added this content in Appendix I of revision.
>
> Thank you again for your constructive feedback and valuable suggestions. We sincerely appreciate your support.

---

### Official Review · Reviewer_LHtP · 2024-11-04

**Soundness:** 3
**Presentation:** 3
**Contribution:** 3
**Rating:** 6
**Confidence:** 3

**Summary:**

The paper proposes a new benchmark, ChartMimic, consisting of (figure, instruction, Python code) triplets covering 4,800 charts that are manually annotated by  the authors after being extracted from arxiv papers (along with other manually-curated sources). The authors propose five new automated metrics for evaluating proposed generations. The authors conduct an empirical study of existing models on the benchmark, demonstrating (among other findings) that existing SOTA GPT-4o attains a score of ~82.2%; that open-source models lag behind proprietary models; etc.

**Strengths:**

# Overall assessment

This is a relevant benchmark for a task likely of high interest and impact for both AI developers and AI users alike (chart generation). In general, the benchmark construction appears reasonable, but at times it is difficult to understand the details of the process and the authors do not provide quantitative details on the highly manual filtering process (these details seem important to assess the filtering procedure, since it relies extensively on manual curation by the authors). The empirical study is useful and shows some interesting, if unsurprising, results.

# Major comments

* In many cases, it would not be possible to fully reconstruct the original plot, as even at the pixel-level it would not be possible to fully recover exact data values. How is this handled in the "direct mimic" metric?

* Does "customized mimic" always *only* involve changing the data of a plot? If so, please state this explicitly. Furthermore, I would suggest changing the name to something more specific and clear (to me, this is really "direct mimic with new data").

* Related to the above point, it seems that there are two, completely independent, tasks being performed here. One is generating code to create a plot that exactly matches the aesthetic style of a given image. The other is recovering the *data* to be plotted, and applying the generated code to this data. I think it would be useful for the authors to also discuss this distinction in the paper, and to clearly motivate in which cases recovering the data is actually useful (in many cases, it seems we can simply take the data as given, and we would relaly just like to evaluate the correctness of the plotting code irrespective of the data it is applied to).

* In the filtering stage, the authors state that in the first phase "Charts with significant differences in complexity or information density are added to ensure diversity and effective communication". How exactly is this process conducted, given that the pool consists of "about 15,800 figures" prior to the first phase?

* There are many places in the dataset description where more detail would be helpful; for example "We preserve the intersection of their selections and finalize the set through a voting process." What voting process is used? How many charts were filtered at this stage? In general, it would be helpful to know precisely how many data points were considered, and how many filtered out, at each stage of the pipeline. Please add this data to the paper.

* Did the authors assess the impact of few-shot learning, chain-of-though prompting, or other methods on the benchmark tasks?

# Minor comments

* The abstract says the benchmark is comprised of "(figure, instruction, code)" triplets; but isn't data also a necessary component of the chart?

* The abstract gives an "average score" but it isn't at all clear what this score represents. Given that the authors propse a set of new metrics in the benchmark, perhaps they could more clearly describe the composite score being reported.

* Please include hyperlinks to relevant sections when referring to them (e.g. in L88-91).

* Figure 2 isn't particularly illuminating; in fact I find it quite confusing. I would suggest to modify the layout substantially and add clear individual labels to the different boxes.

# Typos etc.

* L40: ", such real-life scenarios have yet to be fully explored." This should be a separate sentence.
* L53: "Based on this, we define two tasks, Direct Mimic and Customized Mimic (Sec. 2.1), which utilizes" --> which utilize
* L141: "clarity and visual appeals" --> clarity and visual appeal

**Weaknesses:**

see above

**Questions:**

see above

---

> ### Author Response · Authors · 2024-11-21
> **Response to Reviewer LHtP (Part 1/3)**
>
> Thank you for your thorough and insightful feedback. We appreciate your recognition of ChartMimic's contribution as an impactful benchmark. We provide detailed responses to your specific comments below.
>
> > **Major1: In many cases, it would not be possible to fully reconstruct the original plot, as even at the pixel-level it would not be possible to fully recover exact data values. How is this handled in the "direct mimic" metric?**
>
> R1:
> Thank you for your careful observation. Indeed, fully reconstructing and evaluating the ground-truth figures (e.g., recovering exact data values and style), is challenging and widely acknowledged as a difficult task [1,2,3]. To address this, we choose to focus on key aspects that users care about most in practical ChartMimic scenarios. Therefore, we propose to evaluate based on four critical chart elements—text, layout, type, and colour [4,5]—along with the overall data trends and clarity of the chart .
>
> In detail, ChartMimic consists of two different level metrics (Sec 2.4):
>
> 1. Low-level metric : Using a code tracer, we evaluate the four key low-level elements (text, layout, type, and colour) to compute low-level similarity with the ground-truth figure. For example, when a data point is highlighted with text annotation, our metric ensures that such textual data information is considered in the evaluation.
>
> 2. High-level metric: Leveraging GPT-4o, we evaluate the overall data trends and the clarity of the chart to capture high-level similarity with the ground-truth figure.
>
> The above two-tiered evaluation metric enhances the efficacy and reliability of  evaluation, as evidenced by both high-level and low-level scores having correlations greater than 0.7 with human evaluation (Lines 442- 455).
>
> [1] A Comprehensive Study of Multimodal Large Language Models for Image Quality Assessment
>
> [2] MatPlotAgent: Method and Evaluation for LLM-Based Agentic Scientific Data Visualization
>
> [3] Design2Code: How Far Are We From Automating Front-End Engineering?
>
> [4] Chart Mining: A Survey of Methods for Automated Chart Analysis
>
> [5] ChartReader: A Unified Framework for Chart Derendering and Comprehension without Heuristic Rules
>
>
> > **Major2: Does "customized mimic" always only involve changing the data of a plot? If so, please state this explicitly. Furthermore, I would suggest changing the name to something more specific and clear (to me, this is really "direct mimic with new data").**
>
> R2:
> Thank you for your insightful feedback and for correctly understanding the concept of "customized mimic.". In ChartMimic, "customized mimic" specifically refers to reproducing the chart based on user-provided data. We have described this in Lines 101–106 of the original manuscript. Additionally, in the revised manuscript, we have also included examples of both "direct mimic" and "customized mimic" in Appendix A.4 to further illustrate the distinction and support comprehension.
>
>
> > **Major3: Related to the above point, it seems that there are two, completely independent, tasks being performed here. One is generating code to create a plot that exactly matches the aesthetic style of a given image. The other is recovering the data to be plotted, and applying the generated code to this data. I think it would be useful for the authors to also discuss this distinction in the paper, and to clearly motivate in which cases recovering the data is actually useful (in many cases, it seems we can simply take the data as given, and we would relaly just like to evaluate the correctness of the plotting code irrespective of the data it is applied to).**
>
> R3:
> Thank you for this insightful observation about the task distinction in our benchmark. As defined in Lines 101-106, the definitions of the two tasks are as follows:
>
> 1. Direct Mimic: The LMMs are tasked to generate code that reproduces the given chart.
>
> 2. Customized Mimic: The LMMs are tasked to generate code using user-provided data, while maintaining the aesthetic and design of the original chart.
>
> We designed these two tasks based on practical scenarios, as illustrated in Figure 1. In one case, a user might want to directly reproduce the style of an existing chart (Direct Mimic). Furthermore, they may wish to provide their own data to be combined with the chart’s style (Customized Mimic). Additionally, as mentioned in R1, ChartMimic does not require strictly recovering exact data points (a primary focus of Data Extraction [6], a separate and  important field). Instead, we emphasize aspects such as textual annotations, the number of groups, and overall trends in the data.
>
> To enhance clarity, in the revised manuscript, we have also included examples of both "direct mimic" and "customized mimic" in Appendix A.4 to further illustrate the distinction and support comprehension.
>
> [6] ChartOCR: Data Extraction from Charts Images via a Deep Hybrid Framework

---

> ### Author Response · Authors · 2024-11-21
> **Response to Reviewer LHtP (Part 2/3)**
>
> > **Major4: In the filtering stage, the authors state that in the first phase "Charts with significant differences in complexity or information density are added to ensure diversity and effective communication". How exactly is this process conducted, given that the pool consists of "about 15,800 figures" prior to the first phase?**
>
> R4:
> Thank you for your question regarding the first phase of "Diversity and Information Density Filtering" process. The filtering process involves an annotation team comprising five domain experts from information visualization, digital media, industrial design, visual communication design, and computer science.
> In the first phase, these experts conduct a  manual review of the 15,800 figures, focusing on visual diversity and information communication effectiveness. They maintain pools categorized by chart types. For each new figure, they assess its visual elements—such as layout, axes, line styles, marker styles, and colours—against existing figures in their corresponding type pool. If the figure as long as exhibits a distinctive difference in at least one of these aspects, it is retained; otherwise, it is excluded. This process results in 1,295 figures being selected for the second phase.
>
> We have added these details to Appendix A.2 for complete transparency of our data curation process.
>
> > **Major5: There are many places in the dataset description where more detail would be helpful; for example "We preserve the intersection of their selections and finalize the set through a voting process." What voting process is used? How many charts were filtered at this stage? In general, it would be helpful to know precisely how many data points were considered, and how many filtered out, at each stage of the pipeline. Please add this data to the paper.**
>
> R5:
> Thank you for highlighting the need for additional details regarding the data curation process, specifically for the filtering stages ("General Filtering" and "Diversity and Information Density Filtering").
>
> In the "General Filtering" stage, we start with an initial corpus of 174,100 figures. We filter these figures based on file format and generation method, retaining only Matplotlib-generated PDFs. This process refines the collection to 15,800 figures.
>
> In the "Diversity and Information Density Filtering" stage, the process involves two phases conducted by five domain experts from information visualization, digital media, industrial design, visual communication design, and computer science. In the first phase, these experts select 1,295  figures from the 15,800 figures as detailed in R4. In the second phase, they independently review the 1,295 figures and further select those figures they deem to exhibit significant distinctions and diversity. Figures selected by all experts are automatically included, while the remaining figures are subjected to a majority voting system requiring at least 3/5 votes for inclusion. This phase narrows the collection to a final set of 279 figures from the original 1,295.
>
> We have added these details to Appendix A.2 for complete transparency of our data curation process, including stages beyond filtering in our data pipeline.
>
> > **Major6: Did the authors assess the impact of few-shot learning, chain-of-though prompting, or other methods on the benchmark tasks?**
>
> R6:
> Thank you for this question.
>
> Regarding few-shot learning: We conducted few-shot learning experiments with GPT-4 on line charts and found no significant performance improvements. This can be attributed to the high diversity among charts in our task, making it challenging for few-shot examples to provide meaningful guidance. Moreover, few-shot learning is impractical for open-source models due to context length constraints - with a single in-context learning example (e.g., PIP charts) requiring over 2,000 tokens, while most open-source models have a 4,096-token limit.
>
> Concerning chain-of-thought prompting: We have already implemented this in our Hint-Enhanced approach, as detailed in lines 410-412: 'HintEnhanced uses prompt with chain-of-thought, explicitly prompting the LMMs to pay attention to important details (e.g., layout, type, text, etc).'
>
> As for other methods: We have thoroughly investigated SelfReflection and Scaffold Prompting approaches in Section 4.2, lines 405-431 and provide additional case study in Appendix E.3.

---

> ### Author Response · Authors · 2024-11-21
> **Response to Reviewer LHtP (Part 3/3)**
>
> > **Minor1: The abstract says the benchmark is comprised of "(figure, instruction, code)" triplets; but isn't data also a necessary component of the chart?**
>
> R7:
> Thank you for this astute observation. We would like to clarify that the data is indeed a crucial component of the chart and is integrated within our code structure. To be more specific, in our benchmark, the code is systematically organized into distinct logical blocks, where data is explicitly defined within the "Data Preparation" section.
>
> For example:
>
> ```python
> # Data Preparation
> sizes = [0.40, 0.25, 0.15, 0.10, 0.10]
> labels = ["Fiction\n40%", "Non-Fiction\n25%", ...]
> …
> ```
>
> We have also included examples of (figure, instruction, code) triplets for both "direct mimic" and "customized mimic" tasks in Appendix A.4, demonstrating our instruction format and code structure through representative cases.
>
> > **Minor2: The abstract gives an "average score" but it isn't at all clear what this score represents. Given that the authors propse a set of new metrics in the benchmark, perhaps they could more clearly describe the composite score being reported.**
>
> R8:
> Thank you for raising this important clarification. We revise the abstract to specify "...achieve an average score across Direct Mimic and Customized Mimic tasks of 82.2 and 61.6, respectively..."
>
> > **Minor3: Please include hyperlinks to relevant sections when referring to them (e.g. in L88-91).**
>
> R9:
> Thank you for this helpful suggestion. We have incorporated hyperlinks to all relevant sections in the revised manuscript.
>
> > **Minor4: Figure 2 isn't particularly illuminating; in fact I find it quite confusing. I would suggest to modify the layout substantially and add clear individual labels to the different boxes.**
>
> R10:
> We sincerely appreciate your constructive feedback. We have added labels for Direct Mimic and Customized Mimic in the Results & Evaluation part.  Please let us know if you have further questions.
>
>
> > **Typos etc.
> L40: ", such real-life scenarios have yet to be fully explored." This should be a separate sentence.
> L53: "Based on this, we define two tasks, Direct Mimic and Customized Mimic (Sec. 2.1), which utilizes" --> which utilize
> L141: "clarity and visual appeals" --> clarity and visual appeal**
>
> R11:
> Thank you for your careful observation. We have made the corresponding revisions in the paper.
>
> We sincerely appreciate all the valuable feedback that has helped improve this work.

---

> > ### Comment · Reviewer_LHtP · 2024-11-25
> >
> > I wanted to acknowledge the author response. Thank you for the detailed clarifications. As I think there are some positive contributions of the benchmark but still have some concerns about the metrics and the disentanglement of data vs. plots, I will retain my score.

---

### Author Response · Authors · 2024-11-22
**Updated Manuscript and Response to All Reviewers**

We sincerely thank all reviewers for their dedicated efforts during the review process and their valuable suggestions for improving our work. We have marked all revisions in **blue** in the updated manuscript. Below are the key revisions:

1. Abstract: Added clarification "... across Direct Mimic and Customized Mimic tasks..." as suggested by reviewer LHtP (lines 26-27)

2. Fixed grammar errors in lines 40, 53, and 142 following reviewer LHtP's comments

3. Added hyperlinks to relevant sections for better navigation as recommended by reviewer LHtP (lines 88-91)

4. Figure 2: Added labels for Direct Mimic and Customized Mimic in the Results & Evaluation part based on reviewer LHtP's suggestion

5. Provided clearer definitions of easy, medium, and hard tasks following reviewers XbLj and XRf9's comments (lines 197-199 and Appendix A.3)

6. Enhanced the data annotation pipeline description in Appendix A.2 as requested by reviewers LHtP and XRf9

7. Added task triplet examples in Appendix A.4 following reviewers LHtP and XbLj's suggestions

8. Included GPT-4o scoring examples in Appendix D.1 as suggested by reviewer XbLj

9. Added error analysis of open-weight models in Appendix I based on reviewer XbLj's feedback

10. Provided performance correlation with chart understanding and code generation benchmarks in Appendix G as suggested by reviewer gbor

We greatly appreciate the reviewers' constructive feedback and their recognition of our contribution.

---

### Meta-Review · Area_Chair_6CCd · 2024-12-20

**Metareview:**

This paper introduces ChartMimic, a new benchmark for evaluating large multimodal models (LMMs) on visually grounded code generation tasks. The reviewers are generally positive about the paper and argue for the acceptance. Specifically, the reviewers think the benchmark is novel and relevant, and agree that ChartMimic addresses an important and practical task relevant to both AI developers and users, and fills a gap in existing LMM evaluation datasets. The reviewers also commended the authors for conducting comprehensive evaluation, by proposing a multi-level evaluation metric that provides a holistic view of model performance. The reviewers also appreciated the thorough evaluation and analysis. The reviewers raised some concerns about clarity, which was sufficiently addressed during rebuttal.

**Additional Comments On Reviewer Discussion:**

The reviewers' concerns are mostly clarification questions. The authors provided detailed and comprehensive responses to the reviewers' concerns. They clarified the data curation process, provided more details about the complexity assessment criteria, and addressed the questions regarding the evaluation metrics. They also acknowledged the point about disentangling data and plotting, explaining their rationale for the current design and suggesting potential future directions.

---

### Decision · Program_Chairs · 2025-01-22

Accept (Poster)